# *NeuralSurv*: Deep Survival Analysis with Bayesian Uncertainty Quantification

**Mélodie Monod**[*]
Imperial College London
London, United Kingdom
melodie.monod18@imperial.ac.uk

**Alessandro Micheli**[*]
Imperial College London
London, United Kingdom
a.micheli19@imperial.ac.uk

**Samir Bhatt**
Imperial College London; University of Copenhagen
London, United Kingdom; Copenhagen, Denmark
s.bhatt@imperial.ac.uk

## Abstract

We introduce *NeuralSurv*, the first deep survival model to incorporate Bayesian uncertainty quantification. Our non-parametric, architecture-agnostic framework captures time-varying covariate–risk relationships in continuous time via a novel two-stage data-augmentation scheme, for which we establish theoretical guarantees. For efficient posterior inference, we introduce a mean-field variational algorithm with coordinate-ascent updates that scale linearly in model size. By locally linearizing the Bayesian neural network, we obtain full conjugacy and derive all coordinate updates in closed form. In experiments, *NeuralSurv* delivers superior calibration compared to state-of-the-art deep survival models, while matching or exceeding their discriminative performance across both synthetic benchmarks and real-world datasets. Our results demonstrate the value of Bayesian principles in data-scarce regimes by enhancing model calibration and providing robust, well-calibrated uncertainty estimates for the survival function.

## 1 Introduction

Survival analysis is a branch of statistics focused on the study of time-to-event data, usually called event times. This type of data appears in a wide range of applications such as medicine [33], engineering [35], and social sciences [42]. A key objective of survival analysis is to estimate the hazard function and the survival function that govern the distribution of event times.

Traditional survival models like the Cox proportional hazards model [10] and accelerated failure time models [8] have long delivered reliable inference under strong parametric assumptions. However, such assumptions may fail to adequately capture complex and evolving baseline hazards, especially when risk relationships vary over time. To overcome these limitations, recent work has begun incorporating modern machine-learning techniques [48], and in particular deep architectures [49, 24, 32], which can learn rich, hierarchical representations directly from data. Yet most deep-survival approaches remain purely frequentist, optimizing point-estimate losses and offering no coherent uncertainty quantification. In high-stakes settings like medicine, this lack of reliable uncertainty estimates can undermine trust and impede adoption.

Bayesian statistics, by contrast, inherently quantifies uncertainty: prior beliefs are combined with observed data to yield a posterior distribution over model parameters [15]. In survival analysis, Bayesian

---

[*]Equal contribution.

39th Conference on Neural Information Processing Systems (NeurIPS 2025).

methods can produce full posterior distributions for individual survival functions summarizable via credible intervals that communicate model confidence [21]. Traditional Bayesian survival tools, such as Gaussian processes (GPs) [13, 25], offer nonparametric flexibility and built-in uncertainty but often falter in high-dimensional settings due to scalability issues. To date, no method has combined the representational power of deep learning with full Bayesian uncertainty quantification in a scalable survival framework. Such a synthesis would hold the potential to learn complex, high-dimensional survival dynamics while retaining principled probabilistic interpretations.

In this work, we introduce *NeuralSurv*, an architecture-agnostic, Bayesian deep-learning framework for survival analysis which integrates with modern deep learning architectures. *NeuralSurv* leverages deep Neural Networks (NNs) to learn hierarchical representations from covariates and uses a principled variational inference framework to provide rigorous uncertainty quantification over the survival function. We develop a two-stage data-augmentation strategy using latent marked Poisson processes and Pólya–Gamma variables to enable exact continuous-time likelihood computation, and provide novel theoretical guarantees for this approach. By locally linearizing the Bayesian Neural Network (BNN), we achieve conjugacy and derive closed-form coordinate-ascent updates that scale linearly with network size.

Through extensive experiments on synthetic and real survival datasets, in data-scarce settings, *NeuralSurv* consistently delivers superior calibration compared to state-of-the-art deep survival models, and matches or exceeds their discriminative performance. Its Bayesian formulation captures epistemic uncertainty to prevent overfitting, while informative priors induce a soft regularization that yields smooth, plausible survival functions. The code to reproduce our experiments is available on the GitHub repository `https://github.com/MLGlobalHealth/neuralsurv` under the MIT License.

## 2  NeuralSurv

In this section, we outline the main assumptions underlying *NeuralSurv*. We begin by briefly reviewing key concepts in survival analysis. Survival analysis focuses on modeling time-to-event data. Let $T$ be a continuous nonnegative random variable with probability density $f$ and cumulative distribution function $F$, representing the time until a particular event occurs. Its *survival function* $S(t) = \mathbb{P}(T > t) = 1 - F(t)$ gives the probability of not experiencing the event by time $t$, while the *hazard function* $\lambda(t) = f(t)/S(t)$ represents the instantaneous risk of the event at time $t$, conditional on having survived up to time $t$. In practice, the event time may not be observed for all individuals, because some observations are subject to *right-censoring*, where the event has not *yet* occurred by the end of the observation period. For each observation $i = 1, \ldots, N$, denote the event time by $T_i$ and the censoring time by $C_i$. We observe $y_i = \min(T_i, C_i)$ and $\delta_i = \mathbb{1}_{\{T_i \leq C_i\}}$ where $y_i$ represents the observed time (which may correspond either to the event or to censoring), and $\delta_i$ indicates whether the event time was observed ($\delta_i = 1$) or the observation period was censored ($\delta_i = 0$). We assemble the dataset as $\mathcal{D} = \{(y_i, \delta_i) : i = 1, \ldots, N\}$. Each observation also carries a covariate vector $\mathbf{x}_i \in \mathbb{R}^p$, gathered into $\mathbf{X} = \{\mathbf{x}_i : i = 1, \ldots, N\}$. Throughout this paper, we assume that the censoring time $C_i$ is independent of the event time $T_i$ given $\mathbf{x}_i$ (known as non-informative censoring). Further details on survival analysis theory are provided in Appendix A.

### 2.1  Sigmoidal Hazard Function

Our goal is to model the hazard function $\lambda$, i.e. the instantaneous event rate at time $t$ conditional on survival to $t$ and covariates $\mathbf{x}$. We employ the sigmoid function $\sigma(z) = 1/(1 + \exp(-z))$, which maps real-valued inputs to the interval $(0, 1)$. The sigmoidal hazard model is constructed as the product of a normalized baseline hazard function ($\lambda_0$) and a modulation function ($\sigma$):

$$\lambda(t \mid \mathbf{x}; \phi, g(\cdot; \boldsymbol{\theta})) := \lambda_0(t, \mathbf{x}; \phi) \, \sigma(g(t, \mathbf{x}; \boldsymbol{\theta})), \tag{1}$$

where the *normalized baseline hazard* is given by

$$\lambda_0(t, \mathbf{x}; \phi) := \frac{\lambda_0(t; \phi)}{Z(t, \mathbf{x})}, \tag{2}$$

for the baseline hazard $\lambda_0 : \mathbb{R}_+ \to \mathbb{R}_+$, parametrized by $\phi \in \mathbb{R}_+$, and a normalization factor $Z(t, \mathbf{x})$ that depends on both time and covariates. The term $\lambda_0(t, \mathbf{x}; \phi)$ encodes our prior "best-guess" hazard

profile over time. The flexible function $g : \mathbb{R}_+ \times \mathbb{R}^p \to \mathbb{R}$, parametrized by $\boldsymbol{\theta} \in \mathbb{R}^m$, provides a data-driven adjustment: once passed through the sigmoid, it multiplicatively attenuates the baseline hazard, continuously scaling it between zero and $\lambda_0$. The normalization factor $Z(t, \mathbf{x})$ ensures that the overall hazard remains properly scaled after modulation by the sigmoidal function (see Section 2.3 for details). Modeling hazard and intensity functions using sigmoidal transformations is common in applications such as survival analysis [13] and point process models [52, 12, 45, 1, 2]. This approach is popular due to the balance it offers between modeling flexibility and analytical tractability.

## 2.2 Likelihood Distribution

Given the hazard function in (1), the likelihood density for the observation corresponding to the $i^{\text{th}}$ observation is given by:

$$p(y_i, \delta_i \mid \mathbf{x}_i, \phi, g(\cdot; \boldsymbol{\theta})) =$$
$$\left( \lambda_0(y_i, \mathbf{x}_i; \phi) \, \sigma(g(y_i, \mathbf{x}_i; \boldsymbol{\theta})) \right)^{\delta_i} \exp \left( - \int_0^{y_i} \lambda_0(t, \mathbf{x}_i; \phi) \, \sigma(g(t, \mathbf{x}_i; \boldsymbol{\theta})) \mathrm{d}t \right). \quad (3)$$

Assuming $(y_i, \delta_i)$ are i.i.d. conditional on $(\mathbf{x}_i, \phi, g(\cdot; \boldsymbol{\theta}))$, the full-sample likelihood is simply the product

$$p(\mathcal{D} \mid \mathbf{X}, \phi, g(\cdot; \boldsymbol{\theta})) = \prod_{i=1}^{N} p(y_i, \delta_i \mid \mathbf{x}_i, \phi, g(\cdot; \boldsymbol{\theta})). \quad (4)$$

## 2.3 Prior Distributions

**Prior Distribution on $\boldsymbol{\theta}$.** We assume that $g(\cdot; \boldsymbol{\theta})$ is a BNN parameterized by $\boldsymbol{\theta}$. Furthermore, denote by $\mathbf{I}_m$ the $m \times m$ identity matrix. We place the following isotropic Gaussian prior with zero mean and identity covariance over the NN weights

$$p_{\boldsymbol{\theta}}(\boldsymbol{\theta}) = \mathcal{N}(\boldsymbol{\theta}; \mathbf{0}, \mathbf{I}_m). \quad (5)$$

This common choice [6] assumes weights are independently distributed and centered around zero, acting as an uninformative yet regularizing prior that discourages large weights and helps prevent overfitting via shrinkage.

**Prior Distribution on $\phi$.** We adopt a Weibull-type baseline hazard

$$\lambda_0(t; \phi) = \phi t^{\rho-1}, \quad p_\phi(\phi) = \text{Gamma}(\alpha_0, \beta_0), \quad \rho > 0 \text{ fixed}, \quad (6)$$

where $\alpha_0$ is the shape and $\beta_0$ is the rate of the Gamma distribution. The Weibull-type baseline hazard (6) is the hazard of a Weibull distribution, a common choice in survival analysis [13]. When $\rho = 1$, $\lambda_0(t; \phi)$ becomes constant and the baseline hazard reduces to the hazard of the Exponential distribution.

**Normalization Factor $Z$.** We define the normalization factor introduced in (2) as

$$Z(t, \mathbf{x}) := \mathbb{E}_{\boldsymbol{\theta} \sim p_{\boldsymbol{\theta}}} \left[ \sigma(g(t, \mathbf{x}; \boldsymbol{\theta})) \right]$$

and refer the reader to Appendix D for further details on how it is computed. Introducing this normalization factor ensures that the prior mean of the sigmoidal hazard in (1) coincides with the prior mean of the baseline hazard, i.e.

$$\mathbb{E}_{\phi \sim p_\phi, \boldsymbol{\theta} \sim p_{\boldsymbol{\theta}}} \left[ \lambda(t \mid \mathbf{x}; \phi, g(\cdot; \boldsymbol{\theta})) \right] = \mathbb{E}_{\phi \sim p_\phi} \left[ \lambda_0(t; \phi) \right].$$

This approach, similar to the technique used in [13], centers the distribution around the baseline hazard $\lambda_0(t; \phi)$, favouring hazard trajectories that remain close to this prior "best-guess" profile while still permitting data-driven deviations. Notice that if $g(\cdot; \boldsymbol{\theta})$ has a fully connected architecture, then $Z(t, \mathbf{x}) \equiv \frac{1}{2}$ for all $(t, \mathbf{x})$, resulting in the same normalization factor value as in [13].

## 2.4 Posterior Distribution

Let $p(\phi, \boldsymbol{\theta} \mid \mathcal{D}, \mathbf{X})$ denote the posterior density over the parameters $\phi$ and $\boldsymbol{\theta}$, defined with respect to the product measure $\mathrm{d}\phi \times \mathrm{d}\boldsymbol{\theta}$. By Bayes' rule, this posterior is proportional (up to normalization) to

$$p(\phi, \boldsymbol{\theta} \mid \mathcal{D}, \mathbf{X}) \propto p(\mathcal{D} \mid \mathbf{X}, \phi, g(\cdot; \boldsymbol{\theta})) \, p_\phi(\phi) \, p_{\boldsymbol{\theta}}(\boldsymbol{\theta}). \tag{7}$$

The posterior in (7) is generally intractable to compute for three reasons. First, its normalization constant is unavailable in closed form. Second, the likelihood from (3) requires evaluating $N$ integrals, none of which admits an analytic solution. Finally, the sigmoid in (1) introduces an extra nonlinearity, rendering inference even more analytically challenging.

# 3 Data Augmentation Strategy

In this section, we present a data augmentation scheme that leverages the properties of Poisson processes and Pólya-Gamma random variables. Specifically, Poisson processes help overcome the challenges associated with computing the integrals of the continuous-time function to evaluate the likelihood, while the Pólya-Gamma random variables allow for exact handling of the sigmoid nonlinearity without relying on analytic approximations. This combined approach allows us to efficiently perform posterior inference from the model without resorting to discretization.

This approach builds on analogous strategies previously applied in other settings, including Bayesian inference for Sigmoid Gaussian Cox Processes [12], nonparametric Hawkes processes [52], and, in the case of Pólya–Gamma augmentation alone, mutually regressive point processes [4]. To the best of our knowledge, this is the first application of such a data augmentation strategy in the context of survival analysis. Furthermore, we are the first to provide a rigorous theoretical framework that establishes the validity of a method belonging to this broader class of augmentation-based approaches (see Theorem 3.1).

Detailed reviews of Pólya-Gamma random variables and Poisson processes are provided in Appendices B and C, respectively.

## 3.1 Pólya-Gamma Augmentation Scheme

A primary challenge in our model arises from the sigmoid function, whose inherent nonlinearity complicates the posterior inference. To overcome this, we adopt the Pólya-Gamma data augmentation scheme introduced in [38]. The key insight of this approach is that the sigmoid function can be represented in terms of Pólya-Gamma random variables. Define the function

$$f(\omega, z) := \frac{z}{2} - \frac{z^2}{2}\omega - \log(2). \tag{8}$$

Then, the following identity holds:

$$\sigma(z) = \int_0^\infty e^{f(\omega, z)} p_{\mathrm{PG}}(\omega \mid 1, 0) \mathrm{d}\omega, \tag{9}$$

where $p_{\mathrm{PG}}(\omega \mid 1, 0)$ denotes the density of a Pólya-Gamma random variable with parameters $(1, 0)$.

Since our model considers $N$ observations, we apply this augmentation scheme to each data point. Accordingly, we introduce $N$ independent Pólya-Gamma random variables, denoted by $\boldsymbol{\omega} = \{\omega_i\}_{i=1}^N$, each distributed according to $p_\omega(\omega_i) = p_{\mathrm{PG}}(\omega_i \mid 1, 0)$ and with a joint density

$$p_{\boldsymbol{\omega}}(\boldsymbol{\omega}) = \prod_{i=1}^N p_\omega(\omega_i) = \prod_{i=1}^N p_{\mathrm{PG}}(\omega_i \mid 1, 0). \tag{10}$$

## 3.2 Poisson Process Augmentation Scheme

Evaluating the likelihood in (3) requires computing $N$ integrals involving a sample function drawn from the BNN prior. This integral is generally analytically intractable, due to the nonparametric and highly non-linear nature of BNN sample paths. To address this, we leverage a Poisson process–based

data augmentation scheme, drawing inspiration from methodologies proposed in [12, 52]. By substituting the sigmoid identity from (9), the intractable integral for the $i^{\text{th}}$ data point becomes

$$\int_0^{y_i} \lambda_0(t, \mathbf{x}_i; \phi)\, \sigma(g(t, \mathbf{x}_i; \boldsymbol{\theta}))\mathrm{d}t =$$

$$\int_0^{y_i} \int_0^{\infty} \left( 1 - e^{f(\omega, -g(t,\mathbf{x}_i;\boldsymbol{\theta}))} \right) \lambda_0(t, \mathbf{x}_i; \phi) p_{\text{PG}}(\omega \mid 1, 0)\mathrm{d}\omega\mathrm{d}t, \quad (11)$$

where $p_{\text{PG}}(\omega|1,0)$ is the density of a Pólya-Gamma random variable. The key insight here is that this double integral can be expressed as an expectation over a marked Poisson process.

Before proceeding further, we briefly review the concept of a marked Poisson process. A marked Poisson process extends the standard Poisson process by associating each event (or location) with an additional random variable known as a mark. In our case, each event occurs at time $t$ and is accompanied by a positive mark $\omega$. With this in mind, consider the space $[0, y_i] \times \mathbb{R}_+$ which consists of points $(t, \omega)$ where $t \in [0, y_i]$ and $\omega \in \mathbb{R}_+$. We then denote by $\Psi_i$ a marked Poisson process on $[0, y_i] \times \mathbb{R}_+$ with intensity

$$\lambda_i(t, \omega; \phi) := \lambda_0(t, \mathbf{x}_i; \phi)\, p_{\text{PG}}(\omega \mid 1, 0), \quad (t, \omega) \in [0, y_i] \times \mathbb{R}_+. \quad (12)$$

Under suitable assumptions on the BNN $g(\cdot; \boldsymbol{\theta})$, Campbell's theorem allows us to express the integral in (11) as

$$\exp\left( -\int_0^{y_i} \int_0^{\infty} \left( 1 - e^{f(\omega, -g(t,\mathbf{x}_i;\boldsymbol{\theta}))} \right) \lambda_i(t, \omega; \phi)\mathrm{d}\omega\mathrm{d}t \right) =$$

$$\mathbb{E}_{\Psi_i \sim \mathbb{P}_{\Psi_i|\phi}} \left[ \prod_{(t,\omega)_j \in \Psi_i} e^{f(\omega_j, -g(t_j,\mathbf{x}_i;\boldsymbol{\theta}))} \right], \quad (13)$$

where $\mathbb{P}_{\Psi_i|\phi}$ is the path measure of the process $\Psi_i$. In (13), we take the convention that an empty product equals 1. Equation (13) corresponds to the term with the intractable integral on the right-hand side of (3). This representation enables us to avoid time discretization, allowing an exact and efficient evaluation of the integral. Since our model involves $N$ observations, we apply this augmentation scheme to each data point by introducing $N$ independent marked Poisson processes, denoted by $\boldsymbol{\Psi} = \{\Psi_i\}_{i=1}^N$.

### 3.3 Augmented Likelihood

Leveraging both the Pólya–Gamma and the marked Poisson process augmentation schemes, we can reformulate the likelihood given in (3) in a tractable way. With these auxiliary variables, we define the *augmented likelihood* density for the $i^{\text{th}}$ observation as

$$p\left( y_i, \delta_i \mid \mathbf{x}_i, \phi, g(\cdot; \boldsymbol{\theta}), \omega_i, \Psi_i \right) :=$$

$$\left( \lambda_0(y_i, \mathbf{x}_i; \phi) e^{f(\omega_i, g(y_i, \mathbf{x}_i; \boldsymbol{\theta}))} \right)^{\delta_i} \left( \prod_{(t,\omega)_j \in \Psi_i} e^{f(\omega_j, -g(t_j, \mathbf{x}_i; \boldsymbol{\theta}))} \right), \quad (14)$$

where $f(\omega, z)$ was defined in (8). The following proposition formalizes the data augmentation scheme.

**Theorem 3.1** (Data Augmentation). *Assume for each $i = 1, \ldots, N$ that the function $g(\cdot, \mathbf{x}_i; \cdot) \in C([0, y_i] \times \mathbb{R}^m)$. Let $p(y_i, \delta_i \mid \mathbf{x}_i, \phi, g(\cdot; \boldsymbol{\theta}))$ be the likelihood density given in (3). Additionally, let $p\left( y_i, \delta_i \mid \mathbf{x}_i, \phi, g(\cdot; \boldsymbol{\theta}), \omega_i, \Psi_i \right)$ be the augmented likelihood density defined in (14). Then,*

$$p(y_i, \delta_i \mid \mathbf{x}_i, \phi, g(\cdot; \boldsymbol{\theta})) = \mathbb{E}_{\omega_i \sim p_\omega, \Psi_i \sim \mathbb{P}_{\Psi_i|\phi}} \left[ p\left( y_i, \delta_i \mid \mathbf{x}_i, \phi, g(\cdot; \boldsymbol{\theta}), \omega_i, \Psi_i \right) \right].$$

The proof of Theorem 3.1 is postponed to Appendix N.1. Existing augmentation schemes approaches [12, 52, 4] do not offer any theoretical guarantees regarding the validity of the methodology. In contrast, Theorem 3.1 provides the first rigorous theoretical framework that establishes the soundness of a method within this class of data augmentation techniques.

Using the assumption from Section 2.2 that $(y_i, \delta_i)$ are i.i.d. conditional on $(\mathbf{x}_i, \phi, g(\cdot; \boldsymbol{\theta}))$, and given the structure of the data augmentation, we observe that $(y_i, \delta_i)$ are conditionally independent of $\omega_j$ and $\Psi_j$ for all $j \neq i$. As a result, the full-sample augmented likelihood factorizes as a simple product:

$$p(\mathcal{D} \mid \mathbf{X}, \phi, g(\cdot; \boldsymbol{\theta}), \boldsymbol{\omega}, \boldsymbol{\Psi}) = \prod_{i=1}^{N} p(y_i, \delta_i \mid \mathbf{x}_i, \phi, g(\cdot; \boldsymbol{\theta}), \omega_i, \Psi_i). \tag{15}$$

# 4 Variational Inference in the Augmented Space

In this section, we develop a novel variational inference algorithm based on this augmentation scheme.

## 4.1 Variational Mean–Field Approximation

Computing the posterior distribution $\mathbb{P}(\phi, \boldsymbol{\theta}, \boldsymbol{\omega}, \boldsymbol{\Psi} \mid \mathcal{D}, \mathbf{X})$ is analytically intractable because its normalization constant is unavailable in closed form. We consider a variational inference algorithm that aims to find an approximating variational distribution $\mathbb{Q}(\phi, \boldsymbol{\theta}, \boldsymbol{\omega}, \boldsymbol{\Psi})$ that minimizes the KL divergence from the true posterior distribution.

To make the optimization tractable, we restrict our search to distributions that satisfy the following mean-field factorization:

$$\mathbb{Q}(\phi, \boldsymbol{\theta}, \boldsymbol{\omega}, \boldsymbol{\Psi}) = \mathbb{Q}_\phi(\phi) \times \mathbb{Q}_{\boldsymbol{\theta}}(\boldsymbol{\theta}) \times \mathbb{Q}_{\boldsymbol{\omega}}(\boldsymbol{\omega}) \times \mathbb{Q}_{\boldsymbol{\Psi}}(\boldsymbol{\Psi}).$$

Here, we take $\mathbb{Q}_\phi(\phi)$, $\mathbb{Q}_{\boldsymbol{\theta}}(\boldsymbol{\theta})$ and $\mathbb{Q}_{\boldsymbol{\omega}}(\boldsymbol{\omega})$ to admit densities $q_\phi(\phi)$, $q_{\boldsymbol{\theta}}(\boldsymbol{\theta})$ and $q_{\boldsymbol{\omega}}(\boldsymbol{\omega})$ with respect to the Lebesgue measures $\mathrm{d}\phi$, $\mathrm{d}\boldsymbol{\theta}$ and $\mathrm{d}\boldsymbol{\omega}$. The remaining factor $\mathbb{Q}_{\boldsymbol{\Psi}}(\boldsymbol{\Psi})$ is a measure on the space of marked point-process paths, which does not admit a density with respect to the Lebesgue measures (see, e.g., a similar discussion for GPs in [34]).

To handle this within the variational inference framework, we must introduce a reference measure $\mathbb{P}_{\boldsymbol{\Psi},*}$, which plays the role of a "Lebesgue-like" base measure on path space (see Definition E.1 for details). We then assume our variational law $\mathbb{Q}_{\boldsymbol{\Psi}}$ is absolutely continuous with respect to $\mathbb{P}_{\boldsymbol{\Psi},*}$, so that it admits a strictly positive Radon–Nikodym derivative $\frac{\mathrm{d}\mathbb{Q}_{\boldsymbol{\Psi}}}{\mathrm{d}\mathbb{P}_{\boldsymbol{\Psi},*}}$ which satisfies the normalization $\mathbb{E}_{\boldsymbol{\Psi} \sim \mathbb{P}_{\boldsymbol{\Psi},*}}[\frac{\mathrm{d}\mathbb{Q}_{\boldsymbol{\Psi}}}{\mathrm{d}\mathbb{P}_{\boldsymbol{\Psi},*}}(\boldsymbol{\Psi})] = 1$. These conditions ensure that $\mathbb{Q}_{\boldsymbol{\Psi}}$ is a valid probability measure on the space of marked point-process paths (see Appendix E for further technical details).

This formulation enables us to express the KL divergence between the variational distribution and the true posterior in terms of the ELBO:

$$D_{\mathrm{KL}}(\mathbb{Q}(\phi, \boldsymbol{\theta}, \boldsymbol{\omega}, \boldsymbol{\Psi}) \,||\, \mathbb{P}(\phi, \boldsymbol{\theta}, \boldsymbol{\omega}, \boldsymbol{\Psi} \mid \mathcal{D}, \mathbf{X})) = -\mathcal{L}_{\mathrm{ELBO}}(g) + \mathrm{const}, \tag{16}$$

where the ELBO is defined as

$$\mathcal{L}_{\mathrm{ELBO}}(g) :=$$

$$\mathbb{E}_{\phi \sim q_\phi, \boldsymbol{\theta} \sim q_{\boldsymbol{\theta}}, \boldsymbol{\omega} \sim q_{\boldsymbol{\omega}}, \boldsymbol{\Psi} \sim \mathbb{Q}_{\boldsymbol{\Psi}}} \left[ \log \frac{p(\mathcal{D} \mid \phi, g(\cdot; \boldsymbol{\theta}), \boldsymbol{\omega}, \boldsymbol{\Psi}) \, p_\phi(\phi) p_{\boldsymbol{\theta}}(\boldsymbol{\theta}) p_{\boldsymbol{\omega}}(\boldsymbol{\omega}) \frac{\mathrm{d}\mathbb{P}_{\boldsymbol{\Psi}|\phi}}{\mathrm{d}\mathbb{P}_{\boldsymbol{\Psi},*}}(\boldsymbol{\Psi})}{q_\phi(\phi) q_{\boldsymbol{\theta}}(\boldsymbol{\theta}) q_{\boldsymbol{\omega}}(\boldsymbol{\omega}) \frac{\mathrm{d}\mathbb{Q}_{\boldsymbol{\Psi}}}{\mathrm{d}\mathbb{P}_{\boldsymbol{\Psi},*}}(\boldsymbol{\Psi})} \right] \tag{17}$$

and where $\frac{\mathrm{d}\mathbb{P}_{\boldsymbol{\Psi}|\phi}}{\mathrm{d}\mathbb{P}_{\boldsymbol{\Psi},*}}$ is the Radon-Nykodim derivative of the true conditional law $\mathbb{P}_{\boldsymbol{\Psi}|\phi}$ with respect to $\mathbb{P}_{\boldsymbol{\Psi},*}$. From (16), it follows that minimizing the KL divergence is equivalent to maximizing the ELBO.

## 4.2 Local Linearization of the Bayesian Neural Network

A crucial insight is that the data augmentation strategy transforms the intractable likelihood density in (3) into a form that is conditionally Gaussian, as shown below:

$$p(y_i, \delta_i \mid \mathbf{x}_i, \phi, g(\cdot; \boldsymbol{\theta}), \omega_i, \Psi_i) \propto$$

$$\exp\left(\delta_i \frac{g(y_i, \mathbf{x}_i; \boldsymbol{\theta})}{2} - \delta_i \frac{g(y_i, \mathbf{x}_i; \boldsymbol{\theta})^2}{2} \omega_i\right) \exp\left(\sum_{(t,\omega)_j \in \Psi_i} \frac{g(t_j, \mathbf{x}_i; \boldsymbol{\theta})}{2} - \frac{g(t_j, \mathbf{x}_i; \boldsymbol{\theta})^2}{2} \omega_j\right).$$

This transformation is particularly advantageous when placing a GP prior on $g(\cdot; \boldsymbol{\theta})$, as it induces conjugacy in the model. Conjugacy is crucial for variational inference because it enables efficient computation of the ELBO (17), which involves taking expectations over the distribution of $\boldsymbol{\theta}$. However, when $g(\cdot; \boldsymbol{\theta})$ is a BNN, these expectations generally lack closed-form solutions, making exact Bayesian updates intractable. As a result, we seek to approximate $g(\cdot; \boldsymbol{\theta})$ in a way that retains the expressive power of NNs while preserving Gaussian conjugacy to enable tractable inference.

We adopt the *local linearization* approximation introduced in [22]. This approach approximates the BNN $g(\cdot; \boldsymbol{\theta})$ using a first-order Taylor expansion around a reference point $\boldsymbol{\theta}^\star$:

$$g(t, \mathbf{x}; \boldsymbol{\theta}) \approx g^{\text{lin}}(t, \mathbf{x}; \boldsymbol{\theta}) := g(t, \mathbf{x}; \boldsymbol{\theta}^\star) + \mathbf{J}_{\boldsymbol{\theta}^\star}(t, \mathbf{x})^\top (\boldsymbol{\theta} - \boldsymbol{\theta}^\star), \tag{18}$$

where $[\mathbf{J}_{\boldsymbol{\theta}}(t, \mathbf{x})]_j = \frac{\partial g(t, \mathbf{x}; \boldsymbol{\theta})}{\partial \theta_j}$ is the Jacobian of the BNN with respect to the parameters $\boldsymbol{\theta}$. Following [22], we select $\boldsymbol{\theta}^\star = \boldsymbol{\theta}_{\text{MAP}}$ as the maximum a posteriori (MAP) estimate, which is defined as:

$$(\boldsymbol{\theta}_{\text{MAP}}, \phi_{\text{MAP}}) := \arg\max_{\boldsymbol{\theta}, \phi} p(\boldsymbol{\theta}, \phi \mid \mathcal{D}, \mathbf{X}), \tag{19}$$

where $p(\boldsymbol{\theta}, \phi \mid \mathcal{D}, \mathbf{X})$ is the posterior density defined in (7). By centering the linearization at $\boldsymbol{\theta}_{\text{MAP}}$, we ensure maximal approximation accuracy precisely where Bayesian inference is most sensitive: in the high-probability region of the posterior that dominates both parameter uncertainty quantification and predictive distributions. The procedure used to obtain the MAP estimates of (19) is detailed in Appendix F. Under the assumption of a Gaussian prior on the BNN parameters (5), the local linearization induces the GP prior

$$g^{\text{lin}} \sim \mathcal{GP}(\mu, \kappa)$$

with mean function $\mu$ and and covariance function $\kappa$ given by:

$$\mu(t, \mathbf{x}) := g(t, \mathbf{x}; \boldsymbol{\theta}_{\text{MAP}}) + \mathbf{J}_{\boldsymbol{\theta}_{\text{MAP}}}(t, \mathbf{x})^\top (\mathbb{E}_{\boldsymbol{\theta} \sim p_{\boldsymbol{\theta}}}[\boldsymbol{\theta}] - \boldsymbol{\theta}_{\text{MAP}})$$

$$\kappa((t, \mathbf{x}), (t', \mathbf{x}')) := \mathbf{J}_{\boldsymbol{\theta}_{\text{MAP}}}(t, \mathbf{x}) \mathbf{J}_{\boldsymbol{\theta}_{\text{MAP}}}(t', \mathbf{x}')^\top.$$

Incorporating this approximation into our variational framework allows us to exploit Gaussian conjugacy for fast, closed-form updates, while still preserving the flexibility of NNs. Concretely, we take a Taylor expansion of the ELBO around $g^{\text{lin}}$ and, by truncating at the lowest order term, obtain the simple approximation

$$\mathcal{L}_{\text{ELBO}}(g) \approx \mathcal{L}_{\text{ELBO}}(g^{\text{lin}}).$$

Our approach is analogous to the method introduced in [47, Section 3.2], where the authors apply Delta Method Variational Inference by approximating the ELBO around a fixed point in parameter space. In contrast, we extend this idea by approximating the ELBO around a reference function $g^{\text{lin}}$, rather than a fixed point.

### 4.3 Coordinate Ascent Variational Inference

We adopt a Coordinate Ascent Variational Inference (CAVI) approach, allowing us to draw on standard results from variational inference (see, e.g., [5, Chapter 10.1]). In this framework, the optimal variational distributions are derived by maximizing the linearized ELBO, $\mathcal{L}_{\text{ELBO}}(g^{\text{lin}})$, with each distribution depending on the current state of the others. The algorithm proceeds by cyclically updating each variational distribution while keeping the others fixed. This iterative process progressively refines the optimal variational distributions, ultimately leading to the best possible approximation of the posterior distribution. A complete derivation of each optimal variational distribution is provided in Appendix G while the complete CAVI algorithm is presented in Appendix H.

At the $k^{\text{th}}$ iteration the optimal variational distributions for the parameters $\phi$ and $\boldsymbol{\theta}$ are given by

$$q_\phi^{(k)}(\phi) = \text{Gamma}\left(\tilde{\alpha}^{(k)}, \tilde{\beta}\right), \quad q_{\boldsymbol{\theta}}^{(k)}(\boldsymbol{\theta}) = \mathcal{N}\left(\tilde{\boldsymbol{\mu}}^{(k)}, \tilde{\boldsymbol{\Sigma}}^{(k)}\right),$$

where $(\tilde{\alpha}^{(k)}, \tilde{\beta})$ and $(\tilde{\boldsymbol{\mu}}^{(k)}, \tilde{\boldsymbol{\Sigma}}^{(k)})$ are given in Appendix G.3 and G.4, respectively. At the $k^{\text{th}}$ iteration, the optimal update for the auxiliary parameters $\boldsymbol{\omega}$ is given by

$$q_{\boldsymbol{\omega}}^{(k)}(\boldsymbol{\omega}) = \prod_{i=1}^{N} p_{\text{PG}}(\omega_i \mid 1, \tilde{c}_i^{(k)}),$$

where $\tilde{c}_i^{(k)}$ is given in Appendix G.1. Finally, at the $k^{\text{th}}$ iteration, the optimal variational law $\mathbb{Q}_{\Psi}^{(k)}$ is the probability measure under which each $\Psi_i$ ($i = 1, \ldots, N$) is a marked Poisson process on $[0, y_i] \times \mathbb{R}_+$ with intensity function $\lambda_i^{\mathbb{Q},(k)}$, as given in Appendix G.2.

It is important to emphasize that we did not impose a specific form on the variational distributions; for example, we did not assume $q_{\boldsymbol{\theta}}(\boldsymbol{\theta})$ to be Gaussian. Instead, we derived our results by minimizing the KL divergence over the full space of distributions. This contrasts with methods that fix a parametric form and use the reparameterization trick with Monte Carlo gradient estimates.

Finally, in Appendix I, we demonstrate that, by exploiting the Woodbury matrix identity, our inference updates require only $\mathcal{O}(m)$ time complexity ($m$ is the number of weights in the NN architecture). This linear scaling renders our Bayesian framework feasible for contemporary large-scale deep neural architectures, which are well suited to model high-dimensional data.

## 5 Experiments

Details on the experimental setup, including dataset descriptions, benchmark methods specifications and evaluation metrics definitions are provided in Appendix J. Moreover, the implementation details for *NeuralSurv* are provided in Appendix K.

To comprehensively evaluate *NeuralSurv*, we compare its performance against the following set of benchmark models: *MTLR* [51], *DeepHit* [32], *DeepSurv* [24], *Logistic Hazard* [16], *CoxTime* [29], *CoxCC* [29], *PMF* [28], *PCHazard* [28], *BCESurv* [30], and *DySurv* [36], *Sumo-Net* [43] and *DQS* [50]. A detailed overview of these models is provided in Appendix L and summarized in Table A2. Except for *DySurv*, which employs an autoencoder framework, we adopt the same NN architecture across all benchmark models and *NeuralSurv* to parameterize the hazard function. For *DySurv*, we use the original autoencoder architecture specified in its implementation.

We assess discriminative performance using the Antolini's concordance index (C-index) [3], and evaluate model calibration with the inverse probability of censoring weighting (IPCW) integrated Brier score (IBS) [17], the Distribution Calibration (D-Calibration) [18], and the Kaplan–Meier Calibration (KM Calibration) [9]. The C-index evaluates how well a model performs by measuring the concordance between the rankings of the predicted event times and the true event times. The C-index ranges from 0 to 1, where higher values indicate better discriminative performance; a value of 0.5 corresponds to random guessing. Similar to the mean squared error, the Brier score (BS) assesses the accuracy of an estimated survival function at some time $t$. The IPCW are observation-specific weights that account for censoring in survival data, ensuring that the BS remains unbiased. The IPCW IBS is the integral of the IPCW BS over the observational period. The D-Calibration test bins each individual's predicted survival probability at their observed event time into equal-width bins over and applies a $\chi^2$ test to assess whether those predicted probabilities are uniform across bins. A well-calibrated model should yield a non-significant p-value. The KM-Calibration procedure compares the average predicted survival curve with the Kaplan–Meier estimate. For the KM-Calibration, the closer the two curves align, the better calibrated the model, where 0 indicates perfect calibration and 1 indicates maximal miscalibration and a random prediction yields 0.25. The C-index and the IPCW IBS metrics are computed using the `TorchSurv` package [37]. The D-Calibration and KM-Calibration metrics are computed using the `SurvivalEVAL` package [41].

### 5.1 Synthetic Data Experiment

In this section, we present experiments conducted on synthetic data. The experimental setup was inspired by [13] and constitutes a broadly applicable evaluation benchmark. We simulate the training sets with increasing sizes $N = 25, 50, 100$ and $150$ samples where the event time was drawn from two distributions: $p_0(T) = \text{LogNormal}(3, 0.8^2)$ and $p_1(T) = \text{LogNormal}(3.5, 1^2)$. Each observation includes a covariate indicating whether the event time is sampled from $p_0$ or $p_1$, along with three additional noisy covariates generated from a standard normal distribution. The censoring times are drawn from an exponential distribution with a rate of $0.025$ yielding an average censoring rate of 54% across the four synthetic datasets. The test set is generated using the same data-generating process, fixed to 100 observations, and held constant across all experiments.

Figure 1 presents the true survival function alongside the predicted functions from *NeuralSurv* and the two top-performing benchmark models, selected based on IPCW IBS. Each panel represents

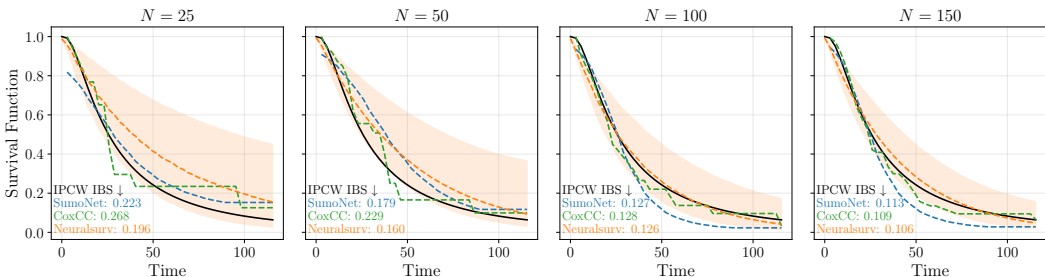

Figure 1: Comparison of the true survival function (black) with the estimated survival functions from *NeuralSurv* and the two top-performing benchmark models (colored) on synthetic data. The time axis is truncated at the maximum observed event time in the training data. Each panel represents a different training set size. The IPCW IBS score is reported for each method in each panel, with lower values indicating better predictive accuracy. *NeuralSurv* estimates the full posterior over survival functions, and the 90% credible interval is shown as a ribbon around its estimate.

a different training set size. As the number of training samples increases, the predicted survival functions match more closely the true survival function. The results show that *NeuralSurv* consistently ranks as the best method according to IPCW IBS, and its predictive accuracy improves with larger sample sizes. Beyond its competitive performance, *NeuralSurv* also provides Bayesian credible intervals, offering uncertainty estimates for survival probabilities, an important feature absent in deep learning benchmark models. Notably, these credible intervals appropriately narrow as more data becomes available, demonstrating well-calibrated uncertainty quantification. Corresponding C-index, IPCW IBS score, D-Calibration p-values, and KM-Calibration scores for all methods are reported in Tables A3-A4.

## 5.2   Real Survival Data Experiments

To comprehensively evaluate *NeuralSurv*, we conduct experiments on eight real survival datasets: the chemotherapy for colon cancer (COLON), the Molecular Taxonomy of Breast Cancer International Consortium (METABRIC), the Rotterdam and German Breast Cancer Study Group (GBSG), the National Wilm's Tumor Study (NWTCO), the Worcester Heart Attack Study (WHAS), the Study to Understand Prognoses and Preferences for Outcomes and Risks of Treatment (SUPPORT), the Veterans administration Lung Cancer trial (VLC) and the Sac 3 simulation study. Each dataset is subsampled to 125 observations to highlight the advantages of a Bayesian approach in data-scarce regimes. The data is randomly partitioned into five equally sized folds, with each fold serving as a distinct train/test split, comprising 100 training samples and 25 test samples per fold.

Table 1 presents the C-index and IPCW IBS on the held-out test sets for three representative datasets, while results for the remaining datasets, as well as D-Calibration and KM-Calibration, are shown in Tables A5–A6. Across eight datasets, *NeuralSurv* achieves the best IPCW IBS score on seven, highlighting superior overall calibration compared to benchmarks. It consistently passes D-calibration, together with *Sumo-Net* as the only benchmark achieving this result, while it ranks fifth in KM-calibration. This strength arises from its Bayesian framework, which naturally models uncertainty and provides effective regularization in data-scarce settings. Beyond calibration, *NeuralSurv* also demonstrates strong discriminative performance, achieving the best C-index in four datasets and the second best in three.

An ablation study using a larger training set of 250 observations is presented in Tables A7-A8. *NeuralSurv* continues to outperform benchmark methods under this setting in terms of calibration performance demonstrating the robustness of the method to training size. Furthermore, we also include results from traditional survival models, such as the Cox Proportional Hazards model [10], the Weibull Accelerated Failure Time model [8], the Random Survival Forest [23], and the Survival Support Vector Machine [40] in Tables A9-A10. These models often achieve strong performance in data-scarce regimes. However, they are not designed to leverage high-dimensional or complex feature representations, which limits their applicability in modern deep learning contexts. Our focus remains on evaluating deep survival methods that can scale with data complexity, but we include these

|  | COLON | | METABRIC | | GBSG | |
| --- | --- | --- | --- | --- | --- | --- |
| Method | C-index ↑ | IPCW IBS ↓ | C-index ↑ | IPCW IBS ↓ | C-index ↑ | IPCW IBS ↓ |
| MTLR [51] | 0.562 | 0.298 | 0.548 | 0.279 | 0.602 | 0.273 |
| DeepHit [32] | 0.478 | 0.28 | 0.511 | 0.243 | 0.578 | 0.309 |
| DeepSurv [24] | 0.572 | 0.326 | 0.523 | 0.289 | 0.618 | 0.252 |
| Logistic Hazard [16] | 0.490 | 0.321 | 0.541 | 0.317 | 0.618 | 0.296 |
| CoxTime [29] | 0.578 | 0.277 | 0.533 | 0.307 | 0.599 | 0.285 |
| CoxCC [29] | 0.584 | 0.289 | 0.575 | 0.257 | 0.646 | 0.240 |
| PMF [28] | 0.509 | 0.324 | 0.440 | 0.336 | 0.655 | 0.250 |
| PCHazard [28] | 0.538 | 0.297 | 0.541 | 0.291 | 0.609 | 0.249 |
| BCESurv [30] | 0.491 | 0.302 | **0.616** | 0.277 | 0.581 | 0.273 |
| DySurv [36] | 0.488 | 0.536 | 0.561 | 0.465 | 0.572 | 0.485 |
| Sumo-Net [43] | 0.485 | 0.241 | 0.447 | 0.223 | 0.476 | 0.250 |
| DQS [50] | 0.635 | 0.246 | 0.564 | 0.261 | 0.611 | 0.229 |
| NeuralSurv (Ours) | **0.671** | **0.218** | 0.584 | **0.212** | **0.657** | **0.188** |

Table 1: Performance comparison of deep survival models over five different train/test splits of each dataset. The best results for each metric are shown in bold, and the second-best results are underlined. ↑ indicates higher is better; ↓ indicates lower is better.

classical baselines for reference and completeness. A prior sensitivity analysis for the parameter $\phi$, using priors with double and half the original variance, is presented in Table A11. While the posterior distributions under different priors largely overlapped, their central tendencies occasionally differed, indicating mild to moderate sensitivity to the choice of prior. Incorporating prior information remains important, as it helps strike a principled balance between model flexibility and regularization.

## 6 Conclusion

We propose the first fully Bayesian framework for deep survival analysis that models time-varying relationships between covariates and risk. On both synthetic and real-world datasets, in data-scarce regimes, our method consistently achieves better calibration than state-of-the-art deep survival models and matches or surpasses their discriminative performance. In contrast to previous approaches in deep survival analysis, which are either constrained to discrete-time settings [51, 32, 16, 28, 30, 36] or lack the ability to provide Bayesian uncertainty quantification [51, 32, 24, 16, 29, 30, 36], *NeuralSurv* introduces a continuous-time modeling framework that naturally incorporates Bayesian inference, enabling both accurate survival predictions and well-calibrated uncertainty estimates.

Pólya–Gamma and Poisson data-augmentation schemes (Section 3.3) have been extensively employed with standard Gaussian process models [12, 52]. Likewise, the local linearization of Bayesian neural networks, which yields a Gaussian process–based approximation, (Section 4.2), is a well established technique [22]. To our knowledge, this work is the first to integrate these two approaches into a unified framework that capitalizes on Gaussian process conjugacy. By combining these methods, we contribute a novel inference strategy at the intersection of Bayesian deep learning and Gaussian process modeling.

Despite its strengths, *NeuralSurv* relies on three key simplifying assumptions. First, we assume a sigmoidal hazard function, a choice shared by prior work (e.g., [13, 25]), which may not capture all risk dynamics. Second, our mean-field variational inference treats parameters $\phi$ and $\boldsymbol{\theta}$ as independent, ignoring posterior correlations. Third, we linearize the network around the MAP estimate to enforce conjugacy. In real-world settings, however, the true posterior can be multimodal and strongly correlated, so this local, factorized approximation may overlook secondary modes or misestimate joint uncertainty.

Concerning the computational efficiency of our method, the coordinate-ascent updates scale linearly with network size but still require full-dataset passes each iteration. For very large cohorts, this becomes a bottleneck. Extending the algorithm to use stochastic or mini-batch updates would preserve conjugacy benefits while improving scalability.

We believe that *NeuralSurv* has the potential to make a positive societal impact. For instance, as healthcare data becomes increasingly diverse, there is a growing need for models that can handle multimodal data within time-to-event analyses effectively. *NeuralSurv* represents an important first step toward accommodating such data within a Bayesian deep learning framework.

## Acknowledgments and Disclosure of Funding

SB acknowledges funding from the MRC Centre for Global Infectious Disease Analysis (reference MR/X020258/1), funded by the UK Medical Research Council (MRC). This UK funded award is carried out in the frame of the Global Health EDCTP3 Joint Undertaking. SB acknowledges support from the Novo Nordisk Foundation via The Novo Nordisk Young Investigator Award (NNF20OC0059309). SB acknowledges the Danish National Research Foundation (DNRF160) through the chair grant. SB acknowledges support from The Eric and Wendy Schmidt Fund For Strategic Innovation via the Schmidt Polymath Award (G-22-63345) which also supports AM and MM.

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

# A  Review of Survival Analysis

This appendix offers a concise summary of the survival-analysis framework on which our approach is built. For an in-depth review, the reader is referred to [31].

Survival data for each observation consist of three components:

- **Feature vector**: A covariate vector $\mathbf{x} \in \mathbb{R}^p$ capturing baseline characteristics;
- **Event time**: a nonnegative random variable $T$ measuring the time from baseline to the occurrence of the event of interest;
- **Event indicator**: A binary variable $\delta$, which takes the value 1 if the event is observed, and 0 if the event is not observed within the observational period. In the latter case, the observation's data is said to be right-censored, meaning that the only available information is the time of the last follow-up before the event could occur.

To handle censoring uniformly, we introduce a censoring time $C$ and record the observed time $y = \min(T, C)$. The event indicator can then be written succinctly as $\delta = \mathbb{1}_{\{T \leq C\}}$. Throughout, we assume noninformative right-censoring, i.e. conditional on the covariates, the censoring time is independent of the event time: $C \perp T \mid \mathbf{x}$. Conditional on $\mathbf{x}$, we let the event time $T$ have cumulative distribution function $F(t \mid \mathbf{x})$ and probability density function $f(t|\mathbf{x})$ such that

$$F(t \mid \mathbf{x}) = \mathbb{P}(T \leq t \mid \mathbf{x}) = \int_0^t f(s \mid \mathbf{x}) \, \mathrm{d}s$$

for $t \in [0, \infty)$. The survival function gives the probability of remaining event-free beyond time $t$:

$$S(t \mid \mathbf{x}) := \mathbb{P}(T > t \mid \mathbf{x}) = 1 - F(t \mid \mathbf{x}) = \int_t^\infty f(s \mid \mathbf{x})\mathrm{d}s,$$

for $t \in [0, \infty)$. An important modeling quantity is the hazard function, which represents the instantaneous event rate at time $t$ given survival up to $t$:

$$\lambda(t \mid \mathbf{x}) := \lim_{\Delta t \to 0} \frac{\mathbb{P}\big(t \leq T < t + \Delta t \mid T \geq t, \mathbf{x}\big)}{\Delta t} = \frac{f(t \mid \mathbf{x})}{S(t \mid \mathbf{x})}.$$

Equivalently,

$$\lambda(t \mid \mathbf{x}) = -\frac{\mathrm{d}}{\mathrm{d}t} \log S(t \mid \mathbf{x}),$$

so that the survival function can be written in terms of the hazard:

$$S(t \mid \mathbf{x}) = \exp\Big(-\int_0^t \lambda(s \mid \mathbf{x}) \, \mathrm{d}s\Big).$$

# B  Review of Pólya-Gamma Random Variables

We follow [38] in defining the family of Pólya–Gamma distributions and their properties.

**Definition B.1** (Pólya–Gamma Distribution). A random variable $\omega$ is said to follow a *Pólya–Gamma distribution* with parameters $b > 0$ and $c \in \mathbb{R}$, denoted by $\omega \sim \mathrm{PG}(b, c)$, if

$$\omega \stackrel{d}{=} \frac{1}{2\pi^2} \sum_{k=1}^{\infty} \frac{g_k}{\left(k - \frac{1}{2}\right)^2 + \frac{c^2}{4\pi^2}}, \quad \text{with } g_k \stackrel{\text{i.i.d.}}{\sim} \text{Gamma}(b, 1). \tag{A1}$$

The following result expresses the reciprocal of the hyperbolic cosine function raised to the power $b$ as an infinite Gaussian mixture. This representation is central to connecting the Pólya–Gamma density with a parameter $c \neq 0$ to the case when $c = 0$.

**Proposition B.2.** *The reciprocal of the hyperbolic cosine raised to the power $b$ can be represented as an infinite Gaussian mixture:*

$$\left[\cosh\left(\frac{c}{2}\right)\right]^{-b} = \int_0^\infty \exp\left(-\frac{c^2}{2}\omega\right) p_{PG}(\omega \mid b, 0)\, d\omega.$$

Notice that Proposition B.2 can also be read as providing a closed-form expression for the expectation $\mathbb{E}_{\omega \sim p_{PG}(\omega \mid b,0)}\left[\exp(-\frac{c^2}{2}\omega)\right]$. Building on this representation, we can relate the density function of a Pólya–Gamma random variable with a non-zero parameter $c$ through an exponential tilting of the Pólya–Gamma random density with $c = 0$. This connection is summarized in the next proposition.

**Proposition B.3.** *The Pólya–Gamma density* (A1) *can be re-written in the form*

$$p_{PG}(\omega \mid b, c) = \exp\left(-\frac{c^2}{2}\omega\right) (\cosh(c/2))^b \, p_{PG}(\omega \mid b, 0). \tag{A2}$$

The previous propositions not only establish key representations of the Pólya–Gamma density but also facilitate the derivation of its moment properties. In particular, one can derive the moment generating function, from which the first moment follows directly. This is captured in the next result.

**Proposition B.4.** *Let $p_{PG}(\omega \mid b, c)$ denote the density function of the random variable $\omega \sim PG(b, c)$, with $b > 0$ and $c \in \mathbb{R}$. Using Propositions B.2 and B.3, the moment generating function is given by*

$$\int_0^\infty e^{\xi\omega} p_{PG}(\omega \mid b, c)\, d\omega = \frac{\cosh^b(c/2)}{\cosh^b\left(\frac{1}{2}\sqrt{c^2 - 2\xi}\right)}. \tag{A3}$$

*In particular, the first moment is obtained by differentiating* (A3) *with respect to $\xi$ at $\xi = 0$:*

$$\mathbb{E}_{\omega \sim p_{PG}(\omega \mid b,c)}[\omega] = \frac{b}{2c} \tanh\left(\frac{c}{2}\right). \tag{A4}$$

Finally, the following theorem illustrates how the Pólya–Gamma distribution can be used to derive useful integral identities.

**Theorem B.5.** *Let $p_{PG}(\omega \mid b, 0)$ denote the density function of the random variable $\omega \sim PG(b, 0)$ with $b > 0$. Then, for all $a \in \mathbb{R}$, the following integral identity holds:*

$$\frac{e^{\psi a}}{(1 + e^\psi)^b} = 2^{-b} e^{\kappa\psi} \int_0^\infty \exp\left(-\frac{\omega\psi^2}{2}\right) p_{PG}(\omega \mid b, 0)\, d\omega,$$

*where $\kappa = a - \frac{b}{2}$.*

The following corollary is a direct application of Theorem B.5.

**Corollary B.6.** *Let $f(\omega, z) := \frac{z}{2} - \frac{z^2}{2}\omega - \log(2)$. Then,*

$$\sigma(z) = \frac{e^{\frac{z}{2}}}{2\cosh(\frac{z}{2})} = \int_0^\infty e^{f(\omega, z)} p_{PG}(\omega \mid 1, 0)\, d\omega. \tag{A5}$$

# C  Review of Poisson Processes

This appendix briefly summarizes the properties of a Poisson process that are most relevant to our analysis. For a more comprehensive treatment, see Chapters 3 and 5 of [26].

**Definition C.1** (Poisson Process). Let $\mathcal{Z}$ be a measurable space. A random countable subset

$$\Psi = \{z \in \mathcal{Z}\}$$

is said to be a *Poisson process* on $\mathcal{Z}$ if it satisfies the following properties:

1. **Independence:** For any sequence of disjoint subsets $\{\mathcal{Z}_k \subset \mathcal{Z}\}_{k=1}^K$, the counts

$$N(\mathcal{Z}_k) = |\Psi \cap \mathcal{Z}_k|$$

   are mutually independent.

2. **Poisson Counts:** For each measurable subset $\mathcal{Z}_k \subset \mathcal{Z}$, the count $N(\mathcal{Z}_k)$ is Poisson distributed with mean

$$\int_{\mathcal{Z}_k} \lambda(z)\, dz,$$

   where $\lambda : \mathcal{Z} \to \mathbb{R}_+$ is the intensity function.

Given a point process $\Psi$, we denote its path measure — that is, the probability measure induced on its sample-path space — by $\mathbb{P}_\Psi$. If the intensity function $\lambda(z)$ is constant, $\lambda(z) \equiv \lambda$, then $\Psi$ is called *homogeneous*; otherwise, it is *inhomogeneous*. We now extend the concept of a Poisson process by incorporating additional random attributes, known as *marks*.

**Definition C.2** (Marked Poisson Process). Let $\Psi = \{z \in \mathcal{Z}\}$ be a Poisson process on $\mathcal{Z}$ with intensity function $\lambda : \mathcal{Z} \to \mathbb{R}_+$. Suppose that for each point $z$, associate a random variable $\omega$, such that $\omega \sim p_{\omega|z}(\omega|z)$, taking values in some space $\mathcal{M}$. Then the collection

$$\Psi_\mathcal{M} = \{(z, \omega) \in \mathcal{Z} \times \mathcal{M}\}$$

defines a Poisson process on the product space $\mathcal{Z} \times \mathcal{M}$. The resulting process is known as a *marked Poisson process* with intensity

$$\lambda(z, \omega) = \lambda(z)\, p_{\omega|z}(\omega|z).$$

Next, we present Campbell's Theorem, which describes the law of sums taken over the points of a Poisson process (see [26, Sec. 3.2]).

**Theorem C.3** (Campbell's Theorem). *Let $\Psi_\mathcal{M}$ be a marked Poisson process on $\mathcal{Z} \times \mathcal{M}$ with intensity function $\lambda(z, \omega)$ and let $f : \mathcal{Z} \times \mathcal{M} \to \mathbb{R}$ be measurable. Then the sum*

$$H(\Psi_\mathcal{M}) = \sum_{(z,\omega)_j \in \Psi_\mathcal{M}} f(z_j, \omega_j)$$

*is absolutely convergent with probability one if and only if*

$$\int_{\mathcal{Z} \times \mathcal{M}} \min(|f(z, \omega)|, 1) \lambda(z, \omega) \mathrm{d}z \mathrm{d}\omega < \infty.$$

*If this condition holds, then*

$$\mathbb{E}_{\Psi_\mathcal{M} \sim \mathbb{P}_{\Psi_\mathcal{M}}} \left[ e^{sH(\Psi_\mathcal{M})} \right] = \exp\left( \int_{\mathcal{Z} \times \mathcal{M}} (e^{sf(z,\omega)} - 1) \lambda(z, \omega) \mathrm{d}z \mathrm{d}\omega \right)$$

*for any $s \in \mathbb{C}$ for which the integral on the right converges. Moreover*

$$\mathbb{E}_{\Psi_\mathcal{M} \sim \mathbb{P}_{\Psi_\mathcal{M}}} [H(\Psi_\mathcal{M})] = \int_{\mathcal{Z} \times \mathcal{M}} f(z, \omega) \lambda(z, \omega) \mathrm{d}z \mathrm{d}\omega$$

*in the sense that the expectation exists if and only if the integral converges.*

# D  Obtaining the Normalization Factor $Z(t, \mathbf{x})$

In this appendix we derive an efficient approximation for the normalization factor

$$Z(t, \mathbf{x}) = \mathbb{E}_{\boldsymbol{\theta} \sim p_{\boldsymbol{\theta}}} \left[ \sigma(g^{\text{lin}}(t, \mathbf{x}; \boldsymbol{\theta})) \right], \tag{A6}$$

which is needed when computing the CAVI optimal updates (see Appendix G).

Recall from (5) that $\boldsymbol{\theta}$ has the following prior distribution

$$\boldsymbol{\theta} \sim \mathcal{N}(\mathbf{0}, \mathbf{I}_m),$$

where $\mathbf{I}_m$ is the $m \times m$ identity matrix. Moreover, recall from Section 4.2 that we approximate the network output $g(t, \mathbf{x}; \boldsymbol{\theta})$ around some reference $\boldsymbol{\theta}^\star$ by its first-order linearization

$$g^{\text{lin}}(t, \mathbf{x}; \boldsymbol{\theta}) := g(t, \mathbf{x}; \boldsymbol{\theta}^\star) + \mathbf{J}_{\boldsymbol{\theta}^\star}(t, \mathbf{x})^\top (\boldsymbol{\theta} - \boldsymbol{\theta}^\star),$$

where $\mathbf{J}_{\boldsymbol{\theta}^\star}(t, \mathbf{x})$ denotes the Jacobian of $g(t, \mathbf{x}; \boldsymbol{\theta})$ with respect to $\boldsymbol{\theta}$. Because $\boldsymbol{\theta}$ is Gaussian, the linearized output is also Gaussian:

$$g^{\text{lin}}(t, \mathbf{x}; \boldsymbol{\theta}) \sim \mathcal{N}\left( g(t, \mathbf{x}; \boldsymbol{\theta}^\star) - \mathbf{J}_{\boldsymbol{\theta}^\star}(t, \mathbf{x})^\top \boldsymbol{\theta}^\star, \|\mathbf{J}_{\boldsymbol{\theta}^\star}(t, \mathbf{x})\|_2^2 \right).$$

In order to approximate $Z(t, \mathbf{x})$ we wish to leverage a well-known asymptotic approximation. Specifically, for a normal random variable $X \sim \mathcal{N}(\mu, \sigma^2)$ it holds that

$$\mathbb{E}_{X \sim \mathcal{N}(\mu, \sigma^2)}[\sigma(X)] \approx \sigma\left( \frac{\mu}{\sqrt{1 + \frac{\pi}{8}\sigma^2}} \right). \tag{A7}$$

We can apply the result in (A7) to the normal random variable $g^{\text{lin}}(t, \mathbf{x}; \boldsymbol{\theta})$ and approximate $Z(t, \mathbf{x})$ as

$$Z(t, \mathbf{x}) \approx \sigma\left( \frac{g(t, \mathbf{x}; \boldsymbol{\theta}^\star) - \mathbf{J}_{\boldsymbol{\theta}^\star}(t, \mathbf{x})^\top \boldsymbol{\theta}^\star}{\sqrt{1 + \frac{\pi}{8}\|\mathbf{J}_{\boldsymbol{\theta}^\star}(t, \mathbf{x})\|_2^2}} \right).$$

Since in (A6) we are taking the expectation under the prior $p_{\boldsymbol{\theta}}(\boldsymbol{\theta})$, it is natural to linearize around the prior mean, therefore, we set $\boldsymbol{\theta}^\star = \mathbf{0}$.

## E    Combining Variational Inference with Poisson Processes

In this appendix, we outline how our variational-inference framework integrates marked Poisson processes — an essential part in the mean-field variational approximation of Section 4.1. For a fully rigorous, measure-theoretic treatment, the reader is referred to Brémaud's text [7]. Our development relies in particular on Theorem T10 in Chapter VIII of that book, which shows how the law of a marked Poisson process arises via a change of measure using the appropriate Radon–Nikodym derivative.

We begin by fixing a reference measure on path space:

**Definition E.1** (Reference measure $\mathbb{P}_{\mathbf{\Psi},*}$). Let $\mathbf{\Psi} = (\Psi_1, \ldots, \Psi_N)$ be $N$ independent marked Poisson processes, where each $\Psi_i$ is defined on the product space $[0, y_i] \times \mathbb{R}_+$. We define $\mathbb{P}_{\mathbf{\Psi},*}$ to be their joint law where each $\Psi_i$ has intensity

$$\lambda_{*,i}(t, \omega) = t^{\rho-1} p_{\mathrm{PG}}(\omega \mid 1, 0) \quad \text{for all } (t, \omega) \in [0, y_i] \times \mathbb{R}_+. \tag{A8}$$

Next, let $\gamma_i^{\mathbb{Q}}(t)$ be a deterministic function on $[0, y_i]$ and let $h_i^{\mathbb{Q}}(t, \omega)$ be a deterministic density on $[0, y_i] \times \mathbb{R}_+$ satisfying

$$\int_0^\infty h_i^{\mathbb{Q}}(t, \omega) p_{\mathrm{PG}}(\omega|1,0) \mathrm{d}\omega = 1 \quad \text{and} \quad \int_0^{y_i} \gamma_i^{\mathbb{Q}}(t) t^{\rho-1} \mathrm{d}t < \infty \tag{A9}$$

for all $t \in [0, y_i]$ and $i = 1, \ldots, N$. It is convenient to introduce the function

$$\lambda_i^{\mathbb{Q}}(t, \omega) := \gamma_i^{\mathbb{Q}}(t) h_i^{\mathbb{Q}}(t, \omega) \lambda_{*,i}(t, \omega) \quad \text{for all } (t, \omega) \in [0, y_i] \times \mathbb{R}_+,$$

as well as the functional

$$L(\mathbf{\Psi}) := \prod_{i=1}^N \left( \prod_{(t,\omega)_j \in \Psi_i} \gamma_i^{\mathbb{Q}}(t_j) h_i^{\mathbb{Q}}(t_j, \omega_j) \right) \exp \left( \int_0^{y_i} \int_0^\infty \left( \lambda_{*,i}(t, \omega) - \lambda_i^{\mathbb{Q}}(t, \omega) \right) \mathrm{d}\omega \mathrm{d}t \right).$$

By Theorem T10.b [7, Chapter VIII], whenever $\mathbb{E}_{\mathbf{\Psi} \sim \mathbb{P}_{\mathbf{\Psi},*}}[L(\mathbf{\Psi})] = 1$, the measure $\mathbb{Q}_{\mathbf{\Psi}}(\mathbf{\Psi})$ defined by $\frac{\mathrm{d}\mathbb{Q}_{\mathbf{\Psi}}}{\mathrm{d}\mathbb{P}_{\mathbf{\Psi},*}}(\mathbf{\Psi}) = L(\mathbf{\Psi})$ is exactly the law under which each $\Psi_i$ is a marked Poisson process on $[0, y_i] \times \mathbb{R}_+$ with intensity $\lambda_i^{\mathbb{Q}}(t, \omega)$. The above result underpins the analysis in Appendix G.2, where we show that the optimal variational measure $\mathbb{Q}_{\mathbf{\Psi}}$ coincides with the law of a collection of independent marked Poisson processes.

Finally, the measure $\mathbb{P}_{\mathbf{\Psi}|\phi}$ also admits a Radon-Nykodim derivative with respect to $\mathbb{P}_{\mathbf{\Psi},*}$ which is given by :

$$\frac{\mathrm{d}\mathbb{P}_{\mathbf{\Psi}|\phi}}{\mathrm{d}\mathbb{P}_{\mathbf{\Psi},*}}(\mathbf{\Psi}) = \prod_{i=1}^N \left( \prod_{(t,\omega)_j \in \Psi_i} \frac{\phi}{Z(t_j, \mathbf{x}_i)} \right) \exp \left( \int_0^{y_i} \int_0^\infty (\lambda_{*,i}(t, \omega) - \lambda_i(t, \omega; \phi)) \mathrm{d}\omega \mathrm{d}t \right). \tag{A10}$$

Notice that $\frac{\phi}{Z(t_j, \mathbf{x}_i)} = \frac{\lambda_i(t_j, \omega_j; \phi)}{\lambda_{*,i}(t_j, \omega_j)}$, i.e. the ratio of the intensities of $\mathbb{P}_{\mathbf{\Psi}|\phi}$ and $\mathbb{P}_{\mathbf{\Psi},*}$.

# F   Obtaining the Maximum a Posteriori $\theta_{\text{MAP}}$

We seek the maximum a posteriori (MAP) estimates

$$(\boldsymbol{\theta}_{\text{MAP}}, \phi_{\text{MAP}}) = \arg\max_{\boldsymbol{\theta}, \phi} \log p(\boldsymbol{\theta}, \phi \mid \mathcal{D}, \mathbf{X}).$$

Applying Bayes' rule gives the following expression for the posterior density

$$\log p(\boldsymbol{\theta}, \phi \mid \mathcal{D}, \mathbf{X}) \propto \log p(\mathcal{D} \mid \mathbf{X}, g(\cdot; \boldsymbol{\theta}), \phi) + \log p_{\boldsymbol{\theta}}(\boldsymbol{\theta}) + \log p_{\phi}(\phi),$$

where the likelihood density $p(\mathcal{D} \mid \mathbf{X}, g(\cdot; \boldsymbol{\theta}), \phi)$ and the prior densities $p_{\boldsymbol{\theta}}(\boldsymbol{\theta})$ and $p_{\phi}(\phi)$ are specified in Equations (4), (5), and (6), respectively. Since the log likelihood distribution is intractable, direct optimization of the posterior distribution is infeasible.

## F.1   Approximating the Log Likelihood distribution

**Variational Mean–Field Approximation.**     Our aim is to approximate the log likelihood density $\log p(\mathcal{D} \mid \mathbf{X}, g(\cdot; \boldsymbol{\theta}), \phi)$. In order to do so, we introduce a variational distribution $\breve{\mathbb{Q}}(\boldsymbol{\omega}, \boldsymbol{\Psi} \mid \boldsymbol{\theta}, \phi)$ to approximate the true distribution $\mathbb{P}(\boldsymbol{\omega}, \boldsymbol{\Psi} \mid \mathcal{D}, \mathbf{X}, g(\cdot; \boldsymbol{\theta}), \phi)$. Such variational distribution differs from the one used for full-model inference in Section 4.3 because it is conditioned on the values of $\boldsymbol{\theta}$ and $\phi$. Hence, we adopt the notation $\breve{\mathbb{Q}}$ (instead of $\mathbb{Q}$) to highlight this difference. We restrict our search to distributions that satisfy the following *mean-field* factorization:

$$\breve{\mathbb{Q}}(\boldsymbol{\omega}, \boldsymbol{\Psi} \mid \boldsymbol{\theta}, \phi) = \breve{\mathbb{Q}}_{\boldsymbol{\omega} \mid \boldsymbol{\theta}, \phi}(\boldsymbol{\omega} \mid \boldsymbol{\theta}, \phi) \times \breve{\mathbb{Q}}_{\boldsymbol{\Psi} \mid \boldsymbol{\theta}, \phi}(\boldsymbol{\Psi} \mid \boldsymbol{\theta}, \phi).$$

Here, we take $\breve{\mathbb{Q}}_{\boldsymbol{\omega} \mid \boldsymbol{\theta}, \phi}(\boldsymbol{\omega} \mid \boldsymbol{\theta}, \phi)$ to admit the density $\breve{q}_{\boldsymbol{\omega} \mid \boldsymbol{\theta}, \phi}(\boldsymbol{\omega} \mid \boldsymbol{\theta}, \phi)$ with respect to the Lebesgue measure $d\boldsymbol{\omega}$.

For the marked point process component, we assume that the variational law $\breve{\mathbb{Q}}_{\boldsymbol{\Psi} \mid \boldsymbol{\theta}, \phi}$ is absolutely continuous with respect to $\mathbb{P}_{\boldsymbol{\Psi}, *}$, so that it admits a strictly positive Radon–Nikodym derivative $\frac{d\breve{\mathbb{Q}}_{\boldsymbol{\Psi} \mid \boldsymbol{\theta}, \phi}}{d\mathbb{P}_{\boldsymbol{\Psi}, *}}$ which satisfies the normalization $\mathbb{E}_{\boldsymbol{\Psi} \sim \mathbb{P}_{\boldsymbol{\Psi}, *}} \left[ \frac{d\breve{\mathbb{Q}}_{\boldsymbol{\Psi} \mid \boldsymbol{\theta}, \phi}}{d\mathbb{P}_{\boldsymbol{\Psi}, *}}(\boldsymbol{\Psi}) \right] = 1$. These two conditions guarantee that $\breve{\mathbb{Q}}_{\boldsymbol{\Psi} \mid \boldsymbol{\theta}, \phi}$ is indeed a probability measure on the space of marked point-process paths.

We decompose the log-likelihood as follows:

$$\log p(\mathcal{D} \mid \mathbf{X}, g(\cdot; \boldsymbol{\theta}), \phi) =$$
$$D_{\text{KL}} \Big( \breve{\mathbb{Q}}_{\boldsymbol{\omega}, \boldsymbol{\Psi} \mid \boldsymbol{\theta}, \phi}(\boldsymbol{\omega}, \boldsymbol{\Psi} \mid \boldsymbol{\theta}, \phi) \,\|\, \mathbb{P}(\boldsymbol{\omega}, \boldsymbol{\Psi} \mid \mathcal{D}, \mathbf{X}, g(\cdot; \boldsymbol{\theta}), \phi) \Big) + \breve{\mathcal{L}}_{\text{ELBO}}, \quad \text{(A11)}$$

where the ELBO is given by:

$$\breve{\mathcal{L}}_{\text{ELBO}} := \mathbb{E}_{\boldsymbol{\omega} \sim \breve{q}_{\boldsymbol{\omega} \mid \boldsymbol{\theta}, \phi}, \boldsymbol{\Psi} \sim \breve{\mathbb{Q}}_{\boldsymbol{\Psi} \mid \boldsymbol{\theta}, \phi}} \left[ \log \frac{p(\mathcal{D} \mid \mathbf{X}, g(\cdot; \boldsymbol{\theta}), \phi, \boldsymbol{\omega}, \boldsymbol{\Psi}) \, p_{\boldsymbol{\omega}}(\boldsymbol{\omega}) \, \frac{d\mathbb{P}_{\boldsymbol{\Psi} \mid \phi}}{d\mathbb{P}_{\boldsymbol{\Psi}, *}}(\boldsymbol{\Psi})}{\breve{q}_{\boldsymbol{\omega} \mid \boldsymbol{\theta}, \phi}(\boldsymbol{\omega} \mid \boldsymbol{\theta}, \phi) \, \frac{d\breve{\mathbb{Q}}_{\boldsymbol{\Psi} \mid \boldsymbol{\theta}, \phi}}{d\mathbb{P}_{\boldsymbol{\Psi}, *}}(\boldsymbol{\Psi} \mid \boldsymbol{\theta}, \phi)} \right], \quad \text{(A12)}$$

and where $\frac{d\mathbb{P}_{\boldsymbol{\Psi} \mid \phi}}{d\mathbb{P}_{\boldsymbol{\Psi}, *}}$ is the Radon-Nykodim derivative of the true conditional law $\mathbb{P}_{\boldsymbol{\Psi} \mid \phi}$ with respect to $\mathbb{P}_{\boldsymbol{\Psi}, *}$, cf. (A10).

**Minimizing the KL Divergence.**     When the variational distribution $\breve{\mathbb{Q}}(\boldsymbol{\omega}, \boldsymbol{\Psi} \mid \boldsymbol{\theta}, \phi)$ matches the true posterior $\mathbb{P}(\boldsymbol{\omega}, \boldsymbol{\Psi} \mid \mathcal{D}, \mathbf{X}, g(\cdot; \boldsymbol{\theta}), \phi)$, the KL divergence term in (A11) vanishes. Consequently, the ELBO becomes equal to the marginal log-likelihood, and maximizing the ELBO is equivalent to maximizing log-likelihood directly. In practice, we minimize the KL divergence so that our ELBO provides the closest possible lower bound to the true log-likelihood. Therefore, in order to obtain the closest lower bound to to the log-likelihood $\log p(\mathcal{D} \mid \mathbf{X}, g(\cdot; \boldsymbol{\theta}), \phi)$ we must find the distribution $\breve{\mathbb{Q}}(\boldsymbol{\omega}, \boldsymbol{\Psi} \mid \boldsymbol{\theta}, \phi)$ which minimizes the KL divergence in (A11).

Using standard mean-field variational inference techniques (see, e.g., Chapter 10.1 of [5]), the optimal distribution for the latent variables $\boldsymbol{\omega}$ given $(\boldsymbol{\theta}^{(\ell)}, \phi^{(\ell)})$ is obtained by computing the expectation of the joint log-density with respect to the other variational factors, that is

$$\log \breve{q}_{\boldsymbol{\omega} \mid \boldsymbol{\theta}, \phi}(\boldsymbol{\omega}) =$$
$$\mathbb{E}_{\boldsymbol{\Psi} \sim \breve{\mathbb{Q}}_{\boldsymbol{\Psi} \mid \boldsymbol{\theta}^{(\ell)}, \phi^{(\ell)}}} \left[ \log p(\mathcal{D} \mid \mathbf{X}, g(\cdot; \boldsymbol{\theta}^{(\ell)}), \phi^{(\ell)}, \boldsymbol{\omega}, \boldsymbol{\Psi}) + \log p_{\boldsymbol{\omega}}(\boldsymbol{\omega}) + \log \frac{d\mathbb{P}_{\boldsymbol{\Psi} \mid \phi^{(\ell)}}}{d\mathbb{P}_{\boldsymbol{\Psi}, *}}(\boldsymbol{\Psi}) \right] + \text{const.}$$

A similar update applies for $\boldsymbol{\Psi}$ given $(\boldsymbol{\theta}^{(\ell)}, \phi^{(\ell)})$,

$$\log \frac{d\breve{\mathbb{Q}}_{\boldsymbol{\Psi}|\boldsymbol{\theta},\phi}}{d\mathbb{P}_{\boldsymbol{\Psi},*}}(\boldsymbol{\Psi}) =$$

$$\mathbb{E}_{\boldsymbol{\omega}\sim\breve{q}_{\boldsymbol{\omega}|\boldsymbol{\theta},\phi}}\left[\log p(\mathcal{D} \mid \mathbf{X}, g(\cdot;\boldsymbol{\theta}^{(\ell)}), \phi^{(\ell)}, \boldsymbol{\omega}, \boldsymbol{\Psi}) + \log p_{\boldsymbol{\omega}}(\boldsymbol{\omega}) + \log \frac{d\mathbb{P}_{\boldsymbol{\Psi}|\phi^{(\ell)}}}{d\mathbb{P}_{\boldsymbol{\Psi},*}}(\boldsymbol{\Psi})\right] + \text{const.}$$

Following the same derivation as in Appendix G.1, we find the optimal variational distribution of $\boldsymbol{\omega}$ given $(\boldsymbol{\theta}, \phi)^{(\ell)}$:

$$\breve{q}_{\boldsymbol{\omega}|\boldsymbol{\theta}^{(\ell)},\phi^{(\ell)}}(\boldsymbol{\omega} \mid \boldsymbol{\theta}^{(\ell)}, \phi^{(\ell)}) = \prod_{i=1}^{N} \breve{q}_{\omega_i|\boldsymbol{\theta}^{(\ell)},\phi^{(\ell)}}(\omega_i \mid \boldsymbol{\theta}^{(\ell)}, \phi^{(\ell)}) = \prod_{i=1}^{N} p_{\text{PG}}\left(\omega_i \mid 1, \breve{c}_i^{(\ell)}\right), \quad (A13)$$

where

$$\breve{c}_i^{(\ell)} = \delta_i \, |g(y_i, \mathbf{x}_i; \boldsymbol{\theta}^{(\ell)})|. \tag{A14}$$

By mirroring the derivation in Appendix G.2, one shows that the optimal measure $\breve{\mathbb{Q}}_{\boldsymbol{\Psi}|\boldsymbol{\theta},\phi}(\boldsymbol{\Psi} \mid \boldsymbol{\theta}, \phi)$ is exactly the law under which each $\Psi_i$, for $i = 1, \ldots, N$, is a marked Poisson process on $[0, y_i] \times \mathbb{R}_+$ with intensity

$$\lambda_i^{\breve{\mathbb{Q}}}(t, \omega \mid \boldsymbol{\theta}^{(\ell)}, \phi^{(\ell)}) = \lambda_i^{\breve{\mathbb{Q}}}(t \mid \boldsymbol{\theta}^{(\ell)}, \phi^{(\ell)}) p_{\text{PG}}\left(\omega \mid 1, |g(t, \mathbf{x}_i; \boldsymbol{\theta}^{(\ell)})|\right),$$

where we set

$$\lambda_i^{\breve{\mathbb{Q}}}(t \mid \boldsymbol{\theta}^{(\ell)}, \phi^{(\ell)}) := \frac{t^{\rho-1}}{Z(t, \mathbf{x})} \phi^{(\ell)} \sigma(|g(t, \mathbf{x}_i; \boldsymbol{\theta}^{(\ell)})|) \exp\left(-\frac{g(t, \mathbf{x}_i; \boldsymbol{\theta}^{(\ell)}) + |g(t, \mathbf{x}_i; \boldsymbol{\theta}^{(\ell)})|}{2}\right). \tag{A15}$$

### F.2 EM Algorithm for MAP Estimation

**EM Algorithm.** The Expectation-Maximization (EM) algorithm provides an efficient framework to iteratively maximize the Q-function. At each iteration $\ell = 0, 1, 2, \ldots$, we perform the following three steps:

1. *Latent Variables Update.* Given the current estimates $(\boldsymbol{\theta}, \phi)^{(\ell)}$, update $\breve{q}_{\boldsymbol{\omega}|\boldsymbol{\theta}^{(\ell)},\phi^{(\ell)}}(\boldsymbol{\omega} \mid \boldsymbol{\theta}^{(\ell)}, \phi^{(\ell)})$ and $\breve{\mathbb{Q}}_{\boldsymbol{\Psi}|\boldsymbol{\theta}^{(\ell)},\phi^{(\ell)}}(\boldsymbol{\Psi} \mid \boldsymbol{\theta}^{(\ell)}, \phi^{(\ell)})$ according to (A13) and (A15), respectively.

2. *E-Step.* Given current estimates $(\boldsymbol{\theta}, \phi)^{(\ell)}$, compute the Q-function:

$$Q((\boldsymbol{\theta}, \phi)|(\boldsymbol{\theta}, \phi)^{(\ell)}) =$$

$$\mathbb{E}_{\boldsymbol{\omega}\sim\breve{q}_{\boldsymbol{\omega}|\boldsymbol{\theta}^{(\ell)},\phi^{(\ell)}}, \boldsymbol{\Psi}\sim\breve{\mathbb{Q}}_{\boldsymbol{\Psi}|\boldsymbol{\theta}^{(\ell)},\phi^{(\ell)}}}\left[\log\left(p(\mathcal{D} \mid \mathbf{X}, g(\cdot;\boldsymbol{\theta}), \phi, \boldsymbol{\omega}, \boldsymbol{\Psi}) \, p_{\boldsymbol{\omega}}(\boldsymbol{\omega}) \frac{d\mathbb{P}_{\boldsymbol{\Psi}|\phi}}{d\mathbb{P}_{\boldsymbol{\Psi},*}}(\boldsymbol{\Psi})\right)\right]$$
$$+ \log p_{\boldsymbol{\theta}}(\boldsymbol{\theta}) + \log p_{\phi}(\phi). \quad (A16)$$

Note that the entropy term of the ELBO (i.e., the denominator) is not included as it does not depend on the parameters $(\boldsymbol{\theta}, \phi)$ but on the current estimates $(\boldsymbol{\theta}, \phi)^{(\ell)}$, hence it is irrelevant to the parameters' optimization.

3. *M-Step.* Update the parameters by maximizing the Q-function:

$$(\boldsymbol{\theta}, \phi)^{(\ell+1)} = \arg\max_{\boldsymbol{\theta},\phi} Q((\boldsymbol{\theta}, \phi)|(\boldsymbol{\theta}, \phi)^{(\ell)}).$$

Steps 1-3 are repeated until a given convergence criterion is met. We provide an algorithmic description of our EM algorithm in Algorithm 1.

**Algorithm 1** Expectation-Maximization (EM) for maximum a posteriori (MAP) Estimation

---

1: Initialize: Set initial value for $(\boldsymbol{\theta}^{(\ell)},\ \phi^{(\ell)})$.
2: **Set:** iteration counter $\ell \leftarrow 0$
3: **repeat**
4:    $\ell \leftarrow \ell + 1$
5:    **Latent Variables Update:**
6:    **Update** $\breve{q}_{\boldsymbol{\omega}}^{(\ell)}$:
7:       Update: $\left\{ \breve{c}_i^{(\ell)} \right\}_{i=1}^{N}$ given $\boldsymbol{\theta}^{(\ell)}$ following (A14).
8:    **Update** $\breve{\mathbb{Q}}_{\boldsymbol{\Psi}}^{(\ell)}$:
9:       Update: $\left\{ \lambda_i^{\breve{\mathbb{Q}},(\ell)}(\cdot) \right\}_{i=1}^{N}$ given $\boldsymbol{\theta}^{(\ell)}$ and $\phi^{(\ell)}$ following (A22).
10:   **E-step:** Evaluate the Q-function $Q\big((\boldsymbol{\theta},\phi) \mid (\boldsymbol{\theta},\phi)^{(\ell)}\big)$ given $\left\{ \breve{c}_i^{(\ell)}, \lambda_i^{\breve{\mathbb{Q}},(\ell)}(\cdot) \right\}_{i=1}^{N}$, $\boldsymbol{\theta}^{\ell}$ and $\phi^{\ell}$ following (A16)
11:   **M-step:** Update parameters by

$$(\boldsymbol{\theta},\phi)^{(\ell+1)} = \arg\max_{\boldsymbol{\theta},\,\phi} Q\big((\boldsymbol{\theta},\phi) \mid (\boldsymbol{\theta},\phi)^{(\ell)}\big)$$

12: **until** Convergence criterion is met
13: **return** $(\boldsymbol{\theta}^{(\ell)},\ \phi^{(\ell)})$

---

**Computing the Q-function.** The optimal distributions which minimize the KL divergence can now be plugged in the ELBO of (A12) to obtain the closest lower bound to the log-likelihood. We now recast the MAP optimization problem in term of this lower bound. Specifically, define the following Q-function

$$Q((\boldsymbol{\theta}, \phi)|(\boldsymbol{\theta}, \phi)^{(\ell)}) = \mathbb{E}_{\boldsymbol{\omega} \sim \breve{q}_{\boldsymbol{\omega}|\boldsymbol{\theta}^{(\ell)}, \phi^{(\ell)}}, \boldsymbol{\Psi} \sim \breve{\mathbb{Q}}_{\boldsymbol{\Psi}|\boldsymbol{\theta}^{(\ell)}, \phi^{(\ell)}}} \left[\log p(\mathcal{D} \mid \mathbf{X}, g(\cdot; \boldsymbol{\theta}), \phi, \boldsymbol{\omega}, \boldsymbol{\Psi})\right]$$

$$+ \mathbb{E}_{\boldsymbol{\omega} \sim \breve{q}_{\boldsymbol{\omega}|\boldsymbol{\theta}^{(\ell)}, \phi^{(\ell)}}, \boldsymbol{\Psi} \sim \breve{\mathbb{Q}}_{\boldsymbol{\Psi}|\boldsymbol{\theta}^{(\ell)}, \phi^{(\ell)}}} \left[\log(p_{\boldsymbol{\omega}}(\boldsymbol{\omega}) + \log \left(\frac{\mathrm{d}\mathbb{P}_{\boldsymbol{\Psi}|\phi}}{\mathrm{d}\mathbb{P}_{\boldsymbol{\Psi}, *}}(\boldsymbol{\Psi})\right)\right]$$

$$+ \log p_{\boldsymbol{\theta}}(\boldsymbol{\theta}) + \log p_{\phi}(\phi).$$

We now wish to derive a closed-form expression for the Q-function which can be used in the MAP optimization. Specifically, using the augmented likelihood factorization in (15), we obtain

$$Q((\boldsymbol{\theta}, \phi)|(\boldsymbol{\theta}, \phi)^{(\ell)}) = \sum_{i=1}^{N} \mathbb{E}_{\omega_i \sim \breve{q}_{\omega_i|\boldsymbol{\theta}^{(\ell)}, \phi^{(\ell)}}, \Psi_i \sim \breve{\mathbb{Q}}_{\Psi_i|\boldsymbol{\theta}^{(\ell)}, \phi^{(\ell)}}} \left[\log p(\mathcal{D} \mid \mathbf{X}, g(\cdot; \boldsymbol{\theta}), \phi, \omega_i, \Psi_i)\right]$$

$$+ \mathbb{E}_{\boldsymbol{\omega} \sim \breve{q}_{\boldsymbol{\omega}|\boldsymbol{\theta}^{(\ell)}, \phi^{(\ell)}}} \left[\log p_{\boldsymbol{\omega}}(\boldsymbol{\omega})\right] + \mathbb{E}_{\boldsymbol{\Psi} \sim \breve{\mathbb{Q}}_{\boldsymbol{\Psi}|\boldsymbol{\theta}^{(\ell)}, \phi^{(\ell)}}} \left[\log \frac{\mathrm{d}\mathbb{P}_{\boldsymbol{\Psi}|\phi}}{\mathrm{d}\mathbb{P}_{\boldsymbol{\Psi}, *}}(\boldsymbol{\Psi})\right] + \log p_{\boldsymbol{\theta}}(\boldsymbol{\theta}) + \log p_{\phi}(\phi) + \text{const.}$$

Next, by substituting the expression for the augmented likelihood in (14), for the priors $p_{\boldsymbol{\theta}}(\boldsymbol{\theta})$ in (5) and $p_{\phi}(\phi)$ in (2) and for the Radon–Nikodym derivative of $\mathbb{P}_{\boldsymbol{\Psi}|\phi}$ with respect to $\mathbb{P}_{\boldsymbol{\Psi}, *}$ from (A10), we obtain

$$Q((\boldsymbol{\theta}, \phi)|(\boldsymbol{\theta}, \phi)^{(\ell)}) = \sum_{i=1}^{N} \left(\delta_i \left(\log \phi + \frac{g(y_i, \mathbf{x}_i; \boldsymbol{\theta})}{2} - \frac{g(y_i, \mathbf{x}_i; \boldsymbol{\theta})^2}{2} \mathbb{E}_{\omega_i \sim \breve{q}_{\omega_i|\boldsymbol{\theta}^{(\ell)}, \phi^{(\ell)}}} [\omega_i]\right)\right.$$

$$+ \mathbb{E}_{\Psi_i \sim \breve{\mathbb{Q}}_{\Psi_i|\boldsymbol{\theta}^{(\ell)}, \phi^{(\ell)}}} \left[\sum_{(t, \omega)_j \in \Psi_i} f(\omega_j, -g(t_j, \mathbf{x}_i; \boldsymbol{\theta}))\right] - \int_0^{y_i} \lambda_0(y_i, \mathbf{x}_i; \phi)\mathrm{d}t$$

$$+ \mathbb{E}_{\Psi_i \sim \breve{\mathbb{Q}}_{\Psi_i|\boldsymbol{\theta}^{(\ell)}, \phi^{(\ell)}}} \left[\sum_{(t, \omega)_j \in \Psi_i} \log \left(\frac{\phi}{Z(t_j, \mathbf{x}_i)}\right)\right]\right)$$

$$- \frac{1}{2}\boldsymbol{\theta}^\top \boldsymbol{\theta} + \log(\phi)(\alpha_0 - 1) - \phi\beta_0 + \text{const.}$$

We apply Campbell's theorem (see Theorem C.3), we substitute the expression for the baseline hazard $\lambda_i(\cdot; \phi)$ from (12) and we substitute the expectation using the optimal variational distribution of $\omega_i$ from (A13), to obtain

$$Q((\boldsymbol{\theta}, \phi)|(\boldsymbol{\theta}, \phi)^{(\ell)}) = \sum_{i=1}^{N} \left[\delta_i \left(\frac{g(y_i, \mathbf{x}_i; \boldsymbol{\theta})}{2} - \frac{g(y_i, \mathbf{x}_i; \boldsymbol{\theta})^2}{4\breve{c}_i^{(\ell)}} \tanh \left(\frac{\breve{c}_i^{(\ell)}}{2}\right)\right)\right.$$

$$- \frac{1}{2} \int_0^{y_i} g(t, \mathbf{x}_i; \boldsymbol{\theta}) \lambda_i^{\breve{\mathbb{Q}}}(t \mid \boldsymbol{\theta}^{(\ell)}, \phi^{(\ell)})\mathrm{d}t$$

$$- \frac{1}{4} \int_0^{y_i} \frac{g(t, \mathbf{x}_i; \boldsymbol{\theta})^2}{|g(t, \mathbf{x}_i; \boldsymbol{\theta}^{(\ell)})|} \tanh \left(\frac{|g(t, \mathbf{x}_i; \boldsymbol{\theta}^{(\ell)})|}{2}\right) \lambda_i^{\breve{\mathbb{Q}}}(t \mid \boldsymbol{\theta}^{(\ell)}, \phi^{(\ell)})\mathrm{d}t\right]$$

$$- \frac{1}{2}\boldsymbol{\theta}^\top \boldsymbol{\theta} + \log(\phi) \left(\alpha_0 + \sum_{i=1}^{N} \left(\delta_i + \int_0^{y_i} \lambda_i^{\breve{\mathbb{Q}}}(t \mid \boldsymbol{\theta}^{(\ell)}, \phi^{(\ell)})\mathrm{d}t\right) - 1\right)$$

$$- \phi \left(\beta_0 + \sum_{i=1}^{N} \int_0^{y_i} \frac{t^{\rho-1}}{Z(t, \mathbf{x}_i)}\mathrm{d}t\right) + \text{const.,}$$

where $\lambda_i^{\breve{\mathbb{Q}}}(t \mid \boldsymbol{\theta}^{(\ell)}, \phi^{(\ell)})$ is shown in (A15).

# G   Coordinate Ascent Variational Inference Optimal Updates

In this Appendix we present a heuristic derivation of the CAVI optimal updates presented in Section 4.3. Before presenting the next results, we define here for convenience

$$\tilde{m}_i^{(k)}(t) := \mathbb{E}_{\boldsymbol{\theta} \sim q_{\boldsymbol{\theta}}^{(k)}} \left[ g^{\mathrm{lin}}(t, \mathbf{x}_i; \boldsymbol{\theta}) \right], \quad \tilde{s}_i^{(k)}(t) := \sqrt{\mathbb{E}_{\boldsymbol{\theta} \sim q_{\boldsymbol{\theta}}^{(k)}} \left[ g^{\mathrm{lin}}(t, \mathbf{x}_i; \boldsymbol{\theta})^2 \right]}$$

for $k \geq 0$.

## G.1   Optimal Update for $\omega$

Using standard mean-field variational inference techniques (see, e.g., Chapter 10.1 of [5]), the optimal update for the latent variables $\omega$ is obtained by computing the expectation of the joint log-density with respect to the other variational factors. In particular, we have

$$\log q_{\boldsymbol{\omega}}^{(k)}(\boldsymbol{\omega}) = \mathbb{E}_{\phi \sim q_\phi^{(k-1)}, \boldsymbol{\theta} \sim q_{\boldsymbol{\theta}}^{(k-1)}, \boldsymbol{\Psi} \sim \mathbb{Q}_{\boldsymbol{\Psi}}^{(k-1)}} \left[ \log p \left( \mathcal{D} \mid \phi, g^{\mathrm{lin}}(\cdot; \boldsymbol{\theta}), \boldsymbol{\omega}, \boldsymbol{\Psi} \right) \right] + \log p_{\boldsymbol{\omega}}(\boldsymbol{\omega}) + \mathrm{const.}$$

Using the augmented likelihood factorization in (15), the expression decomposes as

$$\log q_{\boldsymbol{\omega}}^{(k)}(\boldsymbol{\omega}) = \sum_{i=1}^{N} \mathbb{E}_{\phi \sim q_\phi^{(k-1)}, \boldsymbol{\theta} \sim q_{\boldsymbol{\theta}}^{(k-1)}, \Psi_i \sim \mathbb{Q}_{\Psi_i}^{(k-1)}} \left[ \log p \left( y_i, \delta_i \mid \mathbf{x}_i, \phi, g^{\mathrm{lin}}(\cdot; \boldsymbol{\theta}), \omega_i, \Psi_i \right) \right]$$

$$+ \log p_{\boldsymbol{\omega}}(\boldsymbol{\omega}) + \mathrm{const.}$$

Next, by substituting the expression for the prior $p_{\boldsymbol{\omega}}(\boldsymbol{\omega})$ from (10) and the augmented likelihood from (14), we obtain

$$\log q_{\boldsymbol{\omega}}^{(k)}(\boldsymbol{\omega}) = \sum_{i=1}^{N} \left( -\frac{\omega_i \delta_i}{2} \left( \tilde{s}_i^{(k-1)}(y_i) \right)^2 + \log p_{\mathrm{PG}}(\omega_i | 1, 0) \right) + \mathrm{const.}$$

Finally, by applying the identity in (A2), we deduce that the optimal variational distribution factorizes as

$$q_{\boldsymbol{\omega}}^{(k)}(\boldsymbol{\omega}) = \prod_{i=1}^{N} q_{\omega_i}^{(k)}(\omega_i) = \prod_{i=1}^{N} p_{\mathrm{PG}} \left( \omega_i \mid 1, \tilde{c}_i^{(k)} \right),$$

where

$$\tilde{c}_i^{(k)} = \delta_i \, \tilde{s}_i^{(k-1)}(y_i) \tag{A17}$$

**Optimal Variational Expectations for $\omega$.** From Proposition B.4, we obtain the required expectation for updating the other variational factors with

$$\mathbb{E}_{\omega_i \sim q_{\omega_i}^{(k)}}[\omega_i] = \frac{1}{2\tilde{c}_i^{(k)}} \tanh \left( \frac{\tilde{c}_i^{(k)}}{2} \right) \tag{A18}$$

for $i = 1, \dots, N$. Notably, since this expectation is always multiplied by $\delta_i$ when updating other variational factors, it remains well-defined in all cases.

## G.2   Optimal Update for $\Psi$

Using standard mean-field variational inference techniques (see, e.g., Chapter 10.1 of [5]), we obtain the optimal Radon-Nykodim derivative $\frac{\mathrm{d}\mathbb{Q}_{\boldsymbol{\Psi}}}{\mathrm{d}\mathbb{P}_{\boldsymbol{\Psi},*}}$ by taking the expectation of the joint log-density with respect to the other variational factors. In particular, we have

$$\log \frac{\mathrm{d}\mathbb{Q}_{\boldsymbol{\Psi}}^{(k)}}{\mathrm{d}\mathbb{P}_{\boldsymbol{\Psi},*}}(\boldsymbol{\Psi}) = \mathbb{E}_{\phi \sim q_\phi^{(k-1)}, \boldsymbol{\theta} \sim q_{\boldsymbol{\theta}}^{(k-1)}, \boldsymbol{\omega} \sim q_{\boldsymbol{\omega}}^{(k)}} \left[ \log p(\mathcal{D} \mid \phi, g^{\mathrm{lin}}(\cdot; \boldsymbol{\theta}), \boldsymbol{\omega}, \boldsymbol{\Psi}) \right]$$

$$+ \mathbb{E}_{\phi \sim q_\phi^{(k-1)}} \left[ \log \frac{\mathrm{d}\mathbb{P}_{\boldsymbol{\Psi}|\phi}}{\mathrm{d}\mathbb{P}_{\boldsymbol{\Psi},*}}(\boldsymbol{\Psi}) \right] + \mathrm{const.}, \tag{A19}$$

where the constant term absorbs all terms irrelevant to the optimisation. Using the augmented likelihood factorization in (15), the expression in (A19) decomposes as

$$
\log \frac{\mathrm{d}\mathbb{Q}_{\boldsymbol{\Psi}}^{(k)}}{\mathrm{d}\mathbb{P}_{\boldsymbol{\Psi},*}}(\boldsymbol{\Psi}) = \sum_{i=1}^{N} \mathbb{E}_{\phi\sim q_\phi^{(k-1)},\boldsymbol{\theta}\sim q_{\boldsymbol{\theta}}^{(k-1)},\omega_i\sim q_{\omega_i}^{(k)}} \left[\log p(y_i,\delta_i|\phi,g^{\mathrm{lin}}(\cdot;\boldsymbol{\theta}),\omega_i,\Psi_i)\right]
$$

$$
+ \mathbb{E}_{\phi\sim q_\phi^{(k-1)}} \left[\log \frac{\mathrm{d}\mathbb{P}_{\boldsymbol{\Psi}|\phi}}{\mathrm{d}\mathbb{P}_{\boldsymbol{\Psi},*}}(\boldsymbol{\Psi})\right] + \mathrm{const.}
$$

Next, by substituting the augmented likelihood from (14) and the Radon–Nikodym derivative of $\mathbb{P}_{\boldsymbol{\Psi}|\phi}$ with respect to $\mathbb{P}_{\boldsymbol{\Psi},*}$ from (A10), we arrive at the unnormalised form

$$
\log \frac{\mathrm{d}\mathbb{Q}_{\boldsymbol{\Psi}}^{(k)}}{\mathrm{d}\mathbb{P}_{\boldsymbol{\Psi},*}}(\boldsymbol{\Psi}) = \sum_{i=1}^{N} \sum_{(t,\omega)_j\in\Psi_i} \mathbb{E}_{\boldsymbol{\theta}\sim q_{\boldsymbol{\theta}}^{(k-1)}} \left[f(\omega_j,-g^{\mathrm{lin}}(t_j,\mathbf{x}_i;\boldsymbol{\theta}))\right]
$$

$$
+ \sum_{i=1}^{N} \sum_{(t,\omega)_j\in\Psi_i} \mathbb{E}_{\phi\sim q_\phi^{(k-1)}} \left[\log\left(\frac{\phi}{Z(t_j,\mathbf{x}_i)}\right)\right] + \mathrm{const.} \quad (A20)
$$

Plugging in the definition of $f(\cdot,\cdot)$ from (8) simplifies (A20) to

$$
\log \frac{\mathrm{d}\mathbb{Q}_{\boldsymbol{\Psi}}^{(k)}}{\mathrm{d}\mathbb{P}_{\boldsymbol{\Psi},*}}(\boldsymbol{\Psi}) = -\sum_{i=1}^{N} \sum_{(t,\omega)_j\in\Psi_i} \left[\frac{\tilde{m}_i^{(k)}(t_j)}{2} + \frac{(\tilde{s}_i^{(k)}(t_j))^2}{2}\omega_j + \log(2)\right]
$$

$$
+ \sum_{i=1}^{N} \sum_{(t,\omega)_j\in\Psi_i} \left[\mathbb{E}_{\phi\sim q_\phi^{(k-1)}}[\log\phi] - \log Z(t_j,\mathbf{x}_i)\right] + \mathrm{const.}
$$

To express this in closed form, define for each $i=1,\ldots,N$ and $(t,\omega)\in[0,y_i]\times\mathbb{R}_+$ the functions

$$
h_i^{\mathbb{Q},(k)}(t,\omega) := \exp\left(-\frac{\left(\tilde{s}_i^{(k-1)}(t)\right)^2}{2}\omega\right)\cosh\left(\frac{\tilde{s}_i^{(k-1)}(t)}{2}\right),
$$

$$
\gamma_i^{\mathbb{Q},(k)}(t) := \frac{1}{Z(t,\mathbf{x}_i)}\sigma(\tilde{s}_i^{(k-1)}(t))\exp\left(-\frac{\tilde{m}_i^{(k-1)}(t)+\tilde{s}_i^{(k-1)}(t)}{2} + \mathbb{E}_{\phi\sim q_\phi^{(k-1)}}[\log\phi]\right),
$$

$$
\lambda_i^{\mathbb{Q},(k)}(t,\omega) := \gamma_i^{\mathbb{Q},(k)}(t)h_i^{\mathbb{Q},(k)}(t,\omega)\lambda_{*,i}(t,\omega),
$$

where $\lambda_{*,i}(t,\omega)$ is the intensity defined in (A8). Furthermore, we define for convenience,

$$
\lambda_i^{\mathbb{Q},(k)}(t) := t^{\rho-1}\gamma_i^{\mathbb{Q},(k)}(t). \quad (A21)
$$

Notice that by using expression (A2), the function $\lambda_i^{\mathbb{Q},(k)}(t,\omega)$ can be written as

$$
\lambda_i^{\mathbb{Q},(k)}(t,\omega) = \lambda_i^{\mathbb{Q},(k)}(t)\, p_{\mathrm{PG}}\left(\omega\mid 1,\tilde{s}_i^{(k-1)}(t)\right). \quad (A22)
$$

Finally, enforcing the normalisation condition

$$
\mathbb{E}_{\boldsymbol{\Psi}\sim\mathbb{P}_{\boldsymbol{\Psi},*}}\left[\frac{\mathrm{d}\mathbb{Q}_{\boldsymbol{\Psi}}^{(k)}}{\mathrm{d}\mathbb{P}_{\boldsymbol{\Psi},*}}(\boldsymbol{\Psi})\right] = 1
$$

together with Campbell's theorem (Theorem C.3) yields the normalized derivative

$$
\frac{\mathrm{d}\mathbb{Q}_{\boldsymbol{\Psi}}^{(k)}}{\mathrm{d}\mathbb{P}_{\boldsymbol{\Psi},*}}(\boldsymbol{\Psi}) =
$$

$$
\prod_{i=1}^{N}\left(\prod_{(t,\omega)_j\in\Psi_i}\gamma_i^{\mathbb{Q},(k)}(t_j)h_i^{\mathbb{Q},(k)}(t_j,\omega_j)\right)\exp\left(\int_0^{y_i}\int_0^\infty\left(\lambda_{*,i}(t,\omega)-\lambda_i^{\mathbb{Q},(k)}(t,\omega)\right)\mathrm{d}\omega\mathrm{d}t\right).
$$

Notice that the products $\gamma_i^{\mathbb{Q},(k)}(t_j)h_i^{\mathbb{Q},(k)}(t_j,\omega_j)$ are all strictly positive [2], hence $\frac{\mathrm{d}\mathbb{Q}_{\boldsymbol{\Psi}}^{(k)}}{\mathrm{d}\mathbb{P}_{\boldsymbol{\Psi},*}}$ is also strictly positive. Under suitable regularity conditions on $g$, one can show that $h_i^{\mathbb{Q},(k)}(t,\omega)$ and $\gamma_i^{\mathbb{Q},(k)}(t)$ satisfy the integrability criteria of (A9), so that $\mathbb{Q}_{\boldsymbol{\Psi}}^{(k)}$ is the probability measure under which each $\Psi_i$ $(i=1,\ldots,N)$ is a marked Poisson Process on $[0,y_i]\times\mathbb{R}_+$ with intensity function $\lambda_i^{\mathbb{Q},(k)}(t,\omega)$.

**Optimal Variational Expectations for $\boldsymbol{\Psi}$.** From Proposition B.4, we obtain the required integrals for updating the other variational factors

$$\int_{\mathbb{R}_+}\lambda_i^{\mathbb{Q},(k)}(t,\omega)\mathrm{d}\omega = \lambda_i^{\mathbb{Q},(k)}(t),$$

$$\int_{\mathbb{R}_+}\lambda_i^{\mathbb{Q},(k)}(t,\omega)\omega\mathrm{d}\omega = \lambda_i^{\mathbb{Q},(k)}(t)\frac{1}{2\tilde{s}_i^{(k-1)}(t)}\tanh\left(\frac{\tilde{s}_i^{(k-1)}(t)}{2}\right).$$

### G.3 Optimal Update for $\phi$

Using standard mean-field variational inference techniques (see, e.g., Chapter 10.1 of [5]), the optimal variational factor for the parameter $\phi$ is obtained by computing the expectation of the joint log-density with respect to the other variational factors. In particular, we have

$$\log q_\phi^{(k)}(\phi) = \mathbb{E}_{\boldsymbol{\theta}\sim q_{\boldsymbol{\theta}}^{(k-1)},\boldsymbol{\omega}\sim q_{\boldsymbol{\omega}}^{(k)},\boldsymbol{\Psi}\sim\mathbb{Q}_{\boldsymbol{\Psi}}^{(k)}}\left[\log p\left(\mathcal{D}\mid\phi,g^{\mathrm{lin}}(\cdot;\boldsymbol{\theta}),\boldsymbol{\omega},\boldsymbol{\Psi}\right)+\log\frac{\mathrm{d}\mathbb{P}_{\boldsymbol{\Psi}\mid\phi}}{\mathrm{d}\mathbb{P}_{\boldsymbol{\Psi},*}}(\boldsymbol{\Psi})\right]$$
$$+\log p_\phi(\phi)+\mathrm{const.}$$

Using the augmented likelihood factorization in (15), the expression decomposes as

$$\log q_\phi^{(k)}(\phi) = \sum_{i=1}^N\mathbb{E}_{\boldsymbol{\theta}\sim q_{\boldsymbol{\theta}}^{(k-1)},\omega_i\sim q_{\omega_i}^{(k)},\Psi_i\sim\mathbb{Q}_{\Psi_i}^{(k)}}\left[\log p\left(y_i,\delta_i\mid\mathbf{x}_i,\phi,\boldsymbol{\theta},\omega_i,\Psi_i\right)\right]$$
$$+\mathbb{E}_{\boldsymbol{\Psi}\sim\mathbb{Q}_{\boldsymbol{\Psi}}^{(k)}}\left[\log\frac{\mathrm{d}\mathbb{P}_{\boldsymbol{\Psi}\mid\phi}}{\mathrm{d}\mathbb{P}_{\boldsymbol{\Psi},*}}(\boldsymbol{\Psi})\right]+\log p_\phi(\phi)+\mathrm{const.}$$

Next, by substituting the expression for the augmented likelihood from (14), the Radon–Nikodym derivative of $\mathbb{P}_{\boldsymbol{\Psi}\mid\phi}$ with respect to $\mathbb{P}_{\boldsymbol{\Psi},*}$ from (A10), and the prior of $\phi$ from (2), we obtain,

$$\log q_\phi^{(k)}(\phi) = \sum_{i=1}^N\left(\delta_i\log\lambda_0(y_i,\mathbf{x}_i;\phi)-\int_0^{y_i}\lambda_0(t,\mathbf{x}_i;\phi)\mathrm{d}t\right.$$
$$\left.+\mathbb{E}_{\Psi_i\sim\mathbb{Q}_{\Psi_i}^{(k)}}\left[\sum_{(t,\omega)_j\in\Psi_i}\log\left(\frac{\phi}{Z(t_j,\mathbf{x}_i)}\right)\right]\right)+(\alpha_0-1)\log(\phi)-\beta_0\phi+\mathrm{const.}$$

We apply Campbell's Theorem (Theorem C.3) and substitute the expression for the baseline hazard $\lambda_0(\cdot)$ from (2), to obtain

$$\log q_\phi^{(k)}(\phi)$$
$$=\log(\phi)\left(\alpha_0+\sum_{i=1}^N\left(\delta_i+\int_0^{y_i}\lambda_i^{\mathbb{Q},(k)}(t)\mathrm{d}t\right)-1\right)-\phi\left(\beta_0+\sum_{i=1}^N\int_0^{y_i}\frac{t^{\rho-1}}{Z(t,\mathbf{x}_i)}\mathrm{d}t\right)+\mathrm{const.},$$

where $\lambda_i^{\mathbb{Q},(k)}(t)$ is shown in (A21). We deduce that

$$q_\phi^{(k)}(\phi) = \mathrm{Gamma}(\tilde{\alpha}^{(k)},\tilde{\beta}),$$

where with shape $\tilde{\alpha}^{(k)}$ and rate $\tilde{\beta}$ given by

$$\tilde{\alpha}^{(k)} = \alpha_0+\sum_{i=1}^N\left(\delta_i+\int_0^{y_i}\lambda_i^{\mathbb{Q},(k)}(t)\mathrm{d}t\right),\quad \tilde{\beta}=\beta_0+\sum_{i=1}^N\int_0^{y_i}\frac{t^{\rho-1}}{Z(t,\mathbf{x}_i)}\mathrm{d}t. \tag{A23}$$

---

[2] See Lemma N.1 for a proof of the strict positivity of the normalization factor $Z(t,\mathbf{x}_i)$.

**Optimal Variational Expectation for $\phi$.** We obtain the required expectation for updating the other variational factors with

$$\mathbb{E}_{\phi \sim q_\phi^{(k)}}[\log \phi] = \psi\left(\tilde{\alpha}^{(k)}\right) - \log\left(\tilde{\beta}\right), \tag{A24}$$

where $\psi(\cdot)$ is the digamma function.

### G.4 Optimal Update for $\theta$

Using standard mean-field variational inference techniques (see, e.g., Chapter 10.1 of [5]), the optimal variational factor for the parameters $\theta$ is obtained by computing the expectation of the joint log-density with respect to the other variational factors. In particular, we have

$$\log q_\theta^{(k)}(\theta) = \mathbb{E}_{\phi \sim q_\phi^{(k)}, \omega \sim q_\omega^{(k)}, \Psi \sim \mathbb{Q}_\Psi^{(k)}}\left[\log p\left(\mathcal{D} \mid \phi, g^{\text{lin}}(\cdot; \theta), \omega, \Psi\right)\right] + \log p_\theta(\theta) + \text{const.}$$

Using the augmented likelihood factorization in (15), we obtain

$$\log q_\theta^{(k)}(\theta) = \sum_{i=1}^N \mathbb{E}_{\phi \sim q_\phi^{(k)}, \omega_i \sim q_{\omega_i}^{(k)}, \Psi_i \sim \mathbb{Q}_{\Psi_i}^{(k)}}\left[\log p\left(y_i, \delta_i \mid \mathbf{x}_i, \phi, g^{\text{lin}}(\cdot; \theta), \omega_i, \Psi_i\right)\right]$$
$$+ \log p_\theta(\theta) + \text{const.}$$

Next, by substituting the expression for the augmented likelihood (14) and for the prior for $\theta$ from (5), we obtain,

$$\log q_\theta^{(k)}(\theta) = \sum_{i=1}^N \left(\frac{\delta_i}{2}\left(g^{\text{lin}}(y_i, \mathbf{x}_i; \theta) - \mathbb{E}_{\omega_i \sim q_{\omega_i}^{(k)}}[\omega_i]\, g^{\text{lin}}(y_i, \mathbf{x}_i; \theta)^2\right)\right.$$
$$\left. + \mathbb{E}_{\Psi_i \sim \mathbb{Q}_{\Psi_i}^{(k)}}\left[\sum_{(t,\omega)_j \in \Psi_i} f\left(\omega_j, -g^{\text{lin}}(t_j, \mathbf{x}_i; \theta)\right)\right]\right) - \frac{1}{2}\theta^\top \theta + \text{const.}$$

We apply Campbell's Theorem (Theorem C.3) to obtain,

$$\log q_\theta^{(k)}(\theta) = \sum_{i=1}^N \left(\frac{\delta_i}{2}\left(g^{\text{lin}}(y_i, \mathbf{x}_i; \theta) - \mathbb{E}_{\omega_i \sim q_{\omega_i}^{(k)}}[\omega_i]g^{\text{lin}}(y_i, \mathbf{x}_i; \theta)^2\right)\right.$$
$$\left. + \frac{1}{2}\int_{\mathcal{Z}_i}\left(-g^{\text{lin}}(t, \mathbf{x}_i; \theta) - g^{\text{lin}}(t, \mathbf{x}_i; \theta)^2 \omega\right)\lambda_i^{\mathbb{Q},(k)}(t, \omega)\mathrm{d}t\mathrm{d}\omega\right) - \frac{1}{2}\theta^\top \theta + \text{const.},$$

where $\lambda_i^{\mathbb{Q},(k)}(t, \omega)$ is shown in Equation (A22). Next, we recall the expression for $g^{\text{lin}}(\cdot; \theta)$ from (18) and we notice that

$$g^{\text{lin}}(\cdot; \theta) = \theta^\top \mathbf{J}_{\theta_{\text{MAP}}}(\cdot) + \text{const.}$$
$$g^{\text{lin}}(\cdot; \theta)^2 = \theta^\top \mathbf{J}_{\theta_{\text{MAP}}}(\cdot)\left(2g(\cdot; \theta_{\text{MAP}}) - 2\mathbf{J}_{\theta_{\text{MAP}}}(\cdot)^\top \theta_{\text{MAP}}\right) + \theta^\top \mathbf{J}_{\theta_{\text{MAP}}}(\cdot)\mathbf{J}_{\theta_{\text{MAP}}}(\cdot)^\top \theta + \text{const.},$$

where the constant term represents terms that do not depend on $\theta$. We substitute the expression for $g^{\text{lin}}(\cdot; \theta)$ and $g^{\text{lin}}(\cdot; \theta)^2$ and we obtain,

$$\log q_\theta^{(k)}(\theta) = \theta^T \mathbf{A}^{(k)} - \theta^\top \mathbf{B}^{(k)}\theta + \text{const.},$$

where

$$\mathbf{A}^{(k)} = \sum_{i=1}^N \frac{1}{2}\left(\delta_i \mathbf{J}_{\theta_{\text{MAP}}}(y_i, \mathbf{x}_i)\left(1 - 2\mathbb{E}_{\omega_i \sim q_{\omega_i}^{(k)}}[\omega_i]\left(g(y_i, \mathbf{x}_i; \theta_{\text{MAP}}) - \mathbf{J}_{\theta_{\text{MAP}}}(y_i, \mathbf{x}_i)^\top \theta_{\text{MAP}}\right)\right)\right.$$
$$\left. - \left(\mathcal{I}_{1,i}^{(k)} + 2\left(\mathcal{I}_{2,i}^{(k)} - \mathcal{I}_{3,i}^{(k)}\theta_{\text{MAP}}\right)\right)\right) \tag{A25}$$

$$\mathbf{B}^{(k)} = \sum_{i=1}^N \frac{1}{2}\left(\delta_i\, \mathbb{E}_{\omega_i \sim q_{\omega_i}^{(k)}}[\omega_i]\mathbf{J}_{\theta_{\text{MAP}}}(y_i, \mathbf{x}_i)\mathbf{J}_{\theta_{\text{MAP}}}(y_i, \mathbf{x}_i)^\top + \mathcal{I}_{3,i}^{(k)}\right) + \frac{1}{2}\mathbf{I}_m \tag{A26}$$

and

$$\mathcal{I}_{1,i}^{(k)} = \int_0^{y_i} \mathbf{J}_{\boldsymbol{\theta}_{\mathrm{MAP}}}(t, \mathbf{x}_i) \lambda_i^{\mathbb{Q},(k)}(t) \mathrm{d}t$$

$$\mathcal{I}_{2,i}^{(k)} = \int_0^{y_i} \mathbf{J}_{\boldsymbol{\theta}_{\mathrm{MAP}}}(t, \mathbf{x}_i) g(t, \mathbf{x}_i; \boldsymbol{\theta}_{\mathrm{MAP}}) \lambda_i^{\mathbb{Q},(k)}(t) \frac{\tanh\left(\tilde{s}_i^{(k-1)}(t)/2\right)}{2\tilde{s}_i^{(k-1)}(t)} \mathrm{d}t$$

$$\mathcal{I}_{3,i}^{(k)} = \int_0^{y_i} \mathbf{J}_{\boldsymbol{\theta}_{\mathrm{MAP}}}(t, \mathbf{x}_i) \mathbf{J}_{\boldsymbol{\theta}_{\mathrm{MAP}}}(t, \mathbf{x}_i)^\top \lambda_i^{\mathbb{Q},(k)}(t) \frac{\tanh\left(\tilde{s}_i^{(k-1)}(t)/2\right)}{2\tilde{s}_i^{(k-1)}(t)} \mathrm{d}t.$$

$\mathbf{A}$, $\mathcal{I}_{1,i}$ and $\mathcal{I}_{2,i}$ are vectors of the same length of $\boldsymbol{\theta}$. $\mathbf{B}$ and $\mathcal{I}_{3,i}$ are square matrices for which each dimension is the length of $\boldsymbol{\theta}$, and $\mathbf{I}_m$ is the identity matrix of length of $\boldsymbol{\theta}$. We deduce that

$$q_{\boldsymbol{\theta}}^{(k)}(\boldsymbol{\theta}) = \mathcal{N}\left(\tilde{\boldsymbol{\mu}}^{(k)}, \tilde{\boldsymbol{\Sigma}}^{(k)}\right),$$

where

$$\tilde{\boldsymbol{\mu}}^{(k)} = \frac{1}{2}\left(\mathbf{B}^{(k)}\right)^{-1}\mathbf{A}^{(k)}, \quad \tilde{\boldsymbol{\Sigma}} = \frac{1}{2}\left(\mathbf{B}^{(k)}\right)^{-1}. \tag{A27}$$

**Optimal Variational Expectation for $\boldsymbol{\theta}$.** We obtain the required expectation for updating the other variational factors,

$$\mathbb{E}_{\boldsymbol{\theta} \sim q_{\boldsymbol{\theta}}^{(k)}}[g^{\mathrm{lin}}(t, \mathbf{x}_i; \boldsymbol{\theta})] = g(t, \mathbf{x}_i; \boldsymbol{\theta}_{\mathrm{MAP}}) + \mathbf{J}_{\boldsymbol{\theta}_{\mathrm{MAP}}}(t, \mathbf{x}_i)^\top \left(\tilde{\boldsymbol{\mu}}^{(k)} - \boldsymbol{\theta}_{\mathrm{MAP}}\right),$$

$$\mathbb{E}_{\boldsymbol{\theta} \sim q_{\boldsymbol{\theta}}^{(k)}}[g^{\mathrm{lin}}(t, \mathbf{x}_i; \boldsymbol{\theta})^2] = \left(g(t, \mathbf{x}_i; \boldsymbol{\theta}_{\mathrm{MAP}}) + \mathbf{J}_{\boldsymbol{\theta}_{\mathrm{MAP}}}(t, \mathbf{x}_i)^\top \left(\tilde{\boldsymbol{\mu}}^{(k)} - \boldsymbol{\theta}_{\mathrm{MAP}}\right)\right)^2 \tag{A28}$$

$$+ \mathbf{J}_{\boldsymbol{\theta}_{\mathrm{MAP}}}(t, \mathbf{x}_i)^\top \tilde{\boldsymbol{\Sigma}}^{(k)} \mathbf{J}_{\boldsymbol{\theta}_{\mathrm{MAP}}}(t, \mathbf{x}_i).$$

# H Coordinate Ascent Variational Inference Algorithm

---

**Algorithm 2** Coordinate Ascent Variational Inference (CAVI)

---

1: **Compute:** Compute $\tilde{\beta}$ following (A23).

2: **Initialize::** Set initial values for $\tilde{\alpha}^{(0)}$ and $\left(\tilde{\boldsymbol{\mu}}, \tilde{\boldsymbol{\Sigma}}\right)^{(0)}$.

3: **Compute:** $\mathbb{E}_{\phi \sim q_\phi^{(0)}}[\log \phi]$ given $\left(\tilde{\alpha}^{(0)}, \tilde{\beta}\right)$ following (A24).

4: **Compute:** $\left\{(\tilde{m}_i(\cdot), \tilde{s}_i(\cdot))^{(0)}\right\}_{i=1}^N$ given $\left(\tilde{\boldsymbol{\mu}}, \tilde{\boldsymbol{\Sigma}}\right)^{(0)}$ following (A28).

5: **Set:** iteration counter $k \leftarrow 0$

6: **repeat**

7:     $k \leftarrow k + 1$

8:     **Update** $q_{\boldsymbol{\omega}}^{(k)}$:

9:         Update: $\left\{\tilde{c}_i^{(k)}\right\}_{i=1}^N$ given $\left\{\tilde{s}_i(\cdot)^{(k-1)}\right\}_{i=1}^N$ following (A17).

10:         Compute: $\left\{\mathbb{E}_{\omega_i \sim q_{\omega_i}^{(k)}}[\omega_i]\right\}_{i=1}^N$ given $\left\{\tilde{c}_i^{(k)}\right\}_{i=1}^N$ following (A18).

11:     **Update** $\mathbb{Q}_{\boldsymbol{\Psi}}^{(k)}$:

12:         Update: $\left\{\lambda_i^{\mathbb{Q},(k)}(\cdot)\right\}_{i=1}^N$ given $\left(\{(\tilde{m}_i(\cdot), \tilde{s}_i(\cdot))^{(k-1)}\}_{i=1}^N, \mathbb{E}_{\phi \sim q_\phi^{(k-1)}}[\log \phi]\right)$ following (A22).

13:     **Update** $q_\phi^{(k)}$:

14:         Update: $\tilde{\alpha}^{(k)}$ given $\left\{\lambda_i^{\mathbb{Q},(k)}(\cdot)\right\}_{i=1}^N$ following (A23).

15:         Compute: $\mathbb{E}_{\phi \sim q_\phi^{(k)}}[\log \phi]$ given $(\tilde{\alpha}^{(k)}, \tilde{\beta})$ following (A24).

16:     **Update** $q_{\boldsymbol{\theta}}^{(k)}$:

17:         Update: $\left(\tilde{\boldsymbol{\mu}}, \tilde{\boldsymbol{\Sigma}}\right)^{(k)}$ given $\left\{(\mathbb{E}_{\omega_i \sim q_{\omega_i}^{(k)}}[\omega_i], \lambda_i^{\mathbb{Q},(k)}(\cdot)\right\}_{i=1}^N$ following (A27).

18:         Compute: $\left\{(\tilde{m}_i(\cdot), \tilde{s}_i(\cdot))^{(k)}\right\}_{i=1}^N$ given $\left(\tilde{\boldsymbol{\mu}}, \tilde{\boldsymbol{\Sigma}}\right)^{(k)}$ following (A28).

19: **until** Convergence criterion is met

20: **Return:** Optimized variational distributions $q_{\boldsymbol{\theta}}^{(k^\star)}(\boldsymbol{\theta}) = \mathcal{N}\left(\tilde{\boldsymbol{\mu}}^{(k^\star)}, \tilde{\boldsymbol{\Sigma}}^{(k^\star)}\right)$ and $q_\phi^{k^\star}(\phi) =$ Gamma$\left(\tilde{\alpha}^{(k^\star)}, \tilde{\beta}\right)$, where $k^\star$ is the final iteration after convergence.

---

# I Computational Speed-Ups

Survival-analysis cohorts often comprise only a few hundred to a few thousand observations, yet modern deep learning models may involve millions of parameters, putting us in the $N \ll m$ regime. To exploit this disparity, we develop two complementary strategies that avoid any expensive $m$-dimensional inversions or factorizations by leveraging the fact that the nontrivial part of our key matrix is low-rank relative to the full parameter dimension $m$. We also show how heavy censoring further reduces the computational burden.

To streamline what follows, let us introduce the shorthand

$$\mathbf{J}_i := \mathbf{J}_{\boldsymbol{\theta}_{\mathrm{MAP}}}(y_i, \mathbf{x}_i) \in \mathbb{R}^{m \times 1}$$

for $i = 1, \ldots, N$. With this notation (and dropping the CAVI-iteration index for clarity), the matrix $\mathbf{B} \in \mathbb{R}^{m \times m}$ defined in (A25) becomes

$$\mathbf{B} = \sum_{i=1}^{N} \frac{1}{2} \left( \delta_i \, \mathbb{E}_{\omega_i \sim q_{\omega_i}} [\omega_i] \, \mathbf{J}_i \, \mathbf{J}_i^T \; + \; \mathcal{I}_{3,i} \right) + \frac{1}{2} \, \mathbf{I}_m.$$

Here, each $\mathcal{I}_{3,i}$ is the integral

$$\mathcal{I}_{3,i} = \int_0^{y_i} \mathbf{J}_{\boldsymbol{\theta}_{\mathrm{MAP}}}(t, \mathbf{x}_i) \, \mathbf{J}_{\boldsymbol{\theta}_{\mathrm{MAP}}}(t, \mathbf{x}_i)^T \, \lambda_i^{\mathbb{Q}}(t) \frac{\tanh\left( \tilde{s}_i(t)/2 \right)}{2\tilde{s}_i(t)} \, \mathrm{d}t$$

and in general admits no closed-form solution. We therefore approximate it by any standard quadrature rule (e.g. trapezoid, Simpson's, or Gauss–Legendre). In what follows, we will illustrate the argument with the trapezoid rule, though the same steps apply to any other quadrature method.

We begin by introducing a uniform grid of points along the time axis:

$$t_1, t_2, \ldots, t_K,$$

where $t_1 := 0$ and $t_K := \max\{y_i\}_{i=1}^{N}$. We associate a set of quadrature weights $\{v_{ik}\}_{k=1}^{K}$ to the time grid points, tailored for each observation $i$. These weights correspond to the trapezoidal rule for numerical integration on the interval $[0, y_i]$, and are defined as:

$$v_{ik} = \begin{cases} \frac{t_2 - t_1}{2}, & \text{if } k = 1 \text{ and } t_1 < y_i, \\ \frac{t_{k+1} - t_{k-1}}{2}, & \text{if } 1 < k < K_i \text{ and } t_k < y_i, \\ \frac{t_{K_i} - t_{K_i-1}}{2}, & \text{if } k = K_i, \\ 0, & k > K_i, \end{cases}$$

where $K_i = \max\{k \in \{1, \ldots, K\} : t_k < y_i\}$. Further we denote by $\mathbf{V}_i$ the collection of quadrature weights for observation $i$, such that

$$\mathbf{V}_i := \left( v_{i1}, \ldots, v_{iK} \right) \; \in \; \mathbb{R}^K.$$

We collect the Jacobian evaluations into the matrices

$$\mathbf{Q}_i := [\mathbf{J}_{\boldsymbol{\theta}_{\mathrm{MAP}}}(t_1, \mathbf{x}_i) \quad \mathbf{J}_{\boldsymbol{\theta}_{\mathrm{MAP}}}(t_2, \mathbf{x}_i) \quad \cdots \quad \mathbf{J}_{\boldsymbol{\theta}_{\mathrm{MAP}}}(t_K, \mathbf{x}_i)] \; \in \; \mathbb{R}^{m \times K}.$$

With these definitions in hand, any $K$-point quadrature rule yields the approximation

$$\mathcal{I}_{3,i} \approx \sum_{k=1}^{K} v_{ik} \, \mathbf{J}_{\boldsymbol{\theta}_{\mathrm{MAP}}}(t_k, \mathbf{x}_i) \, \mathbf{J}_{\boldsymbol{\theta}_{\mathrm{MAP}}}(t_k, \mathbf{x}_i)^T = \mathbf{Q}_i \, \mathbf{V}_i \, \mathbf{Q}_i^T.$$

Likewise, each term

$$\delta_i \mathbb{E}_{\omega_i \sim q_{\omega_i}} [\omega_i] \mathbf{J}_i \mathbf{J}_i^T$$

can be written in the form $\mathbf{J}_i \mathbf{C}_i \mathbf{J}_i^T$, where the scalar $\mathbf{C}_i = \delta_i \mathbb{E}_{\omega_i \sim q_{\omega_i}} [\omega_i]$.

We collect all contributions into a single matrix $\mathbf{U} \in \mathbb{R}^{m \times R}$, where $R = N + NK$. This matrix is constructed by horizontally concatenating the vectors $\mathbf{J}_i$ and $\mathbf{Q}_i$ for $i = 1, \ldots, N$, as follows:

$$\mathbf{U} := \left[ \underbrace{\mathbf{J}_1}_{(m \times 1)}, \mathbf{J}_2, \ldots, \mathbf{J}_N, \underbrace{\mathbf{Q}_1}_{(m \times K)}, \mathbf{Q}_2, \ldots, \mathbf{Q}_N \right].$$

Further, we define the block-diagonal weight matrix

$$\mathbf{C} := \mathrm{diag}(\underbrace{\delta_1 \mathbb{E}_{\omega_1 \sim q_{\omega_1}}[\omega_1], \dots, \delta_N \mathbb{E}_{\omega_N \sim q_{\omega_N}}[\omega_N]}_{(N)}, \underbrace{\mathbf{V}_1}_{(K)}, \dots, \mathbf{V}_N) \in \mathbb{R}^{R \times R}.$$

It is straightforward to verify that

$$\mathbf{B} = \frac{1}{2}\left(\mathbf{I}_m + \mathbf{U}\mathbf{C}\mathbf{U}^\top\right).$$

Applying the Woodbury identity (see [20, Appendix B.10]) then reduces the inversion of $\mathbf{B}$ to that of an $R \times R$ matrix:

$$\mathbf{B}^{-1} = 2\left(\mathbf{I}_m + \mathbf{U}\,\mathbf{C}\,\mathbf{U}^T\right)^{-1} = 2\left[\mathbf{I}_m - \mathbf{U}\left(\mathbf{C}^{-1} + \mathbf{U}^T\mathbf{U}\right)^{-1}\mathbf{U}^T\right].$$

Forming the Gram matrix $\mathbf{U}^T\mathbf{U}$ requires $\mathcal{O}(mR^2)$ operations (each of its $R^2$ entries is an inner product of two length-$m$ vectors) while inverting the resulting dense $R \times R$ matrix costs $\mathcal{O}(R^3)$. Therefore, assembling and solving the small system costs

$$\mathcal{O}(mR^2) + \mathcal{O}(R^3) = \mathcal{O}(mR^2 + R^3)$$

instead of $\mathcal{O}(m^3)$ for a full $m \times m$ inversion. Whenever $R \ll m$, this yields a dramatic speed-up. By replacing the direct $\mathcal{O}(R^3)$ factorization with a Conjugate-Gradient (CG) solver — as is commonly done in Gaussian-process toolkits such as GPyTorch [14] — we reduce the cost to $\mathcal{O}(R^2)$.

Finally, many survival datasets exhibit censoring, i.e. $\delta_i = 0$ for a fraction of observations. Since censored observations contribute only through the integral term, we may further partition the low-rank factor $\mathbf{U}$ into blocks for uncensored and censored cases. The effective rank becomes $R' = N_{\text{uncensored}} + NK$ where $N_{\text{uncensored}}$ is the number of uncensored observations, so that any Cholesky or CG solve scales with $(N_{\text{uncensored}} + NK)$ rather than $(N + NK)$. When $N_{\text{uncensored}} \ll N$, this yields an additional, potentially large reduction in computational cost.

## J   Experiment Set-Up

### J.1   Real Survival Data

The real survival data used in Section 5.2 are presented below. In the central experiment, each dataset was subsampled to contain 125 observations in total. In an ablation experiment, each dataset was subsampled to contain 250 observations in total. Then, we performed 5-fold cross-validation, where the dataset was randomly divided into five equal parts. In each fold, one part (20%) was used as the test set (central experiment: 25 samples, ablation experiment: 50 samples), while the remaining four parts (80%) formed the training set (central experiment: 100 samples, ablation experiment: 200 samples). From the training set, 20% (central experiment: 20 samples, ablation experiment: 40 samples) was further attributed to the validation set.

**Colon.**   The first successful trials of adjuvant chemotherapy for colon cancer dataset was obtained from the `survival` package [46]. The dataset contains records of 1,822 observations with 15 covariates among which 49.23% are censored. All rows with missing values were excluded from the dataset.

**NWTCO.**   The National Wilm's Tumor Study (NWTCO) was obtained from the `pycox` package [27]. The dataset contains records of 4,028 observations with 7 covariates among which 14.18% are censored.

**GBSG.**   The Rotterdam and German Breast Cancer Study Group (GBSG) was obtained from the `pycox` package [27]. The dataset contains records of 2,232 observations with 7 covariates among which 43.23% are censored.

**METABRIC.**   The Molecular Taxonomy of Breast Cancer International Consortium (METABRIC) dataset was obtained from the `pycox` package [27]. The dataset contains records of 1,904 observations with 9 covariates among which 42.07% are censored.

**WHAS.**   The Worcester Heart Attack Study (WHAS) dataset was obtained from the `sksurv` package [39]. The dataset contains records of 500 observations with 14 covariates among which 43.00% are censored.

**SUPPORT.**   The Study to Understand Prognoses and Preferences for Outcomes and Risks of Treatment (SUPPORT) dataset was obtained from the `pycox` package [27]. The dataset contains records of 8,873 observations with 14 covariates among which 31.97% are censored.

**VLC.**   The Veterans administration Lung Cancer trial (VLC) dataset was obtained from the `sksurv` package [39]. The dataset contains records of 137 observations with 8 covariates among which 6.57% are censored.

**SAC 3.**   The Sac 3 dataset from the simulation study in [28, Appendix A.1] was obtained from the `pycox` package [27]. The dataset contains records of 100,000 observations with 45 covariates among which 37.20% are censored.

### J.2   Benchmark Methods

#### J.2.1   Benchmark Deep Survival Methods

All deep learning methods share the same neural network architecture, which is detailed in Section K. The benchmark deep survival models were trained using the Adam optimizer with a learning rate selected via grid search. Batch normalization was applied, and a dropout rate of 0.1 was used. Training was conducted for 1,000 epochs with a batch size of 256.

**MTLR.**   The Multi-Task Logistic Regression [51] was implemented using the `MTLR` class from the `pycox` package [27].

**DeepHit.** The DeepHit method [32] was implemented using the `DeepHitSingle` class from the `pycox` package [27]. The hyperparameters $\alpha$ and $\sigma$ were set to 0.2 and 0.1, respectively. Those are the default values.

**DeepSurv.** The DeepSurv model [24] was implemented using the `CoxPH` class from the `pycox` package [27].

**Logistic Hazard.** The Logistic Hazard method [51] was implemented using the `LogisticHazard` class from the `pycox` package [27].

**CoxTime.** The CoxTime method [29] was implemented using the `CoxTime` class from the `pycox` package [27].

**CoxCC.** The CoxCC method [29] was implemented using the `CoxCC` class from the `pycox` package [27].

**PMF.** The PMF method [28] was implemented using the `PMF` class from the `pycox` package [27].

**PCHazard.** The PCHazard method [28] was implemented using the `PCHazard` class from the `pycox` package [27].

**BCESurv.** The BCESurv method [28] was implemented using the `BCESurv` class from the `pycox` package [27].

**DySurv.** The DySurv method [36] was implemented using the official code provided by the authors, available at `https://github.com/munibmesinovic/DySurv/blob/main/Models/Results/Static_Benchmarks_GBSG_Example.ipynb` (Accessed on May 13 2025).

**Sumo-Net.** The Sumo-Net method [43] was implemented using the official code provided by the authors, available at `https://github.com/MrHuff/Sumo-Net` (Accessed on July 25 2025).

**DQS.** The DQS method [50] was implemented using the official code provided by the authors, available at `https://github.com/IBM/dqs` (Accessed on July 25 2025).

### J.2.2 Traditional Survival Methods

**CoxPH.** The Cox Proportional Hazards model [10] was implemented using the `CoxPHFitter` class from the `lifelines` package [11]. The Breslow method was used to compute the survival function.

**Weibull AFT.** The Weibull Accelerated Failure Time model [8] was implemented using the `WeibullAFTFitter` class from the `lifelines` package [11].

**RSF.** The Random Survival Forest [23] was implemented using the `RandomSurvivalForest` class from the `sksurv` package [39]. The number of trees in the forest is set to 1,000. The minimum number of samples required to split an internal node is 10, and the minimum number of samples required to be at a leaf node is 15. Those were the same hyperparameters as used in [36].

**SSVM.** The Survival Support Vector Machine [40] was implemented using the `FastSurvivalSVM` class from the `sksurv` package [39]. The optimal regularization hyperparameter $\alpha$ was selected via grid search by evaluating model performance on the training set using the C-index. This method does not allow for estimation of the survival function. Predicted ranks were used as risk scores for computing the C-index.

## J.3 Evaluation metrics

**C-index.** Let $\hat{q}_i(t)$ be the predicted risk score of observation with covariates $\mathbf{x}_i$ at time $t$. The C-index estimate [19] is given by

$$\text{C-index} = \frac{\sum_{i=1}^{N}\sum_{j\neq i}\delta_i\,\mathbb{1}_{\{y_i<y_i\}}\left(\mathbb{1}_{\{\hat{q}_i(y_i)>\hat{q}_j(y_i)\}}+\frac{1}{2}\mathbb{1}_{\{\hat{q}_i(y_i)=\hat{q}_j(y_i)\}}\right)}{\sum_{i=1}^{N}\sum_{j\neq i}\delta_i\,\mathbb{1}_{\{y_i<y_j\}}}.$$

Let $\hat{S}_i(t)$ be the predicted survival function of observation with covariates $\mathbf{x}_i$ at time $t$. When the predicted risk score is taken to be the negative of the survival function, i.e., $\hat{q}_i(t) = -\hat{S}_i(t)$, the C-index is referred to as the Antolini's C-index [3] and is found with

$$\text{C-index} = \frac{\sum_{i=1}^{N}\sum_{j\neq i}\delta_i\,\mathbb{1}_{\{y_i<y_i\}}\left(\mathbb{1}_{\{\hat{S}_i(y_i)<\hat{S}_j(y_i)\}}+\frac{1}{2}\mathbb{1}_{\{\hat{S}_i(y_i)=\hat{S}_j(y_i)\}}\right)}{\sum_{i=1}^{N}\sum_{j\neq i}\delta_i\,\mathbb{1}_{\{y_i<y_j\}}}.$$

The C-index is obtained using the `ConcordanceIndex` class from the `TorchSurv` package [37].

**IPCW Integrated Brier Score.** Let $\hat{S}_i(t)$ be the predicted survival function of observation with covariates $\mathbf{x}_i$ at time $t$. Let the inverse probability censoring weight (IPCW) at time $t$ be defined as the inverse of the probability of being uncensored, $\xi(t) = 1/\hat{C}(t)$, where $\hat{C}(t)$ denotes the Kaplan–Meier estimate of the censoring survival function. Under right censorship, the IPCW Brier score (BS) [17] at time $t$ is given by

$$\text{IPCW BS}(t) = \frac{1}{N}\sum_{i=1}^{N}\xi(y_i)\mathbb{1}_{\{y_i\leq t,\delta_i=1\}}(0-\hat{S}_i(t))^2 + \xi(t)\mathbb{1}_{\{y_i>t\}}(1-\hat{S}_i(t))^2. \tag{A29}$$

The IBS is the integral of the Brier Score in (A29). The IPCW weights and the IPCW IBS are computed using the `get_ipcw` function and the `BrierScore` class from the `TorchSurv` package [37].

**Distribution Calibration.** D-Calibration [18] evaluates whether predicted survival probabilities at observed times are uniformly distributed. For an individual $i$, let $\hat{S}_i(y_i)$ be the predicted survival probability at their event or censoring time $y_i$. Under perfect calibration, we expect:

$$\hat{S}_i(y_i) \sim \text{Uniform}(0,1). \tag{A30}$$

The predicted probabilities are binned into $B$ quantiles, and a histogram is constructed over both event and censored observations. For censored data, the probability is distributed proportionally across bins beyond the censoring time. A chi-squared test compares the resulting histogram to the expected uniform distribution, and the p-value reflects how well the survival model is calibrated. The D-Calibration is obtained using the `LifelinesEvaluator.d_calibration()` class from the `SurvivalEVAL` package [41].

**Kaplan-Meier Calibration.** Kaplan-Meier (KM)-Calibration, as introduced by [9], evaluates how well the average predicted survival curve from a model aligns with the empirical KM survival curve. Let $\hat{S}_{\text{avg}}(t)$ denote the model's average survival probability at time $t$, and $\hat{S}_{\text{KM}}(t)$ the KM estimate. The KM calibration score is defined as the normalized integrated mean squared error (MSE):

$$\text{KM-Calibration} = \frac{1}{T_{\max}}\int_0^{T_{\max}}\left(\hat{S}_{\text{avg}}(t)-\hat{S}_{\text{KM}}(t)\right)^2 dt. \tag{A31}$$

This score lies in $[0,1]$, where 0 indicates perfect calibration, and values near 0.25 represent uninformative predictions. The KM-Calibration is obtained using the `LifelinesEvaluator.km_calibration()` class from the `SurvivalEVAL` package [41].

# K   Implementation Details

**Code availability.**   The code is available on the GitHub repository `https://github.com/MLGlobalHealth/neuralsurv` under the MIT License.

**Architecture.**   We employed a feedforward neural network with two hidden layers, each containing 16 neurons and using ReLu activations. The input of the network for observation $i = 1, \ldots, N$ is the pair $(t, \mathbf{x}_i)$.

**Time normalization.**   The observation period is normalized to the interval $[0, 1]$ by dividing each time value by the maximum observed time in the training set.

**EM algorithm.**   The parameters are initialized so that they match their prior expected values. Specifically, we set $\boldsymbol{\theta}^{(0)} = \mathbf{0}$ and $\phi^{(0)} = \alpha_0/\beta_0$. The maximization step of the EM algorithm is performed using the L-BFGS-B algorithm. The EM algorithm is considered to have converged when the relative change in the Q-function between consecutive iterations falls below a tolerance threshold of $10^{-6}$ for two successive iterations.

**CAVI algorithm.**   The hyperparameters are initialized so that the expected values of the model parameters match the MAP estimates. Specifically, we set $\tilde{\alpha}^{(0)} = \phi_{\text{MAP}} \times \tilde{\beta}$,   and   $(\tilde{\boldsymbol{\mu}}, \tilde{\boldsymbol{\Sigma}})^{(0)} = (\boldsymbol{\theta}_{\text{MAP}}, \mathbf{I}_m)$. The CAVI algorithm is considered to have converged when the relative change between successive parameter estimates falls below a tolerance threshold of $10^{-6}$.

**Integral approximation.**   The integrals required to compute the Q-function in the EM algorithm, as well as those involved in the optimal variational updates of $\phi$ and $\boldsymbol{\theta}$ in the CAVI algorithm, are approximated using the trapezoidal rule.

**Prior and $\rho$.**   For all experiments, we fix the hyperparameters of the prior distribution over $\phi$, given in (6), to be $\alpha_0, \beta_0 = 1$. Furthermore, we fix $\rho = 1$.

**Machine.**   The experiments were conducted on NVIDIA RTX A6000 GPUs with 48GB of memory.

**Running time**   Table A1 reports the running time for a single fold on the Colon dataset at varying sample sizes. All folds and datasets were processed in parallel across multiple GPUs to ensure consistent timing.

| Method | $N = 25$ | | | $N = 125$ | | | $N = 250$ | | |
|---|---|---|---|---|---|---|---|---|---|
| | 1L | 2L-6U | 2L-16U | 1L | 2L-6U | 2L-16U | 1L | 2L-6U | 2L-16U |
| MTLR [51] | 0.104 | 0.102 | 0.100 | 0.075 | 0.093 | 0.118 | 0.075 | 0.099 | 0.091 |
| DeepHit [32] | 0.164 | 0.139 | 0.140 | 0.120 | 0.126 | 0.142 | 0.122 | 0.145 | 0.135 |
| DeepSurv [24] | 0.082 | 0.098 | 0.072 | 0.063 | 0.079 | 0.072 | 0.071 | 0.080 | 0.101 |
| Logistic Hazard [16] | 0.077 | 0.079 | 0.076 | 0.070 | 0.090 | 0.072 | 0.072 | 0.078 | 0.079 |
| CoxTime [29] | 0.139 | 0.167 | 0.114 | 0.113 | 0.175 | 0.134 | 0.136 | 0.135 | 0.186 |
| CoxCC [29] | 0.130 | 0.118 | 0.094 | 0.100 | 0.119 | 0.114 | 0.107 | 0.132 | 0.149 |
| PMF [28] | 0.129 | 0.119 | 0.094 | 0.069 | 0.104 | 0.095 | 0.072 | 0.087 | 0.125 |
| PCHazard [28] | 0.083 | 0.102 | 0.083 | 0.087 | 0.092 | 0.084 | 0.091 | 0.091 | 0.091 |
| BCESurv [30] | 0.081 | 0.082 | 0.092 | 0.060 | 0.078 | 0.092 | 0.062 | 0.087 | 0.079 |
| DySurv [36] | 0.050 | 0.064 | 0.040 | 0.048 | 0.040 | 0.049 | 0.048 | 0.045 | 0.041 |
| Sumo-Net [43] | 0.058 | 0.058 | 0.060 | 0.062 | 0.071 | 0.070 | 0.073 | 0.073 | 0.078 |
| DQS [50] | 0.019 | 0.020 | 0.024 | 0.023 | 0.025 | 0.025 | 0.029 | 0.030 | 0.030 |
| NeuralSurv (Ours) | 0.621 | 1.165 | 0.566 | 3.926 | 16.454 | 22.673 | 7.373 | 87.098 | 141.389 |

Table A1: Inference runtime for the Colon dataset (in minutes). The central analysis presented in Table A5-A6 is for $N = 125$ and a MNP with 2 layers (2L) and 16 units (16U). The ablation study presented in Table A7-A8 is for $N = 250$ and a MNP with 2L and 16U.

# L   Related Work

Survival analysis methodologies have evolved significantly over the past decades, encompassing parametric, semi-parametric, non-parametric, and more recently, deep learning-based approaches. We review these developments, focusing on their applicability to high-dimensional data and uncertainty quantification capabilities.

**Parametric and Semi-parametric Traditional Models.**    Traditional survival models often impose parametric or semi-parametric assumptions on the hazard function. The Accelerated Failure Time (AFT) model [8] assumes a linear relationship between covariates and the logarithm of survival time, with parametric baseline distributions (e.g., Weibull). While interpretable, such models struggle with high-dimensional data and nonlinear covariate effects. The Cox Proportional Hazards (CoxPH) model [10], a semi-parametric approach, avoids specifying the baseline hazard but assumes proportional hazards. Though widely adopted, CoxPH's linear predictor and proportionality constraints limit its flexibility in complex data regimes.

**Non-parametric Traditional Models.**    To mitigate parametric assumptions, non-parametric methods like Random Survival Forests (RSF) [23] and Survival Support Vector Machines (SSVM) [40] emerged. RSF leverages ensemble learning for risk stratification but faces challenges in high-dimensional settings due to greedy tree induction. GP survival models [13] offer flexibility by modeling the hazard function nonparametrically, with inherent uncertainty quantification. Existing work has sought to address the cubic complexity in sample size of GPs by introducing variational inference techniques [25]. However, GPs remain fundamentally limited in scalability, particularly struggling with high-dimensional inputs and lacking the capacity to learn hierarchical representations, such as those required in image-based tasks [44].

**Deep Survival Models.**    The advent of deep learning revolutionized survival analysis by enabling automatic feature learning from high-dimensional inputs. DeepSurv [24] extended CoxPH with neural networks, while DeepHit [32] employed multi-task learning for competing risks via discrete-time hazards. Discrete-time methods, including MTLR [51] and PCHazard [28], discretize the time axis to simplify likelihood computation, with recent advances like DySurv [36] incorporating conditional variational inference for dynamic prediction. Sumo-Net [43] introduces a partially monotonic NN that directly optimizes the right-censored log-likelihood, which is proven to be a strictly proper scoring rule—achieving strong log-likelihood performance. DQS [50] formulates survival prediction using extensions of strictly proper scoring rules that remain proper under discrete-time survival settings. Despite their predictive prowess, these models rely on frequentist training, yielding point estimates without uncertainty quantification, a significant shortcoming in safety-critical applications. Comprehensive reviews [49] highlight the rapid growth of deep survival methods but underscore their neglect of probabilistic uncertainty.

**Bayesian and Uncertainty-Aware Approaches.**    Bayesian methods provide a natural framework for uncertainty quantification but have seen limited integration with deep survival models. GP-based approaches [13, 25] inherit GP limitations in scalability and high-dimensional processing. Recent works like BCESurv [30] explore bootstrap confidence intervals, yet these post-hoc approximations lack the coherence of Bayesian posteriors. Consequently, existing Bayesian survival models either sacrifice scalability for uncertainty quantification or compromise on model flexibility, leaving a critical gap in high-dimensional, uncertainty-aware survival analysis.

**Summary.**    While parametric and semi-parametric models provide interpretability, they falter in high-dimensional, nonlinear regimes. Non-parametric methods like RSF and GP improve flexibility but face scalability challenges. Deep learning approaches excel at feature extraction yet lack principled uncertainty quantification. Bayesian methods, though theoretically sound, remain confined to traditional architectures or partial approximations. Our work bridges this divide by proposing the first scalable, deep Bayesian survival model that harmonizes neural networks with full probabilistic uncertainty, addressing a critical need in modern applications.

| Method | Uncertainty (Bayesian) | Continuous Time | Deep Learning |
|---|:---:|:---:|:---:|
| CoxPH [10] | ✓ | ✓ | ✗ |
| AFT [8] | ✓ | ✓ | ✗ |
| RSF [23] | ✗ | ✓ | ✗ |
| SSVM [40] | ✗ | ✓ | ✗ |
| GP survival models [13, 25] | ✓ | ✓ | ✗ |
| MTLR [51] | ✗ | ✗ | ✓ |
| DeepHit [32] | ✗ | ✗ | ✓ |
| DeepSurv [24] | ✗ | ✓ | ✓ |
| Logistic Hazard [16] | ✗ | ✗ | ✓ |
| CoxTime [29] | ✗ | ✓ | ✓ |
| CoxCC [29] | ✗ | ✓ | ✓ |
| PMF [28] | ✗ | ✗ | ✓ |
| PCHazard [28] | ✗ | ✓ | ✓ |
| BCESurv [30] | ✗ | ✗ | ✓ |
| DySurv [36] | ✗ | ✓ | ✓ |
| Sumo-Net [43] | ✗ | ✓ | ✓ |
| DQS [50] | ✗ | ✓ | ✓ |
| *NeuralSurv (Ours)* | ✓ | ✓ | ✓ |

Table A2: Summary of Survival Analysis methods: Bayesian Uncertainty Quantification, Time Domain, and Deep-Learning Status.

# M  Further Results

## M.1  Synthetic Data Experiment

| Method | N = 25 | | N = 50 | | N = 100 | | N = 150 | |
|---|---|---|---|---|---|---|---|---|
| | C-index ↑ | IPCW IBS ↓ | C-index ↑ | IPCW IBS ↓ | C-index ↑ | IPCW IBS ↓ | C-index ↑ | IPCW IBS ↓ |
| MTLR [51] | 0.560 | 0.284 | 0.505 | 0.239 | 0.491 | 0.171 | 0.542 | 0.17 |
| DeepHit [32] | 0.473 | 0.239 | 0.469 | 0.214 | 0.502 | 0.171 | 0.574 | 0.114 |
| DeepSurv [24] | 0.492 | 0.313 | 0.471 | 0.241 | 0.507 | 0.169 | 0.517 | 0.169 |
| Logistic Hazard [16] | 0.477 | 0.297 | 0.498 | 0.256 | 0.507 | 0.199 | 0.499 | 0.176 |
| CoxTime [29] | 0.424 | 0.284 | 0.532 | 0.273 | 0.52 | 0.184 | 0.575 | 0.118 |
| CoxCC [29] | 0.421 | 0.268 | 0.497 | 0.229 | 0.526 | 0.128 | 0.513 | 0.109 |
| PMF [28] | **0.573** | 0.261 | 0.551 | 0.334 | 0.523 | 0.168 | **0.607** | 0.184 |
| PCHazard [28] | 0.477 | 0.337 | 0.501 | 0.249 | 0.467 | 0.174 | 0.486 | 0.193 |
| BCESurv [30] | 0.545 | 0.287 | **0.585** | 0.256 | 0.558 | 0.185 | 0.559 | 0.16 |
| DySurv [36] | 0.399 | 0.237 | 0.491 | 0.239 | 0.459 | 0.218 | 0.489 | 0.174 |
| Sumo-Net [43] | 0.473 | 0.223 | 0.503 | 0.179 | 0.588 | 0.127 | 0.495 | 0.113 |
| DQS [50] | 0.435 | 0.326 | 0.48 | 0.232 | 0.525 | 0.131 | 0.556 | 0.124 |
| NeuralSurv (Ours) | 0.378 | **0.196** | 0.554 | **0.160** | **0.589** | **0.126** | 0.589 | **0.106** |

Table A3: Performance comparison of survival models over synthetic data. The best results for each metric are shown in bold, and the second-best results are underlined. ↑ indicates higher is better; ↓ indicates lower is better.

| Method | N = 25 | | N = 50 | |
|---|---|---|---|---|
| | D-Calibration (p-value) | KM-Calibration ↓ | D-Calibration (p-value) | KM-Calibration ↓ |
| MTLR [51] | 0.000 (×) | 0.032 | 0.000 (×) | 0.056 |
| DeepHit [32] | 0.000 (×) | 0.029 | 0.000 (×) | 0.168 |
| DeepSurv [24] | 0.000 (×) | 0.009 | 0.000 (×) | 0.036 |
| Logistic Hazard [16] | 0.000 (×) | 0.037 | 0.000 (×) | 0.062 |
| CoxTime [29] | 0.000 (×) | 0.010 | 0.000 (×) | 0.021 |
| CoxCC [29] | 0.000 (×) | **0.008** | 0.000 (×) | 0.028 |
| PMF [28] | 0.000 (×) | 0.032 | 0.000 (×) | 0.071 |
| PCHazard [28] | 0.000 (×) | 0.037 | 0.000 (×) | 0.073 |
| BCESurv [30] | 0.000 (×) | 0.022 | 0.000 (×) | 0.014 |
| DySurv [36] | 0.000 (×) | 0.201 | 0.000 (×) | 0.241 |
| Sumo-Net [43] | 0.132 (✓) | 0.011 | 0.051 (✓) | 0.036 |
| DQS [50] | 0.000 (×) | 0.051 | 0.000 (×) | 0.048 |
| NeuralSurv (Ours) | 0.833 (✓) | 0.014 | 0.539 (✓) | **0.012** |

| Method | N = 100 | | N = 150 | |
|---|---|---|---|---|
| | D-Calibration (p-value) | KM-Calibration ↓ | D-Calibration (p-value) | KM-Calibration ↓ |
| MTLR [51] | 0.000 (×) | 0.034 | 0.000 (×) | 0.035 |
| DeepHit [32] | 0.000 (×) | 0.030 | 0.000 (×) | 0.040 |
| DeepSurv [24] | 0.000 (×) | 0.003 | 0.007 (×) | 0.003 |
| Logistic Hazard [16] | 0.000 (×) | 0.034 | 0.000 (×) | 0.062 |
| CoxTime [29] | 0.000 (×) | 0.011 | 0.346 (✓) | 0.005 |
| CoxCC [29] | 0.000 (×) | **0.001** | 0.001 (×) | **0.001** |
| PMF [28] | 0.000 (×) | 0.033 | 0.000 (×) | 0.034 |
| PCHazard [28] | 0.000 (×) | 0.056 | 0.000 (×) | 0.066 |
| BCESurv [30] | 0.000 (×) | 0.038 | 0.000 (×) | 0.034 |
| DySurv [36] | 0.000 (×) | 0.172 | 0.000 (×) | 0.183 |
| Sumo-Net [43] | 0.683 (✓) | 0.004 | 0.345 (✓) | 0.003 |
| DQS [50] | 0.005 (×) | 0.032 | 0.000 (×) | 0.031 |
| NeuralSurv (Ours) | 0.419 (✓) | 0.004 | 0.639 (✓) | 0.004 |

Table A4: Performance comparison of survival models over synthetic data (part 2). A checkmark (✓) indicates that the null hypothesis of perfect D-Calibration was not rejected at $\alpha = 0.05$ (model considered well-calibrated); a cross (×) indicates rejection of D-Calibration (model considered not well-calibrated). The best results for the KM-Calibration are shown in bold, and the second-best results are underlined. ↓ indicates lower is better.

## M.2 Real Data Experiment

### M.2.1 Central Analysis

| | COLON | | METABRIC | | GBSG | |
|---|---|---|---|---|---|---|
| Method | C-index ↑ | IPCW IBS ↓ | C-index ↑ | IPCW IBS ↓ | C-index ↑ | IPCW IBS ↓ |
| MTLR [51] | 0.562 | 0.298 | 0.548 | 0.279 | 0.602 | 0.273 |
| DeepHit [32] | 0.478 | 0.28 | 0.511 | 0.243 | 0.578 | 0.309 |
| DeepSurv [24] | 0.572 | 0.326 | 0.523 | 0.289 | 0.618 | 0.252 |
| Logistic Hazard [16] | 0.490 | 0.321 | 0.541 | 0.317 | 0.618 | 0.296 |
| CoxTime [29] | 0.578 | 0.277 | 0.533 | 0.307 | 0.599 | 0.285 |
| CoxCC [29] | 0.584 | 0.289 | 0.575 | 0.257 | 0.646 | 0.240 |
| PMF [28] | 0.509 | 0.324 | 0.440 | 0.336 | 0.655 | 0.250 |
| PCHazard [28] | 0.538 | 0.297 | 0.541 | 0.291 | 0.609 | 0.249 |
| BCESurv [30] | 0.491 | 0.302 | **0.616** | 0.277 | 0.581 | 0.273 |
| DySurv [36] | 0.488 | 0.536 | 0.561 | 0.465 | 0.572 | 0.485 |
| Sumo-Net [43] | 0.485 | 0.241 | 0.447 | 0.223 | 0.476 | 0.250 |
| DQS [50] | 0.635 | 0.246 | 0.564 | 0.261 | 0.611 | 0.229 |
| NeuralSurv (Ours) | **0.671** | **0.218** | 0.584 | **0.212** | **0.657** | **0.188** |

| | NWTCO | | WHAS | | SUPPORT | |
|---|---|---|---|---|---|---|
| Method | C-index ↑ | IPCW IBS ↓ | C-index ↑ | IPCW IBS ↓ | C-index ↑ | IPCW IBS ↓ |
| MTLR [51] | 0.592 | 0.301 | 0.490 | 0.315 | 0.432 | 0.357 |
| DeepHit [32] | 0.516 | 0.296 | 0.510 | 0.303 | 0.452 | 0.341 |
| DeepSurv [24] | 0.527 | 0.248 | 0.654 | 0.281 | 0.505 | 0.354 |
| Logistic Hazard [16] | 0.512 | 0.298 | 0.545 | 0.315 | 0.536 | 0.378 |
| CoxTime [29] | 0.550 | 0.199 | **0.678** | 0.250 | 0.547 | 0.327 |
| CoxCC [29] | 0.531 | 0.237 | 0.654 | 0.281 | 0.566 | 0.312 |
| PMF [28] | 0.482 | 0.312 | 0.520 | 0.299 | 0.512 | 0.399 |
| PCHazard [28] | 0.551 | 0.209 | 0.527 | 0.291 | 0.514 | 0.335 |
| BCESurv [30] | 0.530 | 0.272 | 0.548 | 0.292 | 0.446 | 0.398 |
| DySurv [36] | 0.402 | 0.683 | 0.424 | 0.523 | 0.525 | 0.342 |
| Sumo-Net [43] | 0.595 | 0.170 | 0.556 | 0.260 | 0.444 | **0.289** |
| DQS [50] | 0.567 | 0.242 | 0.590 | 0.269 | 0.538 | 0.331 |
| NeuralSurv (Ours) | **0.712** | **0.166** | 0.602 | **0.233** | **0.599** | 0.333 |

| | VLC | | SAC3 | |
|---|---|---|---|---|
| Method | C-index ↑ | IPCW IBS ↓ | C-index ↑ | IPCW IBS ↓ |
| MTLR [51] | 0.432 | 0.299 | 0.471 | 0.276 |
| DeepHit [32] | 0.409 | 0.236 | 0.456 | 0.289 |
| DeepSurv [24] | 0.642 | 0.186 | 0.530 | 0.264 |
| Logistic Hazard [16] | 0.413 | 0.272 | 0.480 | 0.348 |
| CoxTime [29] | **0.671** | 0.212 | 0.485 | 0.276 |
| CoxCC [29] | 0.645 | 0.169 | **0.533** | 0.261 |
| PMF [28] | 0.445 | 0.284 | 0.472 | 0.270 |
| PCHazard [28] | 0.502 | 0.294 | 0.527 | 0.276 |
| BCESurv [30] | 0.428 | 0.263 | 0.440 | 0.300 |
| DySurv [36] | 0.436 | 0.162 | 0.476 | 0.303 |
| Sumo-Net [43] | 0.527 | 0.157 | 0.457 | 0.237 |
| DQS [50] | 0.568 | 0.218 | 0.481 | 0.293 |
| NeuralSurv (Ours) | 0.667 | **0.142** | 0.532 | **0.204** |

Table A5: Performance comparison of deep survival models over five different train/test splits of each dataset. The best results for each metric are shown in bold, and the second-best results are underlined. ↑ indicates higher is better; ↓ indicates lower is better.

| Method | COLON | | METABRIC | |
| --- | --- | --- | --- | --- |
| | D-Calibration (p-value) | KM-Calibration ↓ | D-Calibration (p-value) | KM-Calibration ↓ |
| MTLR [51] | 0.000 (×) | 0.016 | 0.000 (×) | 0.023 |
| DeepHit [32] | 0.001 (×) | 0.089 | 0.101 (✓) | 0.064 |
| DeepSurv [24] | 0.047 (×) | 0.024 | 0.006 (×) | 0.012 |
| Logistic Hazard [16] | 0.002 (×) | 0.019 | 0.000 (×) | 0.026 |
| CoxTime [29] | 0.014 (×) | **0.011** | 0.012 (×) | 0.012 |
| CoxCC [29] | 0.011 (×) | **0.011** | 0.013 (×) | 0.019 |
| PMF [28] | 0.001 (×) | 0.017 | 0.002 (×) | 0.017 |
| PCHazard [28] | 0.007 (×) | 0.029 | 0.008 (×) | 0.036 |
| BCESurv [30] | 0.000 (×) | 0.012 | 0.000 (×) | 0.021 |
| DySurv [36] | 0.000 (×) | 0.362 | 0.000 (×) | 0.306 |
| Sumo-Net [43] | 0.741 (✓) | 0.014 | 0.600 (✓) | **0.008** |
| DQS [50] | 0.381 (✓) | 0.017 | 0.135 (✓) | 0.020 |
| NeuralSurv (Ours) | 0.594 (✓) | 0.020 | 0.661 (✓) | 0.012 |

| Method | GBSG | | NWTCO | |
| --- | --- | --- | --- | --- |
| | D-Calibration (p-value) | KM-Calibration ↓ | D-Calibration (p-value) | KM-Calibration ↓ |
| MTLR [51] | 0.000 (×) | 0.011 | 0.570 (✓) | 0.011 |
| DeepHit [32] | 0.000 (×) | 0.149 | 0.575 (✓) | 0.014 |
| DeepSurv [24] | 0.084 (✓) | 0.009 | 0.887 (✓) | **0.003** |
| Logistic Hazard [16] | 0.008 (×) | 0.017 | 0.397 (✓) | 0.012 |
| CoxTime [29] | 0.247 (✓) | 0.009 | 0.883 (✓) | **0.003** |
| CoxCC [29] | 0.256 (✓) | **0.005** | 0.954 (✓) | **0.003** |
| PMF [28] | 0.003 (×) | 0.015 | 0.487 (✓) | 0.013 |
| PCHazard [28] | 0.063 (✓) | 0.024 | 0.697 (✓) | 0.008 |
| BCESurv [30] | 0.001 (×) | 0.015 | 0.312 (✓) | 0.010 |
| DySurv [36] | 0.000 (×) | 0.369 | 0.000 (×) | 0.260 |
| Sumo-Net [43] | 0.524 (✓) | 0.010 | 0.991 (✓) | 0.004 |
| DQS [50] | 0.234 (✓) | 0.012 | 0.916 (✓) | 0.005 |
| NeuralSurv (Ours) | 0.735 (✓) | 0.009 | 0.920 (✓) | 0.007 |

| Method | WHAS | | SUPPORT | |
| --- | --- | --- | --- | --- |
| | D-Calibration (p-value) | KM-Calibration ↓ | D-Calibration (p-value) | KM-Calibration ↓ |
| MTLR [51] | 0.001 (×) | 0.035 | 0.000 (×) | 0.099 |
| DeepHit [32] | 0.195 (✓) | 0.067 | 0.000 (×) | 0.124 |
| DeepSurv [24] | 0.076 (✓) | 0.025 | 0.000 (×) | **0.015** |
| Logistic Hazard [16] | 0.013 (×) | 0.033 | 0.000 (×) | 0.095 |
| CoxTime [29] | 0.035 (×) | **0.017** | 0.000 (×) | 0.016 |
| CoxCC [29] | 0.106 (✓) | 0.019 | 0.000 (×) | 0.020 |
| PMF [28] | 0.003 (×) | 0.031 | 0.000 (×) | 0.084 |
| PCHazard [28] | 0.125 (✓) | 0.022 | 0.000 (×) | 0.070 |
| BCESurv [30] | 0.000 (×) | 0.025 | 0.000 (×) | 0.088 |
| DySurv [36] | 0.000 (×) | 0.281 | 0.000 (×) | 0.099 |
| Sumo-Net [43] | 0.735 (✓) | 0.021 | 0.143 (✓) | 0.017 |
| DQS [50] | 0.081 (✓) | 0.033 | 0.002 (×) | 0.038 |
| NeuralSurv (Ours) | 0.335 (✓) | 0.031 | 0.063 (✓) | 0.083 |

| Method | VLC | | SAC3 | |
| --- | --- | --- | --- | --- |
| | D-Calibration (p-value) | KM-Calibration ↓ | D-Calibration (p-value) | KM-Calibration ↓ |
| MTLR [51] | 0.000 (×) | 0.072 | 0.000 (×) | 0.020 |
| DeepHit [32] | 0.000 (×) | 0.107 | 0.003 (×) | 0.094 |
| DeepSurv [24] | 0.004 (×) | **0.006** | 0.000 (×) | 0.020 |
| Logistic Hazard [16] | 0.000 (×) | 0.078 | 0.000 (×) | 0.039 |
| CoxTime [29] | 0.021 (×) | 0.010 | 0.000 (×) | **0.014** |
| CoxCC [29] | 0.004 (×) | 0.011 | 0.000 (×) | 0.016 |
| PMF [28] | 0.000 (×) | 0.077 | 0.000 (×) | 0.016 |
| PCHazard [28] | 0.000 (×) | 0.080 | 0.000 (×) | 0.029 |
| BCESurv [30] | 0.000 (×) | 0.074 | 0.000 (×) | 0.021 |
| DySurv [36] | 0.077 (✓) | 0.028 | 0.000 (×) | 0.146 |
| Sumo-Net [43] | 0.131 (✓) | 0.011 | 0.478 (✓) | 0.016 |
| DQS [50] | 0.000 (×) | 0.054 | 0.021 (×) | 0.034 |
| NeuralSurv (Ours) | 0.436 (✓) | 0.013 | 0.624 (✓) | 0.016 |

Table A6: Performance comparison of deep survival models over five different train/test splits of each dataset (part 2). A checkmark (✓) indicates that the null hypothesis of perfect D-Calibration was not rejected at $\alpha = 0.05$ (model considered well-calibrated); a cross (×) indicates rejection of D-Calibration (model considered not well-calibrated). The best results for the KM-Calibration are shown in bold, and the second-best results are underlined. ↓ indicates lower is better.

## M.2.2 Ablation Study with $N = 250$

| Method | COLON | | METABRIC | | GBSG | |
|---|---|---|---|---|---|---|
| | C-index ↑ | IPCW IBS ↓ | C-index ↑ | IPCW IBS ↓ | C-index ↑ | IPCW IBS ↓ |
| MTLR [51] | 0.545 | 0.291 | 0.572 | 0.290 | 0.567 | 0.312 |
| DeepHit [32] | 0.564 | 0.284 | 0.545 | 0.301 | 0.563 | 0.272 |
| DeepSurv [24] | 0.600 | 0.295 | 0.605 | 0.265 | 0.531 | 0.277 |
| Logistic Hazard [16] | 0.501 | 0.289 | 0.553 | 0.252 | 0.562 | 0.287 |
| CoxTime [29] | 0.621 | 0.259 | **0.621** | 0.264 | **0.578** | 0.255 |
| CoxCC [29] | **0.640** | 0.277 | 0.610 | 0.254 | 0.565 | 0.244 |
| PMF [28] | 0.541 | 0.291 | 0.554 | 0.300 | 0.537 | 0.304 |
| PCHazard [28] | 0.549 | 0.280 | 0.561 | 0.246 | 0.524 | 0.295 |
| BCESurv [30] | 0.537 | 0.289 | 0.565 | 0.289 | 0.554 | 0.301 |
| DySurv [36] | 0.478 | 0.543 | 0.516 | 0.491 | 0.506 | 0.508 |
| Sumo-Net [43] | 0.529 | 0.273 | 0.473 | 0.27 | 0.471 | 0.255 |
| DQS [50] | 0.593 | 0.267 | 0.600 | 0.228 | 0.562 | 0.237 |
| NeuralSurv (Ours) | 0.601 | **0.215** | 0.543 | **0.198** | 0.546 | **0.212** |

Table A7: Performance comparison of deep survival models on the ablation study with 250 observations, over five different train/test splits of each dataset. The best results for each metric are shown in bold, and the second-best results are underlined. ↑ indicates higher is better; ↓ indicates lower is better.

| Method | COLON | | METABRIC | |
|---|---|---|---|---|
| | D-Calibration (p-value) | KM-Calibration ↓ | D-Calibration (p-value) | KM-Calibration ↓ |
| MTLR [51] | 0.000 (×) | 0.007 | 0.000 (×) | 0.017 |
| DeepHit [32] | 0.000 (×) | 0.068 | 0.044 (×) | 0.072 |
| DeepSurv [24] | 0.001 (×) | 0.006 | 0.001 (×) | 0.007 |
| Logistic Hazard [16] | 0.000 (×) | 0.012 | 0.000 (×) | 0.017 |
| CoxTime [29] | 0.022 (×) | 0.005 | 0.001 (×) | 0.008 |
| CoxCC [29] | 0.001 (×) | **0.003** | 0.000 (×) | **0.006** |
| PMF [28] | 0.000 (×) | 0.008 | 0.000 (×) | 0.008 |
| PCHazard [28] | 0.001 (×) | 0.018 | 0.003 (×) | 0.027 |
| BCESurv [30] | 0.000 (×) | 0.012 | 0.000 (×) | 0.013 |
| DySurv [36] | 0.000 (×) | 0.376 | 0.000 (×) | 0.273 |
| Sumo-Net [43] | 0.323 (✓) | 0.007 | 0.192 (✓) | 0.007 |
| DQS [50] | 0.086 (✓) | 0.009 | 0.137 (✓) | 0.013 |
| NeuralSurv (Ours) | 0.404 (✓) | 0.011 | 0.708 (✓) | 0.013 |

| Method | GBSG | |
|---|---|---|
| | D-Calibration (p-value) | KM-Calibration ↓ |
| MTLR [51] | 0.000 (×) | 0.010 |
| DeepHit [32] | 0.159 (✓) | 0.134 |
| DeepSurv [24] | 0.052 (✓) | 0.006 |
| Logistic Hazard [16] | 0.000 (×) | 0.016 |
| CoxTime [29] | 0.171 (✓) | 0.004 |
| CoxCC [29] | 0.059 (✓) | 0.006 |
| PMF [28] | 0.000 (×) | 0.010 |
| PCHazard [28] | 0.004 (×) | 0.015 |
| BCESurv [30] | 0.000 (×) | 0.013 |
| DySurv [36] | 0.000 (×) | 0.336 |
| Sumo-Net [43] | 0.337 (✓) | 0.004 |
| DQS [50] | 0.084 (✓) | 0.008 |
| NeuralSurv (Ours) | 0.617 (✓) | **0.003** |

Table A8: Performance comparison of deep survival models on the ablation study with 250 observations, over five different train/test splits of each dataset (part 2). A checkmark (✓) indicates that the null hypothesis of perfect D-Calibration was not rejected at $\alpha = 0.05$ (model considered well-calibrated); a cross (×) indicates rejection of D-Calibration (model considered not well-calibrated). The best results for the KM-Calibration are shown in bold, and the second-best results are underlined. ↓ indicates lower is better.

## M.2.3  Comparison to Traditional Survival Models

| Method | COLON C-index ↑ | IPCW IBS ↓ | NWTCO C-index ↑ | IPCW IBS ↓ | GBSG C-index ↑ | IPCW IBS ↓ |
|---|---|---|---|---|---|---|
| CoxPH [10] | 0.669 | 0.192 | 0.710 | 0.136 | 0.694 | 0.171 |
| Weibull AFT [8] | 0.681 | 0.198 | 0.697 | 0.134 | 0.673 | 0.179 |
| RSF [23] | 0.590 | 0.210 | 0.604 | 0.156 | 0.588 | 0.193 |
| SSVM [40] | 0.654 | - | 0.734 | - | 0.695 | - |

| Method | METABRIC C-index ↑ | IPCW IBS ↓ | WHAS C-index ↑ | IPCW IBS ↓ | SUPPORT C-index ↑ | IPCW IBS ↓ |
|---|---|---|---|---|---|---|
| CoxPH [10] | 0.653 | 0.171 | 0.655 | 0.207 | 0.653 | 0.225 |
| Weibull AFT [8] | 0.658 | 0.172 | 0.622 | 0.224 | 0.650 | 0.239 |
| RSF [23] | 0.587 | 0.189 | 0.683 | 0.209 | 0.601 | 0.225 |
| SSVM [40] | 0.649 | - | 0.653 | - | 0.636 | - |

| Method | VLC C-index ↑ | IPCW IBS ↓ | SAC3 C-index ↑ | IPCW IBS ↓ |
|---|---|---|---|---|
| CoxPH [10] | 0.697 | 0.125 | 0.569 | 0.190 |
| Weibull AFT [8] | 0.690 | 0.127 | 0.607 | 0.287 |
| RSF [23] | 0.687 | 0.139 | 0.487 | 0.182 |
| SSVM [40] | 0.698 | - | 0.504 | - |

Table A9: Performance comparison of traditional survival models over five different train/test splits of each dataset. ↑ indicates higher is better; ↓ indicates lower is better. The SSVM method does not provide estimates of the survival function; the predicted ranks are used for the corresponding C-index evaluations while the IPCW-IBS metric cannot be computed.

| Method | COLON D-Calibration (p-value) | KM-Calibration ↓ | NWTCO D-Calibration (p-value) | KM-Calibration ↓ | GBSG D-Calibration (p-value) | KM-Calibration ↓ |
|---|---|---|---|---|---|---|
| CoxPH [10] | 0.913 (✓) | 0.006 | 0.979 (✓) | 0.003 | 0.950 (✓) | 0.004 |
| Weibull AFT [8] | 0.788 (✓) | 0.010 | 0.986 (✓) | 0.004 | 0.767 (✓) | 0.008 |
| RSF [23] | 0.791 (✓) | 0.009 | 0.999 (✓) | 0.002 | 0.854 (✓) | 0.003 |

| Method | METABRIC D-Calibration (p-value) | KM-Calibration ↓ | WHAS D-Calibration (p-value) | KM-Calibration ↓ | SUPPORT D-Calibration (p-value) | KM-Calibration ↓ |
|---|---|---|---|---|---|---|
| CoxPH [10] | 0.846 (✓) | 0.008 | 0.730 (✓) | 0.017 | 0.354 (✓) | 0.010 |
| Weibull AFT [8] | 0.759 (✓) | 0.006 | 0.650 (✓) | 0.018 | 0.420 (✓) | 0.012 |
| RSF [23] | 0.746 (✓) | 0.009 | 0.625 (✓) | 0.022 | 0.593 (✓) | 0.010 |

| Method | VLC D-Calibration (p-value) | KM-Calibration ↓ | SAC3 D-Calibration (p-value) | KM-Calibration ↓ |
|---|---|---|---|---|
| CoxPH [10] | 0.597 (✓) | 0.008 | 0.357 (✓) | 0.005 |
| Weibull AFT [8] | 0.759 (×) | 0.007 | 0.038 (✓) | 0.018 |
| RSF [23] | 0.414 (×) | 0.013 | 0.706 (✓) | 0.012 |

Table A10: Performance comparison of traditional survival models over five different train/test splits of each dataset (part 2). A checkmark (✓) indicates that the null hypothesis of perfect D-Calibration was not rejected at $\alpha = 0.05$ (model considered well-calibrated); a cross (×) indicates rejection of D-Calibration (model considered not well-calibrated). The best results for the KM-Calibration are shown in bold, and the second-best results are underlined. ↓ indicates lower is better.

## M.3 Prior Sensitivity Analysis

| Gamma Prior of $\phi$ | **COLON** | | | | | |
|---|---|---|---|---|---|---|
| | Gamma Posterior of $\phi$ | Posterior Median and 95% CI | C-index ↑ | IPCW IBS ↓ | D-Calibration (p-value) | KM-Calibration ↓ |
| $(\alpha_0 = 1, \beta_0 = 1)$ | $(\tilde{\alpha} = 60.877, \tilde{\beta} = 78.789)$ | 0.768 [0.591, 0.978] | **0.671** | **0.218** | 0.594 (✓) | **0.020** |
| $(\alpha_0 = 2, \beta_0 = 2)$ | $(\tilde{\alpha} = 45.865, \tilde{\beta} = 79.789)$ | 0.571 [0.421, 0.753] | 0.593 | 0.237 | 0.601 (✓) | 0.025 |
| $(\alpha_0 = 0.5, \beta_0 = 0.5)$ | $(\tilde{\alpha} = 38.333, \tilde{\beta} = 78.289)$ | 0.485 [0.347, 0.656] | 0.512 | 0.229 | 0.715 (✓) | 0.023 |

| Gamma Prior of $\phi$ | **METABRIC** | | | | | |
|---|---|---|---|---|---|---|
| | Gamma Posterior of $\phi$ | Posterior Median and 95% CI | C-index ↑ | IPCW IBS ↓ | D-Calibration (p-value) | KM-Calibration ↓ |
| $(\alpha_0 = 1, \beta_0 = 1)$ | $(\tilde{\alpha} = 48.406, \tilde{\beta} = 70.176)$ | 0.685 [0.509, 0.897] | **0.584** | 0.212 | 0.661 (✓) | 0.012 |
| $(\alpha_0 = 2, \beta_0 = 2)$ | $(\tilde{\alpha} = 48.034, \tilde{\beta} = 71.176)$ | 0.670 [0.498, 0.879] | 0.536 | 0.201 | 0.819 (✓) | **0.009** |
| $(\alpha_0 = 0.5, \beta_0 = 0.5)$ | $(\tilde{\alpha} = 47.347, \tilde{\beta} = 69.676)$ | 0.675 [0.500, 0.886] | 0.553 | **0.200** | 0.802 (✓) | **0.009** |

| Gamma Prior of $\phi$ | **GBSG** | | | | | |
|---|---|---|---|---|---|---|
| | Gamma Posterior of $\phi$ | Posterior Median and 95% CI | C-index ↑ | IPCW IBS ↓ | D-Calibration (p-value) | KM-Calibration ↓ |
| $(\alpha_0 = 1, \beta_0 = 1)$ | $(\tilde{\alpha} = 58.650, \tilde{\beta} = 85.462)$ | 0.682 [0.522, 0.873] | 0.657 | **0.188** | 0.735 (✓) | **0.009** |
| $(\alpha_0 = 2, \beta_0 = 2)$ | $(\tilde{\alpha} = 62.386, \tilde{\beta} = 86.462)$ | 0.718 [0.554, 0.911] | 0.602 | 0.195 | 0.808 (✓) | 0.010 |
| $(\alpha_0 = 0.5, \beta_0 = 0.5)$ | $(\tilde{\alpha} = 57.549, \tilde{\beta} = 84.962)$ | 0.673 [0.514, 0.863] | **0.665** | 0.189 | 0.772 (✓) | 0.010 |

Table A11: Prior sensitivity analysis on $\phi$ using priors with double and half the original variance..

# N Proofs

## N.1 Proof of Theorem 3.1

Before proving Theorem 3.1 we must show some intermediate results.

**Lemma N.1.** *Assume that for each $i = 1, \ldots, N$ the function $g(\cdot, \mathbf{x}_i; \cdot) \in C([0, y_i] \times \mathbb{R}^m)$. Then, it follows that*

$$\int_0^{y_i} \lambda_0(t, \mathbf{x}_i; \phi) \mathrm{d}t < \infty$$

*for every $i = 1, \ldots, N$.*

*Proof.* Fix an arbitrary index $i \in \{1, \ldots, N\}$. From Section 2.3, recall that $p_{\boldsymbol{\theta}}(\boldsymbol{\theta})$ is the probability density function of a multivariate normal distribution with zero mean and identity covariance matrix $\mathbf{I}_m$. Our goal is to show that the normalization factor $Z(t, \mathbf{x}_i)$ admits a strictly positive lower bound on $[0, y_i]$, from which the integrability of $\lambda_0(t, \mathbf{x}_i; \phi)$ will follow.

**Step 1: Continuity of $Z(t, \mathbf{x}_i)$ on $[0, y_i]$.** Fix any $t_0 \in [0, y_i]$, and let $(t_n)_{n \geq 1}$ be a sequence in $[0, y_i]$ such that $t_n \to t_0$ as $n \to \infty$. Define, for each $n$, the functions

$$h_n(\boldsymbol{\theta}) := \sigma(g(t_n, \mathbf{x}_i; \boldsymbol{\theta})) p_{\boldsymbol{\theta}}(\boldsymbol{\theta}), \quad n \geq 1,$$
$$h(\boldsymbol{\theta}) := \sigma(g(t_0, \mathbf{x}_i; \boldsymbol{\theta})) p_{\boldsymbol{\theta}}(\boldsymbol{\theta}).$$

Since $g(\cdot, \mathbf{x}_i; \cdot) \in C([0, y_i] \times \mathbb{R}^m)$ and the sigmoid $\sigma(\cdot)$ is a continuous function, it follows that

$$\lim_{n \to \infty} h_n(\boldsymbol{\theta}) = h(\boldsymbol{\theta})$$

pointwise for all $\boldsymbol{\theta} \in \mathbb{R}^m$. Furthermore, observe that

$$|h_n(\boldsymbol{\theta})| \leq p_{\boldsymbol{\theta}}(\boldsymbol{\theta})$$

since $0 < \sigma(\cdot) < 1$. Because $p_{\boldsymbol{\theta}}(\boldsymbol{\theta})$ integrates to 1 over $\mathbb{R}^m$, we may apply the Dominated Convergence Theorem (DCT) to conclude that:

$$\lim_{n \to \infty} Z(t_n, \mathbf{x}_i) = \lim_{n \to \infty} \int_{\mathbb{R}^m} h_n(\boldsymbol{\theta}) \mathrm{d}\boldsymbol{\theta} \overset{\mathrm{DCT}}{=} \int_{\mathbb{R}^m} h(\boldsymbol{\theta}) \mathrm{d}\boldsymbol{\theta} = Z(t_0, \mathbf{x}_i).$$

Since $t_0$ was arbitrary in $[0, y_i]$, $Z$ is continuous everywhere on that interval.

**Step 2: Strict positivity of $Z(t, \mathbf{x}_i)$ on $[0, y_i]$.** For each fixed $t \in [0, y_i]$, since $\sigma(g(t, \mathbf{x}_i; \boldsymbol{\theta})) > 0$ and $p_{\boldsymbol{\theta}}(\boldsymbol{\theta}) > 0$ for all $\boldsymbol{\theta} \in \mathbb{R}^m$, we have:

$$Z(t, \mathbf{x}_i) = \int_{\mathbb{R}^m} \sigma(g(t, \mathbf{x}_i; \boldsymbol{\theta})) p_{\boldsymbol{\theta}}(\boldsymbol{\theta}) \mathrm{d}\boldsymbol{\theta} > 0.$$

Since $Z(t, \mathbf{x}_i)$ is a continuous and strictly positive function on the compact interval $[0, y_i]$, the Weierstrass Extreme Value Theorem ensures that $Z$ attains a minimum on this interval. Define:

$$z^* = \min_{t \in [0, y_i]} Z(t, \mathbf{x}_i) > 0$$

**Step 3: Integrability of $\lambda_0(t, \mathbf{x}_i; \phi)$.** Note that for all $t \in [0, y_i]$, we have

$$\lambda_0(t, \mathbf{x}_i; \phi) = \frac{\lambda_0(t; \phi)}{Z(t, \mathbf{x}_i)} \leq \frac{\lambda_0(t; \phi)}{z^*}.$$

It is straightforward to verify that $\lambda_0(t; \phi)$ is integrable on $[0, y_i]$, therefore it follows that

$$\int_0^{y_i} \lambda_0(t, \mathbf{x}_i; \phi) \mathrm{d}t \leq \frac{1}{z^*} \int_0^{y_i} \lambda_0(t; \phi) \mathrm{d}t < \infty.$$

This completes the proof. $\qquad\square$

Our next result verifies a condition needed for applying Campbell's Theorem in the proof of Theorem 3.1. To establish this, we will use the following Pólya–Gamma identity:

$$\mathbb{E}_{\omega \sim p_{\mathrm{PG}}(\omega|1,0)}[\omega] = \frac{1}{4}, \tag{A32}$$

which follows by taking the limit $c \to 0$ in equation (A4). Alternatively, to prove (A32), one can start from the representation in equation (A1), apply Tonelli's theorem to interchange expectation and infinite summation, and then invoke the series identity

$$\sum_{k=1}^{\infty} \frac{1}{(k - \frac{1}{2})^2} = \frac{\pi^2}{2}.$$

We are now ready to present our next result.

**Lemma N.2.** *Assume that for each $i = 1, \ldots, N$ the function $g(\cdot, \mathbf{x}_i; \cdot) \in C([0, y_i] \times \mathbb{R}^m)$. Then, with probability 1 the sum*

$$H(\Psi_i) = \sum_{(t,\omega)_j \in \Psi_i} f(\omega_j, -g(t_j, \mathbf{x}_i; \boldsymbol{\theta}))$$

*is absolutely convergent for every $i = 1, \ldots, N$.*

*Proof.* Fix an arbitrary index $i \in \{1, \ldots, N\}$. Recall the definition of $f(\omega, z)$ from (8). From Theorem C.3, it suffices to show

$$\int_0^{y_i} \int_0^{\infty} \min(|f(\omega, -g(t, \mathbf{x}_i; \boldsymbol{\theta}))|, 1) \lambda_i(t, \omega; \phi) \mathrm{d}\omega \mathrm{d}t < \infty. \tag{A33}$$

Since $\omega \in \mathbb{R}_+$, then it follows from the triangle inequality that

$$\min(|f(\omega, -g(t, \mathbf{x}_i; \boldsymbol{\theta}))|, 1) \leq |f(\omega, -g(t, \mathbf{x}_i; \boldsymbol{\theta}))|$$
$$\leq \frac{|g(t, \mathbf{x}_i; \boldsymbol{\theta})|}{2} + \frac{g(t, \mathbf{x}_i; \boldsymbol{\theta})^2}{2} \omega + \log(2).$$

Hence it remains to prove finiteness of three integrals:

$$\mathcal{I}_1 := \int_0^{y_i} \int_0^{\infty} \frac{|g(t, \mathbf{x}_i; \boldsymbol{\theta})|}{2} \lambda_i(t, \omega; \phi) \mathrm{d}t \mathrm{d}\omega,$$

$$\mathcal{I}_2 := \int_0^{y_i} \int_0^{\infty} \frac{g(t, \mathbf{x}_i; \boldsymbol{\theta})^2}{2} \omega \lambda_i(t, \omega; \phi) \mathrm{d}\omega \mathrm{d}t,$$

$$\mathcal{I}_3 := \log(2) \int_0^{y_i} \int_0^{\infty} \lambda_i(t, \omega; \phi) \mathrm{d}\omega \mathrm{d}t.$$

$\mathcal{I}_1$ **is finite.** Since $g(t, \mathbf{x}_i; \boldsymbol{\theta})$ is continuous on the compact interval $[0, y_i]$, it is bounded by some $M > 0$. Then,

$$\mathcal{I}_1 = \left( \int_0^{\infty} p_{\mathrm{PG}}(\omega|1, 0) \mathrm{d}\omega \right) \int_0^{y_i} \frac{|g(t, \mathbf{x}_i; \boldsymbol{\theta})|}{2} \lambda_0(t, \mathbf{x}_i; \phi) \mathrm{d}t \leq M \int_0^{y_i} \lambda_0(t, \mathbf{x}_i; \phi) \mathrm{d}t < \infty,$$

where the last inequality is Lemma N.1.

$\mathcal{I}_2$ **is finite.** Likewise $g(t, \mathbf{x}_i; \boldsymbol{\theta})^2$ is bounded by some $C > 0$ over $[0, y_i]$ and $\mathbb{E}_{\omega \sim p_{\mathrm{PG}}(\omega|1,0)}[\omega] = \frac{1}{4}$ (see (A32)), so

$$\mathcal{I}_2 = \left( \mathbb{E}_{\omega \sim p_{\mathrm{PG}}(\omega|1,0)}[\omega] \right) \int_0^{y_i} \frac{g(t, \mathbf{x}_i; \boldsymbol{\theta})^2}{2} \lambda_0(t, \mathbf{x}_i; \phi) \mathrm{d}t \leq \frac{C}{8} \left( \int_0^{y_i} \lambda_0(t, \mathbf{x}_i; \phi) \mathrm{d}t \right) < \infty,$$

where the last inequality is Lemma N.1.

$\mathcal{I}_3$ **is finite.** Finally,

$$\mathcal{I}_3 = \log(2) \int_0^{y_i} \lambda_0(t, \mathbf{x}_i; \phi)\mathrm{d}t < \infty,$$

again by Lemma N.1.

Since $\mathcal{I}_1, \mathcal{I}_2, \mathcal{I}_3$, are all finite, the condition in (A33) is satisfied and the sum $H(\Psi_i)$ converges absolutely with probability 1. $\qquad\square$

The next result presents an integral identity which is key to proving the data augmentation scheme of Theorem 3.1.

**Lemma N.3.** *Assume that for each $i = 1, \ldots, N$ the function $g(\cdot, \mathbf{x}_i; \cdot) \in C([0, y_i] \times \mathbb{R}^m)$. Then the double integral*

$$\int_0^{y_i} \int_0^{\infty} \left(1 - e^{f(\omega, -g(t, \mathbf{x}_i; \boldsymbol{\theta}))}\right) p_{PG}(\omega|1, 0)\lambda_0(t, \mathbf{x}_i; \phi)\mathrm{d}\omega\mathrm{d}t$$

*converges, and in fact*

$$\int_0^{y_i} \int_0^{\infty} \left(1 - e^{f(\omega, -g(t, \mathbf{x}_i; \boldsymbol{\theta}))}\right) p_{PG}(\omega|1, 0)\lambda_0(t, \mathbf{x}_i; \phi)\mathrm{d}\omega\mathrm{d}t =$$

$$\int_0^{y_i} \lambda_0(t, \mathbf{x}_i; \phi)\, \sigma(g(t, \mathbf{x}_i; \boldsymbol{\theta}))\mathrm{d}t \quad \text{(A34)}$$

*for every $i = 1, \ldots, N$.*

*Proof.* Fix an arbitrary index $i \in \{1, \ldots, N\}$. By Lemma N.1

$$\int_0^{y_i} \lambda_0(t, \mathbf{x}_i; \phi)\mathrm{d}t < \infty.$$

Since $0 < \sigma(\cdot) < 1$, we have

$$0 \le \lambda_0(t, \mathbf{x}_i; \phi)\sigma(g(t, \mathbf{x}_i; \boldsymbol{\theta})) < \lambda_0(t, \mathbf{x}_i; \phi)$$

and therefore

$$\int_0^{y_i} \lambda_0(t, \mathbf{x}_i; \phi)\sigma(g(t, \mathbf{x}_i; \boldsymbol{\theta}))\mathrm{d}t < \infty. \qquad\qquad \text{(A35)}$$

This shows the finiteness of the right-hand side of (A34). By combining $\sigma(z) = 1 - \sigma(-z)$ with (A5) we obtain that

$$\int_0^{y_i} \lambda_0(t, \mathbf{x}_i; \phi)\, \sigma(g(t, \mathbf{x}_i; \boldsymbol{\theta}))\mathrm{d}t =$$

$$\int_0^{y_i} \int_0^{\infty} \left(1 - e^{f(\omega, -g(t, \mathbf{x}_i; \boldsymbol{\theta}))}\right) p_{PG}(\omega|1, 0)\lambda_0(t, \mathbf{x}_i; \phi)\mathrm{d}\omega\mathrm{d}t. \quad \text{(A36)}$$

Putting together the finiteness from (A35) with the equality of (A36) completes the proof. $\qquad\square$

We are now ready to prove Theorem 3.1.

*Proof of Theorem 3.1.* Fix an arbitrary index $i \in \{1, \ldots, N\}$. The joint expectation factors into two independent pieces:

1. **Expectation over $\omega_i$:** This term recovers $\lambda_0(y_i, \mathbf{x}_i; \phi)^{\delta_i}\sigma(g(y_i, \mathbf{x}_i; \boldsymbol{\theta}))^{\delta_i}$;

2. **Expectation over $\Psi_i$:** This term recovers $\exp\left(-\int_0^{y_i} \lambda_0(t, \mathbf{x}_i; \phi)\sigma(g(t, \mathbf{x}_i; \boldsymbol{\theta}))\mathrm{d}t\right)$.

**Step (1): Expectation over $\omega_i$.** Since $\delta_i \in \{0, 1\}$,

$$\left(e^{f(\omega_i, g(y_i, \mathbf{x}_i; \boldsymbol{\theta}))}\right)^{\delta_i} = \begin{cases} e^{f(\omega_i, g(y_i, \mathbf{x}_i; \boldsymbol{\theta}))}, & \delta_i = 1, \\ 1, & \delta_i = 0. \end{cases}$$

Hence,

$$\int_0^\infty \left(e^{f(\omega_i, g(y_i, \mathbf{x}_i; \boldsymbol{\theta}))}\right)^{\delta_i} p_{\mathrm{PG}}(\omega_i | 1, 0) \mathrm{d}\omega_i = \left(\int_0^\infty e^{f(\omega_i, g(y_i, \mathbf{x}_i; \boldsymbol{\theta}))} p_{\mathrm{PG}}(\omega_i | 1, 0) \mathrm{d}\omega_i\right)^{\delta_i}.$$

By the Pólya–Gamma identity (Eq. (A5)), the bracketed integral equals $\sigma(g(y_i, \mathbf{x}_i; \boldsymbol{\theta}))$. Multiplying by $\lambda_0(y_i, \mathbf{x}_i; \phi)^{\delta_i}$ gives exactly

$$\lambda_0(y_i, \mathbf{x}_i; \phi)^{\delta_i} \sigma(g(y_i, \mathbf{x}_i; \boldsymbol{\theta}))^{\delta_i}.$$

**Step (2): Expectation over $\Psi_i$.** By Lemma N.2 the random sum

$$H(\Psi_i) = \sum_{(t, \omega)_j \in \Psi_i} f(\omega_j, -g(t_j, \mathbf{x}_i; \boldsymbol{\theta}))$$

is absolutely convergent with probability 1, and by Lemma N.3 the corresponding integral converges. Therefore, we may apply Campbell's Theorem (Theorem C.3) together with the PG-sigmoid identity from (A34) to conclude

$$\mathbb{E}_{\Psi_i \sim \mathbb{P}_{\Psi_i | \phi}} \left[\prod_{(t, \omega)_j \in \Psi_i} e^{f(\omega_j, -g(t_j, \mathbf{x}_i; \boldsymbol{\theta}))}\right] = \exp\left(-\int_0^{y_i} \lambda_0(t, \mathbf{x}_i; \phi) \sigma(g(t, \mathbf{x}_i; \boldsymbol{\theta})) \mathrm{d}t\right).$$

Putting Steps (1) and (2) together reproduces precisely the two factors of the original likelihood $p(y_i, \delta_i | \mathbf{x}_i, \phi, g)$. This completes the proof. $\qquad \square$

