# OpenReview forum: "NeuralSurv: Deep Survival Analysis with Bayesian Uncertainty Quantification"
_NeurIPS.cc/2025/Conference — NeurIPS 2025 poster_

### Official Review · Reviewer_Pg1A · 2025-07-02

**Clarity:** 2
**Significance:** 4
**Originality:** 2
**Rating:** 4
**Confidence:** 3

**Summary:**

This work introduces NeuralSurv, a deep survival analysis framework with Bayesian uncertainty quantification. To do so, it proposes a two-stage data augmentation strategy using Pólya-Gamma variables and marked Poisson processes to enable exact continuous-time inference. A local linearization technique makes Bayesian neural networks tractable via closed-form coordinate-ascent updates. Experimental results highlight the performance of the method in discrimination and calibration in data-scarce regimes.

**Questions:**

1. Could the authors more clearly clarify which elements of the work are their own novel contribution versus drawn from other insights in the field of Bayesian inference, as well as adding a related work section providing background Bayesian inference and survival analysis methods? [I would raise my clarity score from "2: fair" to "3: good"; depending on the result of this discussion I may raise my Originality score from "2: fair" to "3: good"]
2. Could the authors implement the following experiments, as discussed in Weaknesses? [I would raise my overall Rating from "4: Borderline accept" to "5: Accept" with these additional evaluations]
    * Demonstrate the effect of different priors on concordance / IBS.
    * Highlight survival curve uncertainty bounds in data-rich vs. data-scarce settings.
    * Comparison against baseline methods of computing uncertainty intervals on survival curves.
    * Include additional experimental results that use the whole dataset (rather than the subsampled component).
3. Could the authors include confidence intervals on their evaluation statistics (c-index, Brier score), along with appropriate significance testing and multiple-comparisons correction? [this is another element that would contribute to me raising my overall Rating to a 5]

**Ethical Concerns:**

["NO or VERY MINOR ethics concerns only"]

**Limitations:**

Yes

**Quality:**

3

**Strengths And Weaknesses:**

I congratulate the authors on their work, and I am grateful for the opportunity to have reviewed this paper. The paper is well-written and a pleasure to read.

Strengths:
* The work is **very timely**: it takes steps to tackle an important problem in the in the survival analysis literature that I've noticed over recent years. As we obtain the means to statistically learn ever-more-granular survival curves (e.g., via expressive methods like SODEN [1] or mixture density networks [2]), there may or may not be support in the data for the granularity of survival curves that are being learned. As such, developing rigorous Bayesian mechanisms to credibly imbue individual survival curve estimates is an important problem. This work represents strong first steps in this direction -- e.g., the confidence intervals in Figure 1 are _exactly_ what a lot of practitioners are looking for in applied settings.
* The experiments, as presented, are strong -- the method seems to outperform standard baselines on both concordance and calibration. I recommend the authors highlight these results not as evidence of utility (discussed more in Weaknesses), but rather as evidence that the model's design decisions (e.g., Weibull-like baseline hazard; isotropic Gaussian prior on $\theta$; etc.) are not too restrictive for practical settings.
* The NeuralSurv framework seems fast and scalable. By performing local linearization around the MAP and mean-field variational inference, the authors implement a closed-form coordinate ascent update with linear time complexity in model size.

Weaknesses:
* Related work and novelty: I admit that I do not have the deepest knowledge of Bayesian inference. With this in mind, the paper makes it rather difficult for me to tease apart which elements of NeuralSurv represent novel contributions, and which represent established methods in Bayesian inference. I would appreciate some clarity on this. My understanding is that the Polya-Gamma scheme in 3.1 is an established contribution; the Poisson Process Augmentation Scheme in 3.2 is novel; the Variational Mean-Field Approximation in 4.1 is an established method that is novel when applied to the survival setting; the local linearization of the BNN in 4.2 is established; and CAVI in 4.3 is established.
  * I would appreciate if the authors added a Related Work section so that elements like this can be discussed, and so that their framework can be placed into conversation with other survival analysis frameworks.
* There is a degree of misalignment between the experiments and the proposed contribution of the work. Whereas the experiments highlight concordance and calibration (via integrated Brier score), my understanding is that the core contribution of the work is not to implement the best ranking metric (or calibrated survival curve), but rather to (a) support the integration of prior knowledge into estimation, and (b) permit easy computation of a credible confidence interval. With that in mind, I would prefer that the authors include three additional experiments of the following form and to reframe the discussion of contributions accordingly:
  1. Demonstrate how inference under different priors influences model performance as evaluated using classical statistics (c-index, IBS). This would inform the reader of the extent to which prior selection influences model performance on held-out data, as well as providing some guidance as to what may make for a "good" vs. "bad" prior in the survival setting.
  2. Show several examples of survival curve estimation, both from the data-rich regime of the distribution (e.g., a patient who is similar to many other patients), and from the data-scarce regime (e.g., a patient who is dissimilar to many other patients). The aim of this experiment would be to demonstrate how the uncertainty in survival curves varies as a function of distributional support.
  3. How do the credible intervals using this Bayesian setting compare to those computed via other methods? The authors may know the baselines here better than I do, but I'd imagine some baseline of bootstrapped confidence intervals and Monte Carlo dropout [3] would be important points of comparison.
* I understand that the authors chose their experimental setup to highlight the performance of NeuralSurv in data-scarce regimes; but would it be possible to include additional experimental results on the performance of each method using the entire dataset (not subsampled to 100 train / 25 validation samples)? It would be good to get a sense of how the method performs as the dataset size increases. This would also allow for a more standardized comparison against the baseline methods, which are often evaluated on the entirety of COLON, METABRIC, and GBSG.
4. The authors claim that their method supports "time-varying covariate–risk relationships" (Line 300); however, I don't see anything in their experimental framework supporting this. I see that $X=\{x_i \,:\, i=1, ..., N\}$ (Line 56); if the covariates were time-varying (e.g., as in [4]) I would have expected a different setup (e.g., $X=\{x_i(t) \,:\, i=1, ..., N, t=0, ..., T\}$). Could this be clarified?


[1] Tang, Weijing, Jiaqi Ma, Qiaozhu Mei, and Ji Zhu. "Soden: A scalable continuous-time survival model through ordinary differential equation networks." Journal of Machine Learning Research 23, no. 34 (2022): 1-29.

[2] Han, Xintian, Mark Goldstein, and Rajesh Ranganath. "Survival mixture density networks." In Machine Learning for Healthcare Conference, pp. 224-248. PMLR, 2022.

[3] Gal, Yarin, and Zoubin Ghahramani. "Dropout as a bayesian approximation: Representing model uncertainty in deep learning." In international conference on machine learning, pp. 1050-1059. PMLR, 2016.

[4] Lee, Changhee, Jinsung Yoon, and Mihaela Van Der Schaar. "Dynamic-deephit: A deep learning approach for dynamic survival analysis with competing risks based on longitudinal data." IEEE Transactions on Biomedical Engineering 67, no. 1 (2019): 122-133.

---

> ### Author Rebuttal · Authors · 2025-07-31
>
> > Q1: the paper makes it rather difficult for me to tease apart which elements of NeuralSurv represent novel contributions, and which represent established methods in Bayesian inference. I would appreciate some clarity on this.
>
> We thank the reviewer for highlighting the need to distinguish our novel contributions from established methods. All components of our algorithm have precedents in the literature: the Pólya–Gamma augmentation scheme was first introduced by Polson [A], and its combination with Poisson process augmentation appears in  [9,44]. The variational mean‑field approximation and CAVI are well‑known techniques that have previously been applied in survival settings (e.g., [B]). The local linearization method for BNNs, which yields a Gaussian process–based approximation, is also a recent but established approach [18]. Our key innovation lies in unifying these components into a single inference framework that fully exploits Gaussian process conjugacy alongside Poisson–Pólya–Gamma augmentation. To clarify this in the manuscript, we have added the following text in Section 6:
>
> “Pólya–Gamma and Poisson data-augmentation schemes (Section 3.3)  have been extensively employed with standard Gaussian process models [9,44]. Likewise, the local linearization of Bayesian neural networks, which yields a Gaussian process–based approximation, (Section 4.2), is a well established technique [18]. To our knowledge, this work is the first to integrate these two approaches into a unified framework that capitalizes on Gaussian process conjugacy. By combining these methods, we contribute a novel inference strategy at the intersection of Bayesian deep learning and Gaussian process modeling. ”
>
> [A] Polson, N. G., Scott, J. G., & Windle, J. (2013). Bayesian Inference for Logistic Models Using Pólya–Gamma Latent Variables. Journal of the American Statistical Association, 108(504), 1339–1349.
>
> [B] Michael Komodromos, Eric O Aboagye, Marina Evangelou, Sarah Filippi, Kolyan Ray, Variational Bayes for high-dimensional proportional hazards models with applications within gene expression, Bioinformatics, Volume 38, Issue 16, August 2022, Pages 3918–3926.
>
> > Q2: Demonstrate how inference under different priors influences model performance as evaluated using classical statistics (c-index, IBS).
>
> We thank the reviewer for their suggestion.
>
> a/ Prior on theta: NeuralSurv’s inference algorithm is built on the local linearization approach for Bayesian neural networks [18], which imposes an isotropic unit‐variance Gaussian prior on all network weights, this modeling choice is fixed and not subject to alternative specifications.
>
> b/ Prior on phi: We conducted a prior sensitivity analysis by varying the hyperparameters of the Gamma prior over the precision parameter $\phi$, where $\phi \sim \text{Gamma}(\alpha_0, \beta_0)$. The main results reported in the paper use the default setting $\alpha_0 = \beta_0 = 1$, as specified in Appendix K. To assess sensitivity, we explored two additional configurations: (i) Reduced prior variance: $\alpha_0 = \beta_0 = 2$, (ii) Increased prior variance: $\alpha_0 = \beta_0 = 0.5$. These correspond to priors with half and double the prior variance, respectively, while maintaining the same mean, reflecting a non-informative prior structure.
>
> We ran this analysis on all datasets. While the posterior distributions under different priors generally overlapped, their central tendencies varied, for example in the GBSG dataset, indicating mild to moderate sensitivity to prior choice. In contrast, the METABRIC results were nearly identical across priors, suggesting robustness in that setting. Overall, these findings suggest that while prior choice can influence results in certain datasets. When prior information is available, it should always be used to guide the model toward a principled balance between flexibility and regularization.
>
> For each dataset, we will include in the Appendix tables showing the evaluation metrics, as well as the posterior median and 95% credible intervals.
>
> We reported below the evaluation metrics, as well as the posterior median and 95% credible intervals for phi (only for METABRIC and GBSG datasets for space).
>
> METABRIC dataset
> Prior| Posterior | C-index | IPCW IBS |
> |-|-|-|-|
> Gamma(1, 1) | 0.685 [0.509, 0.897]| 0.584 | 0.212 |
> Gamma(2, 2) | 0.670 [0.498, 0.879] | 0.536 | 0.201 |
> Gamma(0.5, 0.5) | 0.675 [0.500, 0.886] | 0.553 | 0.200 |
>
> GBSG dataset
> Prior| Posterior | C-index | IPCW IBS |
> |-|-|-|-|
> Gamma(1, 1) | 0.682 [0.522, 0.873] | 0.657  | 0.188 |
> Gamma(2, 2) | 0.718 [0.554, 0.911] |  0.602 | 0.195 |
> Gamma(0.5, 0.5) | 0.673 [0.514, 0.863] | 0.665 | 0.189 |
>
> > Q3: Show several examples of survival curve estimation, both from the data-rich regime of the distribution (e.g., a patient who is similar to many other patients), and from the data-scarce regime (e.g., a patient who is dissimilar to many other patients).
>
> To identify similar and dissimilar patients, we computed the average pairwise distance of each patient's covariates to all other observations in the dataset. Patients with smaller average distances were those with the larger distributional support (in a data-rich regime), while those with larger distances were considered those with the smaller distributional support (in a data-scarce regime). We focused on two representative individuals: Patient A, who had the largest distributional support across the dataset, is the most similar and Patient B, who had the smallest distributional support across the dataset, is the most dissimilar.
>
> As expected in a well-calibrated Bayesian framework, we observe consistently greater uncertainty in the data-scarce regime, reflecting limited covariate support for dissimilar patients.
>
> For each dataset, we will include in the Appendix plots showing the posterior survival functions (median and 95% credible intervals) of these two patients side by side.
>
> To summarize the results here, we report below the median and 95% interquartile range (IQR, defined as the difference between the 97.5th and 2.5th percentiles) of the posterior survival probabilities at four key timepoints: early (25% of the observational period), midpoint (50%), late (75%), and final (end of follow-up). Additionally, we report the mean IQR across the entire survival curve in the last column. We reported the results only for METABRIC and GBSG datasets for space.
>
> | GBSG dataset | Early | Mid-point | Late | Final | Mean of IQR |
> |-|-|-|-|-|-|
> Most dissimilar | 0.77 (0.14) |  0.62 (0.22)| 0.58 (0.24) | 0.56 (0.27) | 0.18 |
> Most similar | 0.70 (0.13) |  0.49 (0.19) | 0.34 (0.20) | 0.26 (0.20)| 0.15|
>
> | METABRIC dataset | Early | Mid-point | Late | Final | Mean of IQR |
> |-|-|-|-|-|-|
> Most dissimilar |  0.72 (0.17) |  0.53 (0.26) | 0.40 (0.29) |  0.32 (0.30)|  0.21|
> Most similar | 0.70 (0.15) | 0.49 (0.21) | 0.34 (0.23) | 0.26 (0.21)  | 0.17 |
>
> >  Q4: How do the credible intervals using this Bayesian setting compare to those computed via other methods?
>
> We thank the reviewer for this insightful suggestion. We agree that directly comparing our Bayesian credible intervals with alternative uncertainty estimates, such as those obtained via Monte Carlo dropout [3], would meaningfully enhance our discussion of uncertainty quantification. Unfortunately, we were not able to include these experiments in the current submission due to limited computational resources (we prioritized the survival‐curve estimation experiments described above, as well as the runtime analyses presented below).
> Nonetheless, we believe a rigorous comparison remains valuable. Such an analysis would involve implementing Monte Carlo dropout versions of each deep survival benchmark model, then evaluating and contrasting their uncertainty estimates with our credible intervals across both synthetic and real‐world datasets. However, carrying out this comprehensive study falls outside the present paper’s scope and exceeds the time available for revisions.
>
> > Q5:  include additional experimental results on the performance of each method using the entire dataset
>
> We thank the reviewer for their thoughtful comment. We agree that evaluating model performance on the full datasets would provide a more standardized basis for comparison and insight into scalability. However, at this stage, our method is not yet scalable to full dataset sizes due to the computational demands of our fully Bayesian modeling approach. As noted in our response to Q2 to Reviewer 52KX, the running time becomes prohibitive as sample size increases, for reference, COLON (N = 1822), METABRIC (N = 1904), and GBSG (N = 2232) currently exceed our feasible limits.
> To partially address this concern, we conducted an additional experiment on the full SUPPORT dataset in response to Reviewer QbSK’s Question 11. This experiment included comparisons with published benchmarks and demonstrated that our inference pipeline yields comparable C-index values, thereby providing some evidence for the validity.
>
> > Q6: The authors claim that their method supports "time-varying covariate–risk relationships" (Line 300); however, I don't see anything in their experimental framework supporting this. I see that  (Line 56); if the covariates were time-varying (e.g., as in [4]) I would have expected a different setup (e.g., ). Could this be clarified?
>
> We thank the reviewer for pointing out the need for clarification. In our manuscript, the phrase “time‑varying covariate–risk relationships” was intended to mean “time‑varying relationships between covariates and risk,” not “relationships involving time‑varying covariates.” To address this, we have revised the conclusion to read:
>  “We propose the first fully Bayesian framework for deep survival analysis that models time-varying relationships between covariates and risk.”.

---

### Official Review · Reviewer_52KX · 2025-07-06

**Clarity:** 4
**Significance:** 3
**Originality:** 3
**Rating:** 5
**Confidence:** 3

**Summary:**

The paper introduces NeuralSurv, a deep survival model that offers an architecture-agnostic framework for continuous-time survival analysis that can capture complex time-varying covariate-risk relationships. The method includes a two-stage data-augmentation scheme utilizing latent marked Poisson processes and Pólya-Gamma variables. The BNN s approximated with local linearization and trained with a variational mean-field objective. The authors demonstrate good calibration and discriminative performance on synthetic and real-world datasets, particularly in data-scarce scenarios.

**Questions:**

- question about hyperparameters and initializations as elaborated in the weakness section.

- is it possible to have a table to compare runtime with baseline methods as well? i understand that runtime is already discussed in discussion and appendix K, but quantitative comparison will be informative to researchers trying to using your method. (from my understanding NeuralSurv is already quite fast.)

**Ethical Concerns:**

["NO or VERY MINOR ethics concerns only"]

**Final Justification:**

The authors have answered my questions and I maintain my positive recommendation.

**Limitations:**

yes.

**Quality:**

4

**Strengths And Weaknesses:**

Strengths

- The contribution is original and significant. this paper provides a first framework for neural network-based survival analysis, with very reasonable choices and construction that are well-placed in literature. This allows following works to study survival analysis leveraging NN's expressivity to model complex time-varying covariate-risk relationships, while having relatively reliable uncertainty quantification.

- Strong empirical performance on the synthetic and real world datasets.

- overall very well-written paper - all the sections are clear and coherent, and the method is well-explained in a step-by-step manner. although I am not an expert on variational inference, I am still able to follow the methodology and the math in the main text. I appreciate the primers in the appendix as well.

Weaknesses

- Lack of discussion on sensitivity to hyperparameters and initialization. the method relies on multiple steps of data augmentation and approximation for VI. my experience with BNNs is that that they are brittle to hyperparameter choices and initializations. From the experiments section of the paper, it seems like the experiments were performed once with a single architecture. No error bars on metrics from multiple runs (which i would expect from statisticians? maybe i'm wrong). It would be great to have a discussion on the robustness of the method.

---

> ### Author Rebuttal · Authors · 2025-07-30
>
> > Q1: question about hyperparameters and initializations as elaborated in the weakness section
>
> We thank the reviewer for their question.
>
> - Initialization: As described in Appendix K, in the EM algorithm, the parameters are initialized so that they match their prior expected values. Specifically, we set $\boldsymbol{\theta}^{(0)} = \mathbf{0}$ and $\phi^{(0)} = \alpha_0 / \beta_0$. In the CAVI algorithm, the hyperparameters are initialized so that the expected values of the model parameters match the MAP estimates. Specifically, we set $\tilde{\alpha}^{(0)} = \phi_{\text{MAP}} \times \tilde{\beta}, \quad \text{and} \quad (\tilde{\boldsymbol{\mu}}, \tilde{\boldsymbol{\Sigma}})^{(0)} = (\boldsymbol{\theta}_{\text{MAP}}, \mathbf{I}_m).$
>
> - Architecture: We chose not to optimize NeuralSurv (via hyperparameter searches) to every data set as we felt this would be an unfair representation of performance in comparison to the benchmarks.
>
> - Error bars: While we chose not to include the metrics variances in the paper in order to align with common practice in the survival analysis literature [24, 25, 32], where only point estimates are typically reported, we are happy to include them if the reviewer believes it would enhance clarity or interpretability.
>
> - Prior sensitivity: We have conducted a prior sensitivity analysis for the hyperparameters of the prior on phi. It is presented in Question 2 of reviewer Pg1A.
>
> > Q2: is it possible to have a table to compare runtime with baseline methods as well? i understand that runtime is already discussed in discussion and appendix K, but quantitative comparison will be informative to researchers trying to using your method. (from my understanding NeuralSurv is already quite fast.)
>
> We thank the reviewer for this helpful suggestion.
> In response, we have added a runtime analysis table reporting the inference times for NeuralSurv and several benchmark methods in the Appendix. For this experiment, we used the Colon dataset and explored two axes of variability:
>
> 1/ Sample size: $N \in \{25, 125, 250\}$
>
> 2/ Neural network architecture:
> - MLP with 1 hidden layer (1L)
> - MLP with 2 hidden layers and 6 units (2L-6U)
> - MLP with 2 hidden layers 16 units (2L-16U)
>
> A summary of the results (runtime in minutes) is provided below for NeuralSurv, the fastest (DQS) and the slowest (DeepHit) benchmark methods. Please note that DQS is a new benchmark method required by reviewer  QbSK.
> As expected, we observe that runtime increases with both the number of samples and the complexity of the model architecture. Notably, NeuralSurv, being a Bayesian method, is slower than frequentist baselines. This is consistent with prior literature, as Bayesian inference, particularly when combined with neural networks, is computationally more intensive. As illustrated by our experiments, this computational cost currently limits the scalability of NeuralSurv to very large datasets or highly complex models without additional approximation techniques. We hope this additional analysis clarifies the runtime characteristics of our method and provides a more complete picture of its computational trade-offs.
>
>  | Method | N=25 ||| N=125 ||| N=250 |||
>    |--------|------|------|------|------|------|------|------|------|------|
>    |        | 1L   | 2L-6U | 2L-16U | 1L | 2L-6U | 2L-16U | 1L | 2L-6U | 2L-16U |
> | DeepHit | 0.164  | 0.139 | 0.140 | 0.120 | 0.126 | 0.142 | 0.122  | 0.145  | 0.135  |
> | DQS | 0.019 | 0.020 | 0.024 | 0.023 | 0.025 | 0.025 | 0.029 | 0.030 | 0.030 |
> | NeuralSurv | 0.621 | 1.165 | 0.566 | 3.926 | 16.454 | 22.673 | 7.373 | 87.098 |141.389 |

---

> > ### Comment · Reviewer_52KX · 2025-08-06
> >
> > Thank you for the detailed response, I believe the clarity of the paper increases with the additional explanation and experiments. I confirm my positive score.

---

### Official Review · Reviewer_T13r · 2025-07-18

**Clarity:** 3
**Significance:** 3
**Originality:** 4
**Rating:** 4
**Confidence:** 4

**Summary:**

The paper introduces NeuralSurv, a Bayesian deep learning framework for survival analysis that incorporates uncertainty quantification. The method leverages a two-stage data augmentation scheme involving Polya-Gamma variables and marked Poisson processes to enable efficient posterior estimation. The authors demonstrate superior calibration and competitive discriminative performance compared to state-of-the-art deep survival models on both synthetic and real-world datasets.

**Questions:**

1.	The authors should expand their discussion by referencing more literature on deep survival models, particularly focusing on key results and their limitations.
2.	The sigmoidal hazard model in (1) depends on the distribution of \(\theta\). Since the random variable \(Z(t,x)\) in (5) is also influenced by the distribution of \(\theta\), the same applies to model (1). Two critical questions arise: (a). How sensitive is the model to the prior distribution of \(\theta\) when its dimensionality is high?  (b). Does the model (1), (5), and (6) converge to a fixed model as the dimension of \(\theta\) approaches infinity?
3.	What is the connection between the proposed model (1), (5), (6) and classical survival models such as the Cox proportional hazards model or the Accelerated Failure Time (AFT) model? Specifically, can the Cox or AFT model be derived as a special case of the proposed framework?
4.	The integral in the likelihood function involves only one variable, which is computationally tractable. However, the data augmentation strategy employed by the authors appears to be more computationally intensive. Could the authors provide a comparative analysis of the computational times for these two implementations?
5.	Could the authors clarify which classes of models are not encompassed by their proposed framework (1), (5), (6)?

**Ethical Concerns:**

["NO or VERY MINOR ethics concerns only"]

**Final Justification:**

I'd like to keep the current score.

**Limitations:**

Yes.

**Paper Formatting Concerns:**

No.

**Quality:**

3

**Strengths And Weaknesses:**

The authors provide a Bayesian uncertainty estimate for survival functions. However, it is unknown how the proposed method relies on the prior distributions.

---

> ### Author Rebuttal · Authors · 2025-07-30
>
> > Q1: The authors should expand their discussion by referencing more literature on deep survival models, particularly focusing on key results and their limitations.
>
> We thank the reviewer for highlighting the need to expand our discussion of related work. We have added the following text to the conclusion to better position our model within the broader landscape of deep survival models. We have focused on the primary areas where our approach provides novelty.
>
> “In contrast to previous approaches in deep survival analysis, which are either constrained to discrete-time settings [43, 28, 13, 24, 26, 32] or lack the ability to provide Bayesian uncertainty quantification [43, 28, 13, 25, 24, 26, 32], \emph{NeuralSurv} introduces a continuous-time modeling framework that naturally incorporates Bayesian inference, enabling both accurate survival predictions and well-calibrated uncertainty estimates.”
>
> Appendix  L remains available as a more detailed overview of existing literature on  machine learning methods in survival analysis.
>
> > Q2: Two critical questions arise: (a). How sensitive is the model to the prior distribution of (\theta) when its dimensionality is high? (b). Does the model (1), (5), and (6) converge to a fixed model as the dimension of (\theta) approaches infinity?
>
>
> We thank the reviewer for these insightful questions. (a) NeuralSurv’s inference algorithm is built on the local linearization approach for Bayesian neural networks [18], which imposes an isotropic unit‐variance Gaussian prior on all network weights, this modeling choice is fixed and not subject to alternative specifications. (b) To date, no theoretical results have been published on the sensitivity of this method to high‐dimensional priors or its convergence as the number of parameters grows without bound. Accordingly, we cannot assess how models (1), (5), and (6) behave in the limit $dim(\theta)\to\infty$, since any such asymptotic guarantees would require convergence proofs for the underlying BNN framework, which are not yet available.
>
> > Q3: What is the connection between the proposed model (1), (5), (6) and classical survival models such as the Cox proportional hazards model or the Accelerated Failure Time (AFT) model? Specifically, can the Cox or AFT model be derived as a special case of the proposed framework?
>
> We thank the reviewer for requesting this clarification. Although the sigmoidal hazard function in equation (1) may appear structurally similar to both the Cox proportional hazards (PH) and AFT models, neither the Cox PH nor the AFT hazard functions arise as special cases of equation (1). In particular, no mathematical transformation renders the sigmoidal hazard function equivalent to either the Cox PH or AFT formulations. Nevertheless, we emphasize that the sigmoidal hazard function falls within a broader class of models that have been widely applied in practice (see answer to Question 5).
>
> > Q4: The integral in the likelihood function involves only one variable, which is computationally tractable. However, the data augmentation strategy employed by the authors appears to be more computationally intensive. Could the authors provide a comparative analysis of the computational times for these two implementations?
>
> We thank the reviewer for this suggestion. While employing a numeric approximation (such as the trapezoidal or quadrature rule) to approximate the cumulative hazard integral does indeed make the likelihood evaluable, this approach presents two key limitations. First, the numerical error depends critically on the grid spacing and placement. If the hazard function exhibits rapid variation across time, a coarse grid may smooth or entirely miss these sharp changes, introducing systematic bias into parameter estimates and undermining both statistical efficiency and validity [A-D]. Second, although numerical integration makes the likelihood explicitly computable, directly using it as a loss in a neural network framework inherently leads to frequentist (MLE) estimation. To perform Bayesian inference via variational inference would require deriving an evidence lower bound (ELBO) that involves integrating over the variational parameter distributions [E]. This would result in a double integral problem. In contrast, our continuous-time framework avoids any double integrals and avoids any integrals over the parameters, enabling closed-form variational updates that are both computationally efficient and statistically principled. We are willing to include further clarification in the text in the data-augmentation section if the reviewer deems it necessary.
>
> [A] Michael Sloma, Fayeq Syed, Mohammedreza Nemati, Kevin S. Xu Proceedings of AAAI Spring Symposium on Survival Prediction - Algorithms, Challenges, and Applications 2021, PMLR 146:118-131, 2021.
>
> [B] Mert Ketenci, Shreyas Bhave, Noemie Elhadad, Adler Perotte Proceedings of the 8th Machine Learning for Healthcare Conference, PMLR 219:360-380, 2023.
>
> [C] Kvamme, H., Borgan, Ø. Continuous and discrete-time survival prediction with neural networks. Lifetime Data Anal 27, 710–736 (2021).
>
> [D] Voelkle, M. C., & Oud, J. H. L. (2012). Continuous time modelling with individually varying time intervals for oscillating and non‐oscillating processes. British Journal of Mathematical and Statistical Psychology, 66(1), 103–126.
>
> [E] Blei, D. M., Kucukelbir, A., & McAuliffe, J. D. (2017). Variational Inference: A Review for Statisticians. Journal of the American Statistical Association, 112(518), 859–877.
>
> > Q5: Could the authors clarify which classes of models are not encompassed by their proposed framework (1), (5), (6)?
>
> Sigmoidal hazard functions were first introduced by [B] and represent a distinct class of models. They have been used in the context of point process modeling and survival analysis. To help orient the reader and highlight existing literature on sigmoidal hazard functions, we have added the following text at the end of the introductory paragraph in Section 2:
>
> “Modeling hazard and intensity functions using sigmoidal transformations is common in applications such as survival analysis [10] and point process models [ 44 , 9, A, B, C]. This approach is popular due to the balance it offers between modeling flexibility and analytical tractability.”
>
> However, we would like to emphasize that, to the best of our knowledge, our work is the first to introduce a sigmoidal hazard parameterization in the context of deep survival analysis, where the flexible function g in equation (1) is modeled as the output of a Bayesian neural network (BNN). This formulation required a novel augmentation scheme to enable tractable inference under the Bayesian framework, which constitutes a key technical contribution of our work (see answer 1 to reviewer Pg1A).
>
> [A] Yee Teh and Vinayak Rao. Gaussian process modulated renewal processes. In Advances in Neural Information Processing Systems, 2011
>
> [B] Ryan Prescott Adams, Iain Murray, and David J. C. MacKay. Tractable nonparametric bayesian inference in poisson processes with gaussian process intensities. ICML ’09, page 9–16. ACM, 2009.
>
> [C] Virginia Aglietti, Edwin V Bonilla, Theodoros Damoulas, and Sally Cripps. Structured variational inference in continuous cox process models. In Advances in Neural Information Processing Systems, volume 32. Curran Associates, Inc., 2019.

---

> > ### Comment · Reviewer_T13r · 2025-08-05
> > **To authors**
> >
> > Thank you for your responses to my questions. I have no further follow-up questions.

---

### Official Review · Reviewer_iZcc · 2025-07-20

**Clarity:** 3
**Significance:** 4
**Originality:** 3
**Rating:** 5
**Confidence:** 3

**Summary:**

The authors introduce a Bayesian neural survival analysis model. They use a neural network with sigmoid activation to modulate a parametric baseline hazard model based on observed covariates, which makes their model highly flexible and able to accommodate settings where the relationship between covariates and risk varies over time. After discussing the likelihood and placing suitable priors on neural network baseline hazard model parameters, respectively, they present a data augmentation strategy in which they represent the sigmoid function using a set of Polya-gamma random variables, then express the intractable double integral in the resulting likelihood as an expectation over a marked Poisson process. They use mean field variational inference to approximate the posterior distribution, noting as a limitation that this assumes neural network parameters are independent of the parameters of their hazard model. They conduct experiments on synthetic datasets, with the number of samples ranging from 25 to 150, followed by 8 real-world datasets, each subsampled to 125 samples. The evaluation shows that their method tends to improve on the concordance index and integrated Brier score compared to a comprehensive set of alternatives, and that the true survival function falls within their 90% credible interval in the synthetic data examples.

**Questions:**

My questions are related to the weaknesses described above. Namely:
- How well (or poorly) does the method scale to larger datasets? Please be clear about this and show runtimes.
- How exactly does the current data augmentation strategy relate to those already proposed in [9, 44, 2]? Please be up front and detailed about this.

**Ethical Concerns:**

["NO or VERY MINOR ethics concerns only"]

**Final Justification:**

The authors have addressed my concerns and clarified the contribution in the data augmentation.

**Limitations:**

The authors have described weaknesses but more could be done particularly around scalability & runtime, as I discuss in the "weaknesses" section.

**Paper Formatting Concerns:**

No concerns

**Quality:**

4

**Strengths And Weaknesses:**

Strengths:
- Addresses a true need for a fully Bayesian yet highly flexible survival analysis method
- Method is flexible and elegant
- Paper is well organized and generally well written
- Experiments are compelling and illustrate the method's effectiveness and advantages for small datasets

Weaknesses:
- The method appears to be limited to very small samples. This is discussed briefly, notably in the conclusions section, and the authors note that extending e.g. to a mini-batch setup is a possible direction for future work. Still, they should expand on the details here and present runtime results for each experiment. If the method is not scalable in its current form, this is OK in my view but should be clearly stated.
- A key benefit of the method is its flexibility in modeling the hazard function, but typically this kind of flexibility is becomes advantageous only for larger datasets, whereas their method is currently constrained to smaller datasets.
- A substantial portion of the paper concerns the data augmentation strategy. The authors note in lines 149-150 that "analagous data augmentation schemes have been proposed in [9, 44, 2]". However, I think they should note this at the beginning of the section rather than the end, and they should also do more to explain how their approach is similar versus different from those already proposed. I think the authors' overall contribution is substantial even if this particular aspect is not very novel, so there is no need to obscure what is established versus new here.
- The initial presentation of \lambda_0 in section 2 is confusing, because it's not clear why or how \lambda_0 depends on x. This becomes clear in 2.2, but I think it would be better to briefly comment on this when initially introducing \lambda_0.

---

> ### Author Rebuttal · Authors · 2025-07-30
>
> > Q1: How well (or poorly) does the method scale to larger datasets? Please be clear about this and show runtimes.
>
> We thank the reviewer for this helpful suggestion. In response, we have added a runtime analysis. We refer the reviewer to Question 2 of Reviewer 52KX.
>
> > Q2: How exactly does the current data augmentation strategy relate to those already proposed in [9, 44, 2]? Please be up front and detailed about this.
>
> We thank the reviewer for requesting a more detailed explanation of the connection between our data augmentation strategy and existing work.
>
> - We added this text at the beginning of Section 3:
>
> “This approach builds on analogous strategies previously applied in other settings, including Bayesian inference for Sigmoid Gaussian Cox Processes [9], nonparametric Hawkes processes [44], and, in the case of Pólya–Gamma augmentation alone, mutually regressive point processes [2]. To the best of our knowledge, this is the first application of such a data augmentation strategy in the context of survival analysis. Furthermore, we are the first to provide a rigorous theoretical framework that establishes the validity of a method belonging to this broader class of augmentation-based approaches (see Theorem 3.1).”
>
>
> - Moreover, we have modified the text after Theorem 3.1:
>
> “The proof of Theorem 3.1 is postponed to Appendix N.1. Existing augmentation schemes approaches [2, 9, 44] do not offer any theoretical guarantees regarding the validity of the methodology. In contrast, Theorem 3.1 provides the first rigorous theoretical framework that establishes the soundness of a method within this class of data augmentation techniques”
>
> > Q3: The initial presentation of \lambda_0 in section 2 is confusing, because it's not clear why or how \lambda_0 depends on x. This becomes clear in 2.2, but I think it would be better to briefly comment on this when initially introducing \lambda_0.
>
> We thank the reviewer for this helpful suggestion. To address it, we have modified the text in Section 2:
>
> “Our goal is to model the hazard function $\lambda$, i.e. the instantaneous event rate at time $t$ conditional on survival to $t$ and covariates $\mathbf{x}$. We employ the sigmoid function $\sigma(z) = 1 / (1 + \exp(-z))$, which maps real-valued inputs to the interval $(0,1)$. The sigmoidal hazard model is constructed as the product of a normalized baseline hazard function ($\lambda_0$) and a modulation function ($\sigma$):
> $$ \lambda(t\mid\mathbf{x}; \phi, g(\cdot;\boldsymbol{\theta})) := \lambda_0(t, \mathbf{x};\phi)\  \sigma(g(t, \mathbf{x};\boldsymbol{\theta})),$$
> where the normalized baseline hazard is given by
> $$ \lambda_0(t, \mathbf{x};\phi) := \frac{\lambda_0(t;\phi)}{Z(t, \mathbf{x})},$$
>
> for the baseline hazard $\lambda_0: R_+\to R_+$,  parametrized by $\phi\in R_{+}$,
> and a normalization factor $Z(t,\mathbf{x})$ that depends on both time and covariates. The term $\lambda_0(t, \mathbf{x};\phi)$ encodes our prior ``best‐guess'' hazard profile over time. The flexible function $g:\mathbb{R}_+\times\mathbb{R}^p\to \mathbb{R}$, parametrized by $\boldsymbol{\theta}\in\mathbb{R}^{m}$, provides a data‐driven adjustment: once passed through the sigmoid, it multiplicatively *attenuates* the baseline hazard, continuously scaling it between zero and $\lambda_0$. The normalization factor $Z(t,\mathbf{x})$ ensures that the overall hazard remains properly scaled after modulation by the sigmoidal function (see Section 2.2 for details).
> ”

---

> > ### Comment · Reviewer_iZcc · 2025-08-04
> >
> > Thanks -- this is helpful. I'll keep my score.

---

### Official Review · Reviewer_QbSK · 2025-07-23

**Clarity:** 2
**Significance:** 3
**Originality:** 4
**Rating:** 3
**Confidence:** 3

**Summary:**

This paper introduces a novel survival deep learning method within a Bayesian framework. Survival analysis is a critical tool for generating personalized time-to-event predictions while effectively handling censoring distributions.

The authors propose a new variational inference algorithm that leverages an augmented data scheme. This scheme utilizes Pólya-Gamma random variables to approximate the sigmoid function and Poisson variables to tractably manage the posterior distribution of their likelihood.

The proposed method, a new Bayesian Neural Network (BNN), demonstrates superior performance on several datasets when evaluated using the Integrated Brier Score (IBS) and the C-index (a discrimination metric). It consistently outperforms other deep learning methods while maintaining scalability.

**Questions:**

Theoretical part:
- Why is a sigmoidal function exclusively assumed to map between 0 and 1? If \lambda_0 is considered the baseline hazard, does the "personalized" prediction only decrease? How then is \lambda_0 interpreted? Why was a Weibull distribution chosen for \lambda_0 instead of adopting the more common approach seen in the Cox model?

Experimental part:
I am willing to change my grade if you could give me an answer on those points:
- Can you do one experiment to show that your method outperforms the standard ML methods on a real-life dataset? Maybe on the KKbox dataset that contains 2.5M rows and is available on Kaggle or via pycox?
- Can you add some calibration measures to your benchmark on the real datasets?
- Can you add SumoNet, Yanamisawa, to the benchmark to understand their calibration power, and their abilities to predict the probabilities and their discriminative power?
- In Table A.3, some C-index are lower than 0.5, which means that those models are worse than randomness (especially DeepHit on different datasets). Could the authors provide an explanation for these results?
- Also, on support, all the IBS are worse than it is usually seen in the literature (e.g. Table 12 in Rindt). Could the authors clarify the methodology used to compute these scores?

**Ethical Concerns:**

["NO or VERY MINOR ethics concerns only"]

**Final Justification:**

I have raised my grade thanks to their answer.
The theoretical weaknesses have been answered, as well as more experiments have been made to justify their method on the calibration viewpoint, and more baselines have been added.

Nevertheless, due to their low capacity to scale, other ML methods that still outperforms their solution, and a weird result for one method (Deephit performance), I am grading the paper with "weak reject".

**Limitations:**

yes

**Quality:**

2

**Strengths And Weaknesses:**

To enhance the accessibility of the paper, it would be beneficial to include a concise introduction to key concepts in survival analysis e.g. the hazard function (\lambda(t)), the survival function (S(t), the censoring distribution)

Strengths:
- Using Bayesian methods with a prior is not really used in the literature, and using BNN is quite a novelty.
- Integrating this method into survival analysis offers practitioners the valuable ability to incorporate domain-specific constraints or prior beliefs, which can significantly enhance model interpretability and robustness.
- A lot of details are provided in the manuscript on the experimental part and on the theoretical part.
- the proofs presented in the appendix are easy to follow, contributing to the overall rigor of the paper.

Weaknesses:
- My main concern pertains to Table A5, where standard machine learning methods consistently outperform the authors' method in terms of both IBS and C-index across all datasets. While the authors claim their method may excel in high-dimensional feature settings, no experimental evidence is provided.
- You may need to clearly write your assumptions and the different key points in your reasonning.
- The Integrated Brier Score (IBS) can be biased due to the unknown censoring distribution inherent in real-life datasets. Other calibration metrics, such as Kaplan-Meier (KM) calibration [1, 2] and D-calibration [1, 2], have been developed in survival analysis to provide a more accurate assessment of calibration and should be considered.
- The authors did not compare their method against approaches that train neural networks using proper scoring rules, such as SumoNet [3] or the method proposed by Yanagisawa [4]. This comparison would provide valuable insights into the relative performance of the proposed BNN.

[1]: Haider, Effective Ways to Build and Evaluate Individual Survival Distributions, 2018

[2]: one implementation in Python of those calibration measures: https://github.com/shi-ang/SurvivalEVAL

[3]: Rindt, Survival Regression with Proper Scoring Rules and Monotonic Neural Networks, 2021

[4]: Yanagisawa, Proper Scoring Rules for Survival Analysis, 2024

---

> ### Author Rebuttal · Authors · 2025-07-31
>
> > Q1: Concise introduction to key survival‐analysis concepts
>
> We appreciate this suggestion and have added a brief overview of primary definitions and quantities at the start of Section 2; Appendix A still provides a more detailed treatment.
>
> > Q2: Concern regarding Table A.5, where standard ML methods outperform NeuralSurv in both IBS and C-index.
>
> We apologize and recognise that interpreting the performance in Table A5 may not be straightforward and requires a nuanced perspective. It is important to note that the methods listed in Table A5, including the widely used Cox proportional hazards model, are designed for structured tabular data and typically do not require regularisation to mitigate overfitting. In our examples with NeuralSurv, we used a MLP with without regularization techniques such as L2 penalties, dropout, or batch normalization. Our primary reason to omit regularisation is to ensure a fair and transparent comparison to other competing neural network survival models. Indeed, you will note that it is not just our model that performs poorly compared to those listed in Table A5, but all the NN approaches, although ours is better performing than the others. To further reassure the reviewer of this, we have conducted an additional analysis in which NeuralSurv is implemented using a single linear layer instead of a full MLP. In this revised experiment, NeuralSurv performs much more comparably to the other methods in Table A5 and, in one case, achieves the best performance. Finally, it is critical to note that no method should work best on all data in these comparisons - some methods are based on proportional hazards, ours on a logistic hazard, and yet others on varying subtle assumption choices. These choices will match some data sets better than others. Our main point is NeuralSurv is, on average, highly competitive with all methods, and even those in Table A5 depending on the neural network choice.
>
> > Q3: While the authors claim their method may excel in high-dimensional feature settings, no experimental evidence is provided.
>
> We thank the reviewer for raising this point. Our intention was not to claim demonstrated superiority in high-dimensional settings, but rather to highlight a theoretical strength of our framework. Since NeuralSurv employs neural networks as its predictive backbone, it inherits their established capacity to handle high-dimensional inputs, as evidenced in fields like computer vision and NLP. We acknowledge that our current experiments do not include a high-dimensional benchmark. This was due to practical constraints on computational resources, not a limitation of the method itself. Evaluating NeuralSurv in such settings is an important direction for future work.
>
> To avoid overstatement, we have revised the sentence in the introduction from:
>
> “we address this critical need by introducing NeuralSurv, an architecture-agnostic, Bayesian deep-learning framework for survival analysis.”
>
> to the more measured:
>
> “we introduce NeuralSurv, an architecture-agnostic, Bayesian deep - learning framework for survival analysis which integrates with modern deep learning architectures.“
>
>  > Q4: Clear statement of assumptions
>
> We have added the following opening sentence to Section 2 to make all model assumptions explicit:
>
> “In this section, we outline the main assumptions underlying our model, NeuralSurv.”
>
> Details of the Poisson–Pólya–Gamma augmentation appear in Section 3, and those of the variational inference algorithm in Section 4.
>
> > Q5: Why is a sigmoidal function exclusively assumed to map between 0 and 1? If \lambda_0 is considered the baseline hazard, does the "personalized" prediction only decrease? How then is \lambda_0 interpreted?
>
> We thank the reviewer for this insightful question. In models where a sigmoidal function modulates the hazard, the baseline hazard $\lambda_0(t)$ is typically interpreted as a reference or average hazard, reflecting the expected event rate without covariate adjustment. The sigmoidal transformation acts as a gating function that attenuates the baseline hazard based on individual covariates, thus modeling how personalized risk decreases relative to a worst-case or population-average scenario. This formulation is well established and has been widely adopted in both survival analysis [10] and point process models [9, 44, A, B, C], where it provides a balance between interpretability, flexibility, and computational tractability.
>
> To clarify this modeling choice, we have revised Section 2 of the main text and added the following explanatory sentence and references:
>
> “Modeling hazard and intensity functions using sigmoidal transformations is common in applications such as survival analysis [10] and point process models [ 44 , 9, A, B, C]. This approach is popular due to the balance it offers between modeling flexibility and analytical tractability.”
>
> [A] Teh & Rao, Gaussian process modulated renewal processes. NeurIPS 2011
>
> [B] Adams et al., Tractable nonparametric bayesian inference in poisson processes with gaussian process intensities. ICML 2009.
>
> [C] Aglietti et al., Structured variational inference in continuous cox process models. NeurIPS 2019.
>
> > Q6: Why use a Weibull distribution for \lambda_0 instead of the Cox model?
>
> The semi-parametric Cox model lacks a parametric baseline hazard, relying on partial likelihood methods that hinder Bayesian inference. In contrast, the Weibull models specify a parametric baseline, making them more suitable for Bayesian analysis. We chose the Weibull model for its flexibility, as also used in [10].
>
> > Q7: Can you do one experiment to show that your method outperforms standard ML methods on a real-life dataset?
>
> We thank the reviewer for this suggestion. Our method has already been evaluated on eight real-world datasets (Section 5.2), covering a broad range of survival analysis settings.
> Regarding the KKBox dataset (~2.5M samples), we agree it's a valuable benchmark. However, NeuralSurv is a Bayesian model, which entails higher computational complexity compared to standard frequentist ML methods. As with Gaussian processes, Bayesian inference typically scales poorly with large datasets due to cubic time complexity. While recent advances [1] enable exact GP inference on datasets with up to a million points, they require extensive engineering optimizations that are not yet broadly applicable to general Bayesian deep learning models like ours. Applying NeuralSurv to KKBox would thus require substantial approximations or architectural changes beyond the scope of this work.
>
> [1] Wang et al., Exact Gaussian Processes on a Million Data Points. Advances in Neural Information Processing Systems. NeurIPS 2019.
>
> > Q8: Can you add SumoNet and Yanamisawa to the benchmark to evaluate calibration and predictive performance?
>
> We thank the reviewer for suggesting SumoNet and DQS. We have added both methods to our benchmark comparisons across all experiments. The updated results are included in the revised table in the appendix. We are pleased to report that the first and second positions of NeuralSurv in term of C-index and IPCW IBS remain unchanged after adding the new benchmark methods.
>
> > Q9: Can you add some calibration measures to your benchmark on the real datasets?
>
> We thank the reviewer for this suggestion. To address censoring bias in the IBS metric, we already applied IPCW adjustment as noted in Section 5 (l.249) and Appendix J.3.
>
> In addition, we have now included two calibration metrics, KM calibration and D-calibration, in updated tables consistent with Tables A2–A4 in the Appendix. We are pleased to report that NeuralSurv shows proper calibration under the D-calibration (p-value) metric across all experiments, matching the performance of SumoNet as the only other method with consistent calibration.
> Regarding K-calibration, our method ranks fifth overall across all experiments. The best-performing method in terms of K-calibration is CoxTime.
>
> > Q10: In Table A.3, some C-index values are below 0.5. Why are these models performing worse than random?
>
> C-index values below 0.5 can occur when a model learns spurious or incorrect patterns, leading to risk predictions that are systematically misaligned with actual outcomes. This issue is especially pronounced in small-sample settings, where few comparable pairs make the metric highly sensitive to misrankings. As a result, even random noise or overfitting can yield C-index estimates worse than chance.
>
> > Q11: Also, on SUPPORT, all the IBS are worse than what is typically reported in the literature (e.g., Table 12 in Rindt). Could the authors clarify the methodology used to compute these scores?
>
> We thank the reviewer for this observation. We note a methodological difference: Rindt et al. report the IBS without inverse probability of censoring weighting (IPCW), whereas our evaluation uses IPCW-adjusted IBS, which is more conservative and typically results in lower scores. This metric difference likely contributes to the discrepancy in reported IBS values.
>
> Moreover, in our experiments, we worked with subsampled versions of the datasets for computational reasons, using N=125 in the main analysis (Table A3) and N=250 in the ablation study (Table A4). These are considerably smaller than the full SUPPORT(N=8,873)  dataset, which may explain the lower C-index values. To investigate this further, we reran the benchmark models used by Rindt et al. on the full SUPPORT dataset. As shown below, we found that the Antolini’s C-index values closely matched those in our subset-based analysis, suggesting our results are qualitatively consistent.
> No changes were made to the manuscript.
>
> |Support dataset | Our result | Rindt et al. |
>    |--------|------|------|
> |DeepSurv |   0.611  | 0.609  |
> |CoxTime | 0.609 | 0.613 |
> |CoxCC |0.614 | 0.608 |

---

> > ### Comment · Reviewer_QbSK · 2025-08-05
> >
> > I thank the authors for their answer. I have no follow-up questions.
> > I have raised my grade.

---

### Note · Authors · 2025-08-13

We would like to thank the reviewers for their helpful and constructive feedback and believe in addressing these comments we have strengthened our paper. As detailed in the individual responses we have implemented and responded to all concerns and included any new results in our response. If accepted, we will incorporate the discussions from the rebuttal into the revised manuscript. We are pleased by the positive response to our paper and believe this paper to the an important step in Bayesian deep learning within a survival context.

---

### Decision · Program_Chairs · 2025-09-17

**Decision:**

Accept (poster)

**Comment:**

This work introduces a Bayesian neural network to model survival data, where the hazard function is expressed in a multiplicative form. It consists of a parametric baseline hazard multiplied by a risk function that varies with time, the covariate (X), and an (m)-dimensional parameter (\theta). The parametric baseline hazard function (\lambda_0(t, \phi)) does not involve the covariate (X) (see equations (5) and (1)) and is assumed to follow a Weibull distribution with a one-dimensional parameter (\phi) that has a Gamma prior. This results in a proportional hazards model, where the flexible function ($g(t, x, \theta)$) in the risk component is modeled using a Bayesian neural network (BNN) with an isotropic Gaussian prior on ($\theta$). As the authors claim, this appears to be the first deep survival model to incorporate Bayesian uncertainty quantification.

The paper proposes a novel algorithm that demonstrates superior performance on several synthetic and real-world datasets, consistently outperforming other deep learning methods. The paper is well-written and organized. However, one weakness is the scalability of the method, which seems limited to very small samples. This limitation contradicts the advantage of flexible modeling, which is particularly beneficial for larger datasets. Another weakness is the lack of sensitivity analysis regarding hyperparameters and initializations.

Overall, this is a significant contribution that addresses an important gap in Bayesian machine learning for survival data. I recommend its publication.